# Abrasive Waterjet Machining

**DOI:** 10.3390/ma17133273

**Published:** 2024-07-02

**Authors:** Mohamed Hashish

**Affiliations:** Flow International Corporation, Kent, WA 98032, USA; mhashish@flowcorp.com

**Keywords:** waterjet, abrasive waterjet, cutting, trimming, composites, metal, glass, titanium, drilling, milling, surface finish

## Abstract

The abrasive waterjet machining process was introduced in the 1980s as a new cutting tool; the process has the ability to cut almost any material. Currently, the AWJ process is used in many world-class factories, producing parts for use in daily life. A description of this process and its influencing parameters are first presented in this paper, along with process models for the AWJ tool itself and also for the jet–material interaction. The AWJ material removal process occurs through the high-velocity impact of abrasive particles, whose tips micromachine the material at the microscopic scale, with no thermal or mechanical adverse effects. The macro-characteristics of the cut surface, such as its taper, trailback, and waviness, are discussed, along with methods of improving the geometrical accuracy of the cut parts using these attributes. For example, dynamic angular compensation is used to correct for the taper and undercut in shape cutting. The surface finish is controlled by the cutting speed, hydraulic, and abrasive parameters using software and process models built into the controllers of CNC machines. In addition to shape cutting, edge trimming is presented, with a focus on the carbon fiber composites used in aircraft and automotive structures, where special AWJ tools and manipulators are used. Examples of the precision cutting of microelectronic and solar cell parts are discussed to describe the special techniques that are used, such as machine vision and vacuum-assist, which have been found to be essential to the integrity and accuracy of cut parts. The use of the AWJ machining process was extended to other applications, such as drilling, boring, milling, turning, and surface modification, which are presented in this paper as actual industrial applications. To demonstrate the versatility of the AWJ machining process, the data in this paper were selected to cover a wide range of materials, such as metal, glass, composites, and ceramics, and also a wide range of thicknesses, from 1 mm to 600 mm. The trends of Industry 4.0 and 5.0, AI, and IoT are also presented.

## 1. Introduction

Abrasive waterjet (AWJ) technology was first introduced in the 1980s as a new cutting tool that could cut almost any material [1,2]. Using such tools, hydraulic power is first concentrated to form a high-velocity waterjet. These waterjets, reaching Mach 3–4 velocities relative to the air, are used to accelerate abrasive particles with relatively sharp edges in focusing tubes to form a beam of high-velocity abrasive water. These sharp edges focus the kinetic power of the abrasives on the material, causing high levels of stress, resulting in a micromachining process and material removal.

There is a wide spectrum [3,4] of abrasive waterjet machining research, development, and applications, covering many disciplines, such as materials, fluid mechanics, solid mechanics, physics, tribology, electromechanics, controls, and software. The applications, on the other hand, are spread over some 50 industries, such as aerospace, glass, electronics, shipyard, construction, fabrication, stone and tile, medical, utility, and nuclear fields. Accordingly, in this paper, only the selected areas studied by the author will be discussed. There are, however, other review papers [5,6,7,8,9] and books [10,11] that contain reviews on different AWJ machining aspects, such as parametric studies, modeling, surface finish, and other topics. However, no reference is available that compiles the different machining applications with the models, data, and trends presented in this paper.

Historically, early studies of AWJ technology focused on tool development, modeling tool-related phenomena [12,13], and addressing tool wear characteristics [14]. Other studies have included modeling the mechanics of the AWJ process [15], visualization of the cutting process [16], parametric cutting studies [17], database generation, surface morphology studies [18], and qualifying AWJs for industrial applications. Early applications were mainly driven by market needs, with funding from organizations interested in benefiting from this technology. For example, early applications included pavement concrete cutting, funded by the utility industry [19]; cutting thin sheet metal and composites, funded by the aerospace industry [20]; glass cutting, funded by the optics [21] and automotive industries; and thick metal cutting, funded by the shipyard and military industries. This led to the rapid commercialization of AWJ systems and their spread in many industries.

In this paper, we will start by briefly describing the AWJ system. The AWJ process and its characteristics, parameters, mechanics, and cutting observations are then presented. The applications of the AWJ system in major industries will be briefly listed, followed by detailed discussions on the relevant AWJ machining processes such as shape cutting, trimming, drilling, milling, and turning, with a focus on actual industrial applications. Special AWJ tools and the system requirements for the different applications will be discussed. The basic process models and relationships developed by the author are also presented for the different machining operations.

## 2. AWJ System

An AWJ system consists of several components or platforms that affect its operation and machining results, as shown in Figure 1. These are as follows:UHP pumps are the most upstream component in a waterjet system, where the water enters the pump at ambient pressure and exits the pump into the plumbing system at higher pressure. UHP plumbing is used to transport the pressurized water to the jet-forming nozzle. This plumbing system may consist of tubing, hoses, fittings, swivel joints, and rotary swivels. Pump operating pressures have increased over time, from 200 MPa in the 1980s to over 600 MPa at present.It is important that the high-pressure UHP transmission line tubing is of high strength, is flexible in some areas, and does not cause a significant drop in pressure.The waterjet’s on/off valve is a critical component, and is naturally closed. Pneumatic actuators are used to open the valve. The response time to the open and close commands is about 100–200 milliseconds, and needs to be faster for some high-speed cutting applications, such as cutting food.The cutting head is downstream of the on/off valve and is where the pressure energy is converted to kinetic energy. Special orifices are used to form the waterjets. In AWJ technology, the cutting head design is also important to ensure an optimally sized mixing chamber, no vacuum leaks, a smooth entry for the abrasives, and a concentrically aligned mixing tube.For abrasive feeds and metering, the abrasives are placed in a relatively large hopper and fed using pneumatic pressure to a local hopper mounted on the machine; this local hopper also serves as a metering device for the abrasive flow rate.The motion system is used to manipulate the cutting head (or the workpiece) to affect the cutting process when the jet interacts with the material. These motion systems are most commonly of a gantry or cantilever architecture, and may range from one to five axes of motion. PC-based CNC controllers are typically used to control the interpolating motion of the machine. A robotic arm system is also used as a manipulator.The software is used to enable operators to interface with the machine controller. Models that relate jet parameters to cutting results are used in the front end of the software to aid in identifying the motion kinematics in a transparent way to the operator whose concern may be limited to the CAD portion of the software, such as path planning and nesting.Catcher tanks are used for shape cutting, while point catchers are used for trimming the edge of a part. The catcher needs to be cleaned either periodically or continuously.Other system components that have been used include enclosures, waste removal systems, abrasive recyclers, water chillers for recycling, special fixtures, and a wide range of sensors.

**Figure 1 materials-17-03273-f001:**
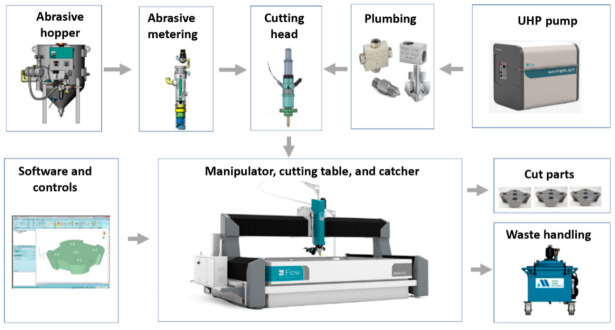
Waterjet system components.

## 3. AWJ Process

### 3.1. Tool Description and Parameters

Figure 2 shows a schematic of the AWJ nozzle. Typical waterjet jet diameters are from 0.08 to 0.5 mm, and typical jet velocities are up to 900 m/s at 400 MPa. The flow of the high-velocity waterjet into the concentrically aligned mixing tube creates a vacuum, which is used to transport abrasives from a hopper to the nozzle abrasive chamber via a suction hose. A typical abrasive material is garnet, which has flow rates from a few grams per minute to 2 kg/min. Medium and fine abrasives (60-mesh to 200-mesh) are most commonly used for metal, glass, and resin composites. The abrasives are accelerated and axially oriented (focused) in the mixing tube, which has a length-to-diameter ratio from 50 to 100. Typical tube diameters are 0.5 to 1.3 mm, with lengths up to 150 mm. A hard and tough material, such as tungsten carbide, is used in the mixing tube to resist erosion. Figure 2 lists the different parameters associated with the AWJ process. The independent parameters of AWJ technology are hydraulic, abrasive, mixing, and kinematic parameters. The hydraulic parameters are the pressure and the orifice size, and they determine the other parameters such as the velocity, water flow rate, power, power density, and jet force. The abrasive parameters include the abrasive material, particle size, and abrasive flow rate. The mixing parameters include the mixing tube length and diameter. The kinematic parameters for cutting are the cutting speed, angle, and the standoff distance. There are other parameters related to the design of the cutting head itself, such as the mixing chamber size and abrasive entry geometry. The effects of many of these parameters are discussed in several sections in this paper.

### 3.2. AWJ Tool Characteristics

Water, abrasives, and air flow through an AWJ nozzle to form an AWJ tool. In this section, we describe the characteristics of these three media and their interactions, such as water/abrasive mixing and mixing tube wear.

#### 3.2.1. Water: Orifices and Flow Parameters

A high-velocity waterjet is formed by an orifice made out of sapphire, ruby, natural diamond, or synthetic mono- or polycrystalline diamond, as shown in Figure 3. It is important for the orifice to be composed of a hard material to resist wear by the high-velocity waterjet, and also to withstand the fatigue stresses due to on/off cycles. The edge must be sharp, with a minimal filet radius, to form high-coherency waterjets, as this increases the jet’s power density. The orifices are sealed to the metallic mount either by polymeric or metallic seals. Natural diamond orifices are sintered into the mount. Upstream water filters may be used to prevent any solid particles from chipping the edge. The jet coherency also depends on the upstream turbulence level. Accordingly, the waterbody and the on/off valve upstream of the orifice should not contribute to turbulence through improper sizing. Figure 4 shows examples of waterjets with three upstream conditions.

Exploring waterjet hydraulic parameters, Hashish [3] presented the effect of compressibility on the jet velocity and other flow parameters, using an equation of state that matches Bridgman’s [22] data. However, in this paper, we present the simple waterjet velocity, $V_{w}$, for incompressible flow based on Bernoulli’s equation, which is as follows:(1)$$V_{w}=\sqrt{\frac{2\;P}{\rho_{o}}}$$

The water flow rate can be determined from this velocity; the orifice cross-sectional area, $A_{o}$; and a coefficient of discharge, $C_{d}$, as shown below:(2)$$Q=C_{d}A_{o}\sqrt{\frac{2P}{\rho_{o}}}$$

The coefficient of discharge accounts for friction and compressibility. The waterjet kinetic power, $E_{w}$ [23], can be derived from the above equations to yield the following:(3)$$E_{w}=\frac{\pi }{\sqrt{8\rho_{o}}}\;\;C_{d}\;d_{n}^{2}\;P^{1.5}$$

The relationships between the waterjet pressure, flow rate, orifice size, and power are shown in Figure 5.

The power density of the waterjet is its power divided by its cross-sectional area. From the above equations, the power density of a waterjet can be expressed as follows:(4)$$E_{wd}=\sqrt{\frac{2}{\rho_{o}}}\;\;C_{d}\;\;P^{1.5}$$

The waterjet power density is in the order of 107 W/cm^2^, which is equal to some lasers [23]. Equation (4) highlights the importance of pressure on the power density. Increasing the pressure from 400 MPa to 600 MPa, a 50% increase in pressure, will increase the power density by 83%. The power density also depends on the orifice coefficient of discharge. For example, an orifice with a *C_d_* of 0.7 has 16% more power than an orifice with *C_d_* = 0.6. This places an important requirement on orifice edge quality to be of a higher coefficient of discharge.

#### 3.2.2. Abrasives

Among the abrasives used in AWJ tools are garnet, aluminum oxide, glass beads, olivine sand, steel grit, chilled iron, and copper slag. The most commonly used abrasive is garnet, due to its availability, safety, and properties. The most important properties of abrasives are hardness, specific gravity, size, shape, and frangibility, as shown in Table 1.

Hardness: A common relative scale for hardness is the Mohs’ hardness scale, which ranges from 1 to 10, with 1 being the softest (talc) and 10 being the hardest (diamond).Specific gravity: The specific gravity is a relevant property for momentum transfer, acceleration, and cutting. Heavier abrasives will not be easily deflected off the material by water flowing over the cut surface.Size: For particle size, the US Sieve Series and Tyler Standard Sieve Series are used to classify abrasives. The most common mesh opening sizes for these scales are given in Table 1, and provide an indication of particle sizes. The abrasive particle size affects both the flowability and the cutting results. Fine abrasives do not flow well in abrasive feed lines; thus, lines may plug, and they may not be as effective in cutting.Shape: The abrasive particle shape can be characterized by a few parameters, such as the sphericity and roundness, which are the most commonly known indices; see Figure 6 for a visualization. The sphericity expresses the degree to which a particle approaches a spherical shape. The roundness refers to the sharpness of the corners and the edges of an abrasive particle. For AWJ cutting, the preferred abrasives are those of a high sphericity and low roundness index.Frangibility: This expresses the ease of breaking and crumbling. For example, an old chocolate chip cookie is more frangible than a fresh one. Frangibility is related to the toughness of a material.

**Table 1 materials-17-03273-t001:** Selected characteristics of common abrasives.

Abrasive	Specific Gravity	Hardness		Rough Relative Cost	Roundness	Sphericity	Frangibility Level (*)
		Knoop	Moh				
Garnet	3.4–4.3	1350	7.5	1	0.45	0.78	medium
Aluminum Oxide	3.95–4.0	2100	9	4–6	0.35	0.78	medium
Silicon carbide	3.2	2500	9.2	3–4	0.31	0.75	medium
Chilled iron	7	520	6	4–5	0.5	0.8	medium
Steel grit	7	500	5	5–7	0.52	0.82	low
Steel shot	7	460	5	4–6	0.89	0.93	low
Copper slag	2.8–3.8	1050	7	0.5	0.5	0.78	high
Silica sand	2.2–2.65	700	7	0.5	0.57	0.79	high
olivine	3.2–4.5	1100	6.5	0.75–1	0.6	0.82	high
Staurolite	3.7–3.8	1275	7.5	0.6–0.7	0.46	0.79	medium
Glass beads	2.5	700	5.5	1.5–2	0.95	0.95	high
Tungsten carbide	14.92	1870		7–10	0.47	0.77	low

(*) Author’s estimate for 80–120 mesh range impacting on steel. (*) depends on grit size and target material.

#### 3.2.3. Water–Abrasive Mixing

The abrasive particle velocity in the mixing tube can be calculated from the momentum equation. Hashish [3,24] developed the following expression for abrasive particle axial velocity:(5)$$\frac{x}{\lambda }=\left[\frac{\nu }{\nu -1}\;-\;\ln\left(\frac{1}{1-\nu }\right)\right]$$
where ν is defined as a velocity ratio, which is the velocity of the abrasive particle, $V_{a}$, to the maximum possible velocity, $V_{max}$. This maximum velocity can be obtained from the ideal momentum transfer between the water and abrasives. It can also be the water velocity, as no more velocity slip will occur between the water and the abrasives. $V_{max}$ can easily be obtained from the overall momentum balance equation, considering no momentum losses:(6)$$V_{max}=\frac{V_{w}}{\left(1+r\right)}$$

The actual abrasive particle velocity, $V_{a}$, can be expressed in terms of a mixing efficiency, $\zeta_{0}$, as follows:(7)$$V_{a}=\zeta_{0}\frac{V_{w}}{\left(1+r\right)}$$

Using the above equations, the following expression [23] can be derived for the abrasive kinetic power, $E_{a};$ as a function of the water kinetic power, $E_{w}$; loading ratio, r; and mixing efficiency, $\zeta_{0}$:(8)$$E_{a}=E_{w}\;\;\zeta_{o}^{2}\;\;\frac{r}{{\left(1+r\right)}^{2}}$$

The power efficacy, η, which is the ratio of the abrasive kinetic power to the waterjet hydraulic power (*E_a_*/*E_w_*), can be expressed as follows:(9)$$\eta =\;\zeta_{o}^{2}\;\;\frac{r}{{\left(1+r\right)}^{2}}$$

This equation highlights the importance of improving the mixing efficiency to minimize the momentum transfer losses for the water to the abrasives. This is affected by the design of the cutting head and how concentrically aligned the waterjet is in the mixing tube [24].

#### 3.2.4. Air Entrainment

The entrainment of air in AWJ nozzles is a key process for ensuring reliable cutting performance. The air is used as a carrier for the abrasives, and thus must be dry, with sufficient momentum and velocity to effectively perform this transport process. Studies have been performed to characterize the jet pump performance of AWJ nozzles [3,13]. Figure 7 shows a graph of AWJ nozzle suction characteristics at different pressures; the curves are typical of jet pump performance.

The air flow rate, *Q_a_*, at standard ambient conditions can be approximated by the following equation, based on the above experimental trends in the suction characteristics:(10)$$Q_{a}=Q_{max}\;\left(1-\frac{\left(P_{a}-P_{v}\;\right)}{{\left(P_{a}-P_{v}\;\right)}_{max}}\right)$$
where ${\left(P_{a}-P_{v}\;\right)}_{max}$ is the vacuum gauge pressure reading with no air flow. Theoretically, this should be equal to *P_a_* if the mixing chamber is completely sealed. $Q_{max}$ is the maximum air flow rate obtained when there are no restrictions on the intake flow and no hose is utilized.

The air entrainment characteristics in feed lines (hoses) can be obtained experimentally by measuring the air flow rate at different pressure differences between the ambient pressure and the suction pressure. Data were generated for several types of feed line materials, lengths, and diameters. A model for this air flow was also developed, assuming isothermal flow conditions, based on the work of Shapiro [18]:(11)$$Q_{a}=A_{h}\sqrt{\frac{P_{r}\;P_{a}\left(2-P_{r}\right)}{2\rho_{a}\left[\ln\frac{1}{\left(1-P_{r}\right)}+\frac{f\;l_{h}}{2\;d_{h}}\right]}}$$
where *P_r_* is the pressure ratio (*P_a_* − *P_v_*)/*P_a_*. Figure 7 shows hose air flow rate characteristics. Solving the above two equations provides the air flow rate that flows into the AWJ cutting head, which is represented in Figure 7 by the intersection points. Selecting conditions with adequate air flow rates is vital for achieving a reliable abrasive feed process. However, it is important to specify that the air velocity in the feed line is a parameter of equal importance. This velocity should exceed a certain threshold for stable flow. The use of vacuum assistance was introduced [19,25,26] to enable jets with weak air entrainment performance to draw more air, and thus provide a more effective abrasive-carrying capacity.

#### 3.2.5. Mixing Tube Wear

The mixing and acceleration of abrasives in an abrasive waterjet (AWJ) nozzle, depicted in Figure 2, creates a severe erosion and abrasion environment. This causes wear to the mixing tube; thus, it must be composed of a wear-resistant material. Many materials were tested during the early development of AWJ technology [27], including tungsten carbide grades, alumina, zirconia, silicon nitride, boron carbide, and diamond sections. Tungsten carbide (WC) was found to be the most suitable due to its combination of toughness and hardness. However, binder materials such as cobalt or nickel cause faster wear; thus, different binders, or no binders at all, have been recommended [27]. This led to the innovation of the ROCTEC^®^ process by Kennam et al. [28], which uses alternative binders such as molybdenum carbide and finer WC powder. This results in an order-of-magnitude improvement in its lifetime. This lifetime is expressed by the growth of the exit diameter, as this affects the kerf width and the effectiveness of the cutting. The wear of the mixing tube is affected by several parameters, such as the abrasive material, abrasive flow rate, pressure, mixing tube diameter relative to the orifice diameter, mixing tube length, and the alignment of the waterjets inside the mixing tube length. Examples of parametric studies on mixing tube wear can be found elsewhere [27,29]. Attempts to extend nozzle life using a lubricating film have also been performed [30]. Figure 8 shows a sample of recent results of mixing tube wear trends. For precision cutting, the wear of the mixing tube must be compensated for in planning the cutting path.

## 4. Cutting Observations

AWJ tools are beam cutting tools, and thus not only share some macro characteristics with other beam tools such as lasers, waterjets, and plasma, but also have some special characteristics related to the mechanics of material removal. The attributes of a cut created using an AWJ tool are presented in this section, followed by a description of the mechanics of material penetration both on micro- and macro-levels.

### 4.1. Geometrical Attributes

As an AWJ tool cuts through and separates the material, three main phenomena are observed [31]. The first is that the jet is deflected opposite to the direction of the motion. This means that the exit of the jet from the material lags behind the point at the top of the material where the jet enters. The distance the exit lags behind the entrance is typically called the trailback, lag, or drag, as shown in Figure 9. In this figure, the jet is moving from the left to the right. It can be observed that the jet–material interface is a curved surface, although this is commonly straight for the range of thicknesses used in aerostructures.

The second phenomenon is that the width of the jet varies along the cut from top to bottom. This difference in width is typically called the taper of the cut. A taper can be either positive or negative, depending on whether the width at the exit of the cut is either smaller or larger than the width at the top, respectively. Typically, the kerf width at the exit side is smaller than that at the entry at practical cutting speeds. Figure 9 shows a cut with a taper.

The third phenomenon is related to the surface finish of the cut. Due to the transient nature of the jet penetration process and jet instability, striations will form along a cut surface, especially near the exit. Figure 9 shows a striated surface cut using AWJ technology.

Additional geometric attributes are detailed below:Bow: A bow or curvature is observed when cutting thick materials at relatively slow speeds. The shape of the kerf takes the shape of the jet.Burr: A cut burr is observed at the bottom surface of a cut, especially for relatively thin and ductile materials such as steel and its high-strength alloys. Instead of cutting through, the jet hydrodynamic force causes bending and deformation, manifesting as burrs.Top edge rounding: Top edge rounding occurs when cutting at relatively large standoff distances.

### 4.2. Integrity Attributes

Several integrity attributes may be associated with AWJ cutting when the wrong set of parameters is used, as follows.

Embedding: It has been observed that embedding occurs on cut surfaces, and the size of the embedded particle can range from one-hundredth to one-tenth of the original particle size. For example, SEM observations showed a 1- to 15-micron embedded particle size range when 150-micron abrasives were used. The measurements showed that approximately 0.02% of the surface area may contain embedded particles. The use of plain waterjets to clean an AWJ-cut surface dramatically reduced the number of embedded abrasives to almost 0% [31]. Several companies, such as Boeing, confirmed that the degree of particle embedding does not affect the weldability or the fatigue life of the parts. On the contrary, AWJ-cut surfaces showed favorable peening effects and thus improvement in fatigue life [32,33].Chipping: Edge chipping occurs when cutting brittle materials such as glass, sapphire, and amorphous metal. The size of the chips is proportional to the grit size.Delamination: This may occur in laminated materials or composites. When the jet is not cutting through with enough momentum and above a certain critical rate, a relatively high hydrodynamic force will be exerted on the face or the step being cut. This causes the jet to spread sideways, inducing layer separation.Frosting and rounding: Top surface edge frosting and rounding may occur due to increased standoff distances as some abrasives on the fringe of the jet cause pitting instead of cutting [31]. The width of this pitted (and rounded) zone increases as the standoff distance increases. Cutting at a shorter standoff distance may eliminate this zone, especially when the mixing tube has a high aspect ratio (~100). Cutting underwater may also eliminate this zone, resulting in sharp cut edges free from pitting. A tube shroud used to flow low-pressure water around the jet, simulating underwater cutting, proved effective in eliminating the frosting or hazing.

### 4.3. Cutting Mechanics

The AWJ material removal process occurs through the impact of abrasive particles on the target material. This impact may occur at shallow angles or large angles. At shallow angles, the abrasive particle acts like a single point tool and plows off a chip. This mode is termed “cutting wear” [34,35]. At large angles of impact, the repeated impact causes material deformation and the eventual chipping of the material. This mode is termed “deformation wear” [36]. This subsection presents a discussion on these material removal modes, as well as a discussion on both the micro- and macromechanics of the AWJ beam cutting process.

#### 4.3.1. Micromechanics (Erosion)

In order to model the AWJ cutting process, a model for single particle erosion is necessary. Hashish [34] developed an improved physical model for the volume removal per particle impact at velocity *V_a_* and at angle α, as visualized in Figure 10. This model is unique as it accounts for particle-specific gravity and roundness, unlike the other existing physical erosion models at that time, such as those developed by Finnie [35] and Bitter [36]. The analysis paralleled Finnie’s and Bitter’s approaches in assuming that at shallow angles, the particle acts as a micro tool tip and plows a chip off the material, or causes deformation and becomes stuck if the angle of impact is larger than a critical value. The simplified erosion model [33] for the erosion volume, δv, is in the following form:(12)$$\delta v=k\;\left(\;\frac{m_{p}\;\rho_{a}^{0.25}\;V_{a}^{2.5}}{\sigma_{f}^{1.25}\;R_{f}^{0.75}}\right)\sin2\alpha \;\sqrt{sin\alpha }$$

This equation is unique in that it accounts for particle density and sharpness. However, the particle hardness is not explicitly expressed. The above expression can be re-expressed using the proportionality symbol, α, as shown below:(13)$$\delta v\;\alpha \;d_{p}^{3}\;\rho_{p}^{1.25}\;V_{a}^{2.5}\;\sigma_{f}^{-1.25}\;R_{f}^{-0.75}$$

The flow strength in the above equation is the material resistance to flow by an erodent particle. Therefore, for a given material with a strength property of s, a harder abrasive particle with hardness, *H_a_*, will remove more material than a softer abrasive material. Simplifying an expression that includes the abrasive hardness, we can replace the material flow strength with a strength property divided by the ratio of abrasive hardness to material hardness, yielding the following equation:(14)$$\delta v\;\alpha \;d_{p}^{\;3}\;\rho_{p}^{1.25}\;V_{a}^{2.5}\;\sigma^{-1.25}\;\frac{H_{a}}{H_{m}}\;R_{f}^{-0.75}$$

Equation (14) can be rewritten as:(15)$$\delta v\;\alpha \left[\;d_{p}^{\;3}\;\rho_{p}^{1.25}\;R_{f}^{-0.75}\;H_{a}\;\right][\sigma^{-1.25}\;H_{m}]\;V_{a}^{2.5}$$

The abrasive properties are shown inside the first brackets in the above equation; the second bracket contains the target material properties; and the last term is the abrasive particle velocity. This modified equation includes all the known properties that affect the erosion process. The coefficients in this equation, although derived analytically, need to be further verified experimentally, despite some of these coefficients agreeing with erosion observations [37].

#### 4.3.2. Macromechanics (Depth of Cut)

A physical AWJ cutting model was developed by Hashish [15,38], based on the division of the cutting front into two zones: the steady-state cutting wear zone at shallow angles of impact at the upper zone, followed by the “wavy” deformation wear mode at large angles of impact. Figure 11 demonstrates this model, supported by visualization studies and wear theories of single particle impact. Henning [39,40,41] conducted studies on kerf development and modeling, supporting prior observations [31].

An equation for the cutting wear depth, $h_{c}$, at the top of the kerf, was derived by Hashish [15], as follows:(16)$$h_{c}=\frac{\;d_{m}}{2.5}\;{\left(\frac{14{\dot{m}}_{a}}{\pi \;\;u\;d_{m\;}^{2}\;\rho_{a}}\right)}^{2.5}\;\;\;\frac{V_{a}}{V_{i}}$$

The intrinsic velocity, $V_{i}$, combines both particle and material characteristics. It is expressed [15] as follows:(17)$$V_{i}={\left(\frac{3\sigma_{f}R_{f}^{3}}{\rho_{a}}\right)}^{1/2}$$
where *R_f_* is a particle roundness factor defined by *R_f_ = d_c_/d_p_*. The equation for deformation wear depth, $h_{d}$, was also derived by Hashish [8], incorporating a threshold particle velocity, $V_{c}$; this equation is expressed as follows:(18)$$h_{d}=\frac{1}{\frac{\pi \;d_{m}\;\sigma_{f}\;u}{{\dot{m}}_{a}\;{\left(V_{a}-V_{c}\right)}^{2}}+\frac{C_{f}}{d_{m}\;\;}\frac{V_{a}}{\left[V_{a}-V_{c}\right]}}$$

The total depth of cut h is the sum of $h_{c}$ and $h_{d}$. It should be noted that at relatively high traverse rates, no steady zone, $h_{c}$, will be established. In this case, the whole cutting action will belong to the deformation wear zone, and its attributes will cover the entire cut surface. Hashish posited that the critical rate, $u_{c}$, at which this transition occurs is related to the critical impact angle ($\alpha_{c}$) of erosion by particle impact [8]. This angle can be considered a material characteristic. The critical traverse rate can be determined from the following equation:(19)$$u_{c}=\frac{14\;{\dot{m}}_{a}}{\pi \;\rho {\;}_{a\;}d_{m}^{2}}{\left(\alpha_{c}\frac{V_{a}}{\;V_{i}}\right)}^{2.5}$$

In its simplest form, at cutting speeds of *u* >> *u_c_*, as in practical cases, and for $V_{a}$ >> $V_{c}$ and ignoring $C_{f}$ for relatively thin materials, Equation (4) can be reduced as follows:(20)$$h=h_{c}+h_{d}\approx \;\frac{2\;m_{a}^{.}\;V_{a}^{2}}{\pi \;\;\sigma_{f}\;d_{m}\;u\;}$$

This equation shows that the abrasive kinetic power is directly related to the material removal rate. The proportionality factor is the material’s specific energy.

### 4.4. Surface Finish

The surface finish resulting from the AWJ cutting process is characterized by waviness and roughness, as discussed in this subsection.

#### 4.4.1. Surface Waviness

The surface waviness (striations) is the macro-level surface finish of the cut, as described above. It occurs with any beam cutting process, such as AWJ, WJ, and laser technology. Figure 9 and Figure 12 show typical striated (wavy) surfaces produced by AWJ tools. It can be observed that the upper surface of the cut is free from waviness, but still rough due to the abrasive erosion process (micro-level material removal). The hypothesis of the waviness is that the jet–material interface is not steady [15,16]. A step of material moves under the jet until it reaches the bottom of the workpiece. Accordingly, the kerf at the bottom of the cut may be considered as an adjacent “pierce” of the AWJ. The spacing between these bottom surface “pierces” increases as the speed increases, contributing to a wavier surface until these pierces become unconnected and the separation does not occur. The AWJ diameter also changes with depth; reductions in this diameter contribute to a wavier surface.

Hashish [42] developed the following expression for waviness based on this hypothesis, which was found to correlate well with the waviness data:(21)$$\frac{2R_{w}}{d_{j}}=1-{\left\{1-{\left(\pi /4\right)}^{2}{\left[\frac{d_{m}\;h\;u}{0.5m_{a}V_{a}^{2}/\;\sigma_{f}}\right]}^{2}\right\}}^{1/2}$$

#### 4.4.2. Surface Roughness

The surface roughness of an AWJ cut is the smaller scale feature on an AWJ-cut surface [43]; it is related to the micromechanics (erosion) of the solid particles. The characteristics of the abrasive particles, such as the size, shape, and hardness, affect the roughness of the cut. The smaller the abrasive particle, the finer the surface will be. However, smaller particles may not be as effective in material removal; thus, the waviness will increase. Figure 13 shows the surface roughness in the upper zones of an AWJ cut where the surface is free from striations. The surface morphology will be similar to a sand-blasted surface, with surface roughness values proportional to the grit size. The lower surface of the cut will also be rough between the larger-scale striations. However, the surface roughness may vary top to bottom based on the momentum and sharpness of the abrasives.

### 4.5. Trail-Back

Figure 14 shows the trailback data for cutting 300 mm thick titanium 6AL-4V under different conditions [43,44,45]. This shows that the trend is similar but varies based on the parameters, and is generally of a parabolic shape. A simple model for the shape of the trailback is a parabola in the form *t_b_ = k x*^2^, where k is a constant for every curve and *x* is the depth; the model depends on process parameters such as the traverse rate, abrasive flow rate, and jet structure.

To develop a trailback model, Crow and Hashish [46] hypothesized that the material removal process is due to centrifugal force causing an inertial grinding process, similar to the sliding abrasion process, as elucidated by Rabinowicz [47]. The resulting model for the trailback, *x*, as a function of the depth, *h*, is derived as follows:(22)$$\frac{xu}{\nu }=\ln\left[\sec\left(\frac{hu}{\nu }\right)\right]$$
where
(23)$$\nu =\frac{\mu {\dot{m}}_{a}V_{a}^{2}}{\sigma_{f}\;d_{m}}$$

A plot of the above equation is shown in Figure 15. This is a simplified universal steady-state kerf shape that can be used to normalize data for further refinement.

Although this kerfing theory results in an elegant universal kerf shape, it fails to account for particle properties such as density and hardness. Based on observations of the jet–material interface during the cutting of transparent materials, the location of the step over which the centrifugal action takes place changes with time. Accordingly, a collection of curves that describes the locations of several points on the kerf will show the kerf progression if a more complex numerical treatment is used. It was also observed in several studies [16] that the kerf is laterally unstable. A comprehensive kerfing code was developed in another study by Hashish and Crow [48], in which this instability was considered as an unsteady wave. An important feature of numerical treatment is expressing the traverse rate as a function of time. This facilitates the inspection of manipulator motion unsteadiness on the kerf shape and, thus, on striation.

### 4.6. Taper

Hashish and Duplessis [49], based on a jet-spreading model developed by Yanaida [50], expressed the shape of a cut produced by a pure waterjet as follows:(24)$$\frac{w_{e}}{d_{n}\sqrt{R}}=0.335\sqrt{\frac{X}{X_{c}}}{\left[1-\sqrt{\frac{\sigma_{c}}{2P_{1}}\;\frac{X}{X_{c}}}\right]}^{2/3}$$

Figure 16 shows a plot of the effective jet width in terms of the non-dimensional numbers of the above equation: width number = $\frac{w_{e}}{d_{n}\sqrt{R}}$; material strength number = $\frac{\sigma_{c}}{P_{1}}$; and standoff distance number = $\frac{X}{X_{c}}$.

It has been observed that the AWJ-produced kerf profiles are similar to those shown in Figure 17. The picture on the left shows the actual cut part with the different kerf width profiles at different speeds. The rightmost graph is similar to that for a pure waterjet in Figure 16 and shows kerf width data for 150 mm thick titanium. It can be observed that most of the kerf profiles are converging because the traverse rate was too high to produce a divergent shape profile.

Based on this similarity, it is proposed here that the above equation be applied to AWJs with some modifications. The jet hydrodynamic force can be replaced by the abrasive rate of the change in momentum, *m_a_V_a_*, and the material strength, *σ_c_*, can be expressed using the material resistance to abrasive erosion, *σ_f_*. With this adaptation, the following equation results can be presented:(25)$$\frac{w_{e}}{d_{m}\sqrt{R}}=0.335\sqrt{\frac{X}{X_{c}}}{\left[1-\sqrt{\frac{\pi \;d_{m}^{2}\sigma_{f}}{8m_{a}^{.}V_{a}^{2}}\;\frac{X}{X_{c}}}\right]}^{2/3}$$

The characteristic length, *X_c_*, is to be determined experimentally. Using the above equation for predicting the kerf shape may be applicable for the prediction of tapers and their correction.

### 4.7. Cutting Strategy with Trailback, Taper, and Surface Waviness

To overcome the effects of taper and trailback, automatic jet angulation was developed [51]. The taper of the cut can be corrected by tilting the jet perpendicular to the direction of traverse. To correct for the effect of trailback, especially at the corners, a lead angle was used. Figure 18 shows the morphology of AWJ-cut surfaces at different cutting speeds [52], and the general qualitative trends in taper, trailback, and surface waviness as functions of speed are illustrated.

This figure shows general speed zones separated by four critical cutting speeds. The first critical cutting speed, u_1_, is that at which zero taper occurs. Slower speeds than u_1_ will result in divergent cuts with negative taper and no waviness. The second critical cutting speed, u_2_, characterizes the initial waviness formation. Increasing the speed beyond u_2_ will continue to increase taper to a maximum value at the third critical speed, u_3_. Beyond u_3_, the taper will decrease and the surface will be highly wavy and irregular. At speed u_4_, the jet will barely cut through the material, or not cut through completely.

A cut surface at speeds slightly below u_2_ will produce a waviness-free surface similar to, but slightly rougher than, that obtained at speed u_1_. Usually, u_2_ is several times faster than u_1_.

Based on these trends, the cutting speed can be maximized based on the required surface finish, regardless of taper, by tilting the jet normal to the direction of travel to correct for taper on the required side of the kerf. With this approach, taper angles are used to obtain the required accuracy with or without minimal wall taper. Assuming that an acceptable surface finish is Ra, this will identify a cutting speed *u_f_*. The taper obtained at this speed can then be determined, as shown in Figure 19. This will define the taper angle to be used. Figure 20 shows the potential gain in speed due to the use of taper angles.

For shape cutting, the trailback will manifest itself in distortions to the geometry of the cut at the exit side. The sketch in Figure 21 shows an undercut due to the trailback phenomena [52]. The picture in the same figure shows distorted square-shaped cuts at the bottom surface of the material due to trailback and taper.

## 5. Industries and Applications

The AWJ process has been utilized in numerous applications in many industries, the most common of which are as follows:shape cutting;trimming;slicing and turning;milling and grooving;hole making;near-net shaping;surface modification.

In this section, we select some industries that have benefited from AWJ technology in many applications and then discuss detailed applications related to these industries.

### 5.1. Jet Engines

AWJ technology has been applied to the jet engine industry since the beginning of its commercialization in the early 1980s in the cutting of thin sheet metal used in nacelles. Applications grew to include many other functions [53], as shown in Figure 22. Now, with the use of additive manufacturing for many jet engine parts, new applications are emerging, such as UHP densification, powder removal, and surface finish improvements. Some of the applications shown in Figure 22 are discussed in a subsequent section.

### 5.2. Aircraft

Abrasive waterjets (AWJs) have been one of the great enabling and timely tools that has expedited the use of composites since the early 1980s. This is because AWJs offer several advantages over conventional machining methods:higher cutting speeds than routers;no distortion due to limited jet forces and the nature of micromachining action;no heat-affected zones;no delamination, splintering, fraying edges, or any other integrity problems;no subsequent processes are needed;reduced fixturing and tooling;process automation and multiple operations are possible;no dust;versatile for different composites and laminated structures.

Figure 23 shows the continuously rising trend in the use of carbon fiber composites in aerostructures [54,55]. For example, the use of composites on the Boeing 787 and Airbus 350 is about 50% by weight and 90% by volume. In comparison, the Boeing 777, which entered service in 1995, contains only 10% composite structure by weight.

More and more parts are now being made from carbon-fiber-reinforced plastics (CFRPs) due to their superior strength and lightweight characteristics. These parts range in size from relatively large, such as the wing covers and fuselage, to small parts, such as clips and doors. Figure 24 shows a list of the major CFRP parts on the Airbus 350.

### 5.3. Micro Electronics

AWJ applications in the microelectronics industry are rapidly growing. The cutting of printed circuit boards (PCBs) was one of the early pure waterjet cutting applications evaluated by IMB. Recently, elevated pressures up to 690 MPa were demonstrated cleanly cutting through complex PCBs; the recycling of PCBs using AWJ technology is a promising new application. Cutting microSD cards was perhaps the largest scale application for AWJs in microelectronics, where over 50 machines were used to cut millions of microSD cards in the mid-2000s. Cutting the multi-layered carbon fiber composite material for mini hard drive arms has been commercially implemented. Cutting multi-module components (MMCs) has also been demonstrated and applied using AWJs. Small-diameter holes have also been drilled in composites for speaker holes and in an ALN (aluminum oxynitride) material. Smartphone display glass has been cut with AWJs, but work is needed for surface finish improvements. Figure 25 shows some example applications.

AWJ technology has the potential to address more applications in microelectronics when additional progress is obtained, regarding the following developments:AWJ kerf width reduction to the 130-micron range;achieve surface finish in the 10-micron range;pure waterjet dicing of PCBs;high-accuracy milling of glass and sapphire to less than a 10-micron surface finish;pure waterjet trimming of composites without delamination.Some of these advances will be discussed in later sections.

### 5.4. Glass and Optics

Significant waterjet technology developments have been demonstrated in glass cutting for several applications. Among these applications are automotive glass, laminated glass in aircraft windshields, glass in artwork, a wide range of optical system sizes for land-based and space telescopes, solar panels, and recently, in microelectronics for display glass, as mentioned above. Early studies addressed the precision machining and the minimization of subsurface damage. Later studies addressed the precision deep cutting and pocket milling for lightweighting. Figure 26 shows examples of glass cutting, milling, and drilling operations.

### 5.5. Stone and Tile

AWJ was demonstrated during the early stages of its development for cutting pavement concrete and stone. The shape cutting of the dimension stone followed quickly in Europe, and especially in Italy, where the marble industry quickly capitalized on advantages in the AWJ process. The use of vacuum assistance for piercing was critically enabling at that time. Intricate inlays were cut to make furniture pieces, floors with artistic designs, and wall decorations in large mosques. The AWJ technology soon became the standard for cutting countertops in almost all stone shops. Hybrid AWJ–mechanical systems were developed to use both the AWJ process and conventional finishing tools. Now, robotic arms are also used in the stone industry for hybrid processing. The software for vein matching is used in many facilities. Examples of AWJ shape cuts in stone are shown in Figure 27.

### 5.6. Military

A wide range of military applications have benefited from the AWJ process for cutting metal, ceramics, composites, and the demilitarization of munition. The US DoD has sponsored many projects directly or through its subcontractors to develop generic applications such as the turning, milling, and cutting of materials. The milling of isogrid structures for rocket skin lightweighting was among the first studies funded by the military. Other example applications are the milling of titanium aluminide heat tiles [56], the near-net shaping of Ti-Al blades, small hole drilling in alumina and SiC/SiC composites, the drilling of small-diameter holes in jet engine components, the cutting of titanium and steel tank armor plates, and the cutting of boron carbide, Kevlar, and ceramic materials for personal armor shields. The decommissioning of the World War II M55 rocket was conducted using ammonia abrasive jets, as ammonia could dissolve the propellent material, and through complex chemical processes, the rocket material would be converted to harmless salt [57].

### 5.7. Automotive

Robotic pure waterjets have been used for the trimming of many interior automotive features [58], such as carpets, headliners, and the dashboard instruments. Many configurations, such as pedestals, wall mounts, or ceiling-mounted multi-robot cells, are common in many automotive factories. Other applications use shuttle or turn tables to load and cut the parts inside an enclosed cell. Engine gaskets made of different materials also represent a common waterjet application. Window glass trimming was the first automotive application to be addressed using the AWJ process in the early 1980s, when the Libby-Owens-Ford company implemented the AWJ process for beveling material edges. The waterjet trimming of composite helmets has become a standard industry process. Today, with the introduction of thermoplastic carbon fiber composite materials in automotive body parts, the AWJ process is being used for the trimming of parts that pure waterjets are not capable of trimming. For example, many carbon fiber composite parts on the BMW i-3 electric vehicle are trimmed with robotic AWJ systems. Figure 28 shows examples of automotive applications.

## 6. AWJ Machining Processes

AWJ tools have been used to perform different machining operations, such as shape cutting, trimming, drilling, and milling. In this section, we discuss the observations, trends, and advancements in these areas.

### 6.1. Shape Cutting

Shape cutting most commonly requires piercing first, and then contour cutting. Piercing clearly needs to be performed outside the required shape to be cut out of a larger sheet, plate, or slab. Then, AWJ advances to cut the contours of the part and exits that path at the end of the cut, or continues a short distance to blend in the ends and cuts the trailback area at the bottom that may prevent complete separation. The part which is separate must be supported so as not to fall in the catcher tank. Slats, which are thin, deep strips of metal, are used to support the workpiece and the cutouts similar to some laser and plasma cutting machines. If the parts are smaller than the spacing between the slats, then tabs are used to keep the cutouts connected to the sheet and then knock the cut parts off the sheet. This practice is mainly used with metals. To facilitate the programming of shape cutting, software was developed based on predictive models to let the operators only identify the material type, thickness, and the required cut finish. The software identifies the speeds and offsets to perform the cutting operation.

To improve the accuracy and precision of AWJ shape cutting, several techniques have been developed using sensors and machine kinematics. In the following sections, we present studies on these enhancements, including the following:kerf width compensation;terrain following;vision-assisted cutting;first article compensation;kerf taper compensation;corner geometry compensation;three-dimensional wrist with SOD sensor.

#### 6.1.1. Thin Materials

There is no specific definition for thin materials as opposed to thick materials. Thus, in this section, we discuss three examples of thin material cutting with different requirements.

##### Thin Sheet Metal

One of the earliest AWJ studies in the mid-1980s, performed by Hashish et al. [20], was sponsored by the Wright Patterson AFB through Rohr Industries to establish a database for thin sheet metal cutting used in aircraft nacelles and other parts. The cutting tests consisted of cutting the materials under different parameters, such as varied orifice mixing tube combinations, pressures, abrasive flow rates, and abrasive particle sizes. Mixing tube wear data were also generated in this study. The materials considered in this study were Inconel, 718, Inconel, 625, Titanium 6Al-4V, Titanium, Hastelloy, Chromoly 4130, CRES 301 full-hard and half-hard, and CRES 15-7 Ph. The measurements included kerf tape at different speeds, burr height at the exit, and the waviness at the bottom of the cut. Figure 29 shows the sample results on the burr height at different speeds and abrasive flow rates. As shown, the burr was reduced by increasing the abrasive flow rate. Notably, the parameters used are not as optimal as those currently used in industry. One of the most important findings was the microcharacteristics of the cut edges determined from SEM images, see Figure 29 (right). The cuts showed no evidence of any effects, such as heat or deformation, unlike laser and shear cutting, which exhibited these issues correspondingly.

A CNC 3-axis system was then provided to incorporate a vision camera and a terrain follower, see Figure 30. The camera was used to periodically check the kerf width for compensation. This was necessary at that time, as the mixing tube wore out rapidly. The terrain follower used four eddy current sensors around the jet to keep the standoff distance fixed, as large sheets of thin metal tend to warp. This was the first AWJ system that used these features.

##### Vent Screens

Another example of thin metal cutting is the cutting of titanium vent screens for use on the F-22 fighter aircraft [59]. The AWJ process was selected over laser cutting to avoid the generation of heat-affected zones and screen warpage. Between 7000 and 30,000 shaped holes were drilled in 35 screen parts made of 4.8 mm thick titanium 6AL-4V. The holes were 2.3 mm on the side and square- or rhombus-shaped, with 0.5 mm corner radii and edge-to-edge spacing of 0.64 mm. About 14 measurements needed to be made for every hole, including the side lengths, corner radii, corner angles, and spacing between holes on all sides. A 0.23/0.50 mm orifice/mixing tube combination was found to be most suitable. The matrices of holes were cut at different speeds to select an acceptable value for cutting full-scale screens to accurate specifications. A speed of 38 mm/min at 345 MPa and 120-mesh, 3.4 g/s garnet abrasives were selected to cut geometrically accepted screen holes with tapers of less than 0.015 mm. Figure 31 shows some machine holes cut at two different speeds.

Initially, optical calibers were used to manually check the hole features using a video camera with adjustable horizontal and vertical lines. The measured data were downloaded to a computer. The corner radii were also measured with optical calipers, using three points on the curves. For production cutting, the vision inspection system was automated, and the measurements were used to adjust the size or the location of the adjacent holes. Figure 32 shows the screens on the F22 aircraft. Some screens were drilled with holes at 45 degrees to the surface, which increased the cut thickness; thus, a slower cutting speed of 19 mm/min was used.

It was necessary to be able to restart cutting after a process interruption. This was tested to improve the robustness of the AWJ process. For example, mixing tube clogging, abrasive loss or plug, sudden loss of pressure, and chipped orifice were simulated. None of these issues caused damage to the required hole, but needed to be detected to stop the operation. Vacuum sensors, an acoustic sensor, and a pressure transducer were used for this purpose. Additionally, cutting head flushing was used to periodically clean the mixing chamber. All the F-22 screens were machined using the AWJ process.

##### Solar Glass

An example of precision cutting using an articulated arm robot is that used for solar cell glass concentrators. The AWJ process is used to cut the glass for the concentrator photovoltaic (CPV) systems. Here, it was required to convert a glass dome shape about 355 mm in diameter into a square shape with sides of 263 mm and a 51 mm hole diameter in the middle. The corners of the square were also cut and beveled as shown. This meant that the AWJ tool would cut the sides and the hole while the jet was vertical, and then cut the bevel while the jet was horizontal (sidefire). Accordingly, a robotic arm was used for this machining process. A special vacuum fixture was used to incorporate the sidefire catchers at the corners for the bevel cuts. A takt time of one minute was met with a yield exceeding 99%. Tests on piercing glass at 400 MPa were needed to meet the takt time so as not to lower the pressure for piercing. It was found that increasing the standoff distance to at least 25 mm and using vacuum assistance to draw abrasives before firing the waterjet were essential for the reliable piercing of glass at 400 MPa pressure. This enabled the sequence of first cutting the bevels, which required four pierces, as shown in Figure 33. An AWJ tool with 0.25/0.75 mm combination was used with 6 g/s of 120-mesh garnet to cut at a speed of 63 mm/s and 21 mm/s for the central hole.

To obtain accurate parts using the robot arm, two strategies were followed. The first was to use first article compensation. In this method, a part is cut, measured, and the error function is used for compensation. This method was later replaced with volumetric compensation of the cut area, to support offline programming. In this case, a map of the actual versus measurement locations was developed for the tool center point using laser tracking. The data were used in the CNC controller to modify the path so that accurate parts are produced. With these methods, the tightest part tolerance of 0.2 mm was met.

##### Display Glass

The main requirements for cutting display glass is the surface finishing of the edge and ensuring that it is free from subsurface damage, although the cutting speeds are relatively high to meet most production demands. A study was undertaken to cut display glass, such as Corning Gorilla glass, which is chemically strengthened and highly scratch-resistant, by creating a compression layer on the surface. This makes it more difficult for the AWJ process which, upon cutting, relieves the stresses associated with cracking and chipping. To produce fine edges, fine abrasives need to be used, such as 320- or 400-mesh. A commercial microblaster was used to feed 400-mesh aluminum oxide abrasives in a cutting head. Although the use of these fine abrasives commonly reduced the chip size from 100 microns to about 23 microns, as shown in Figure 34 below, it was also observed that, occasionally, a large chip was formed of 100 microns in size or more. These random large chips are attributed to the temporary abrasive flow rate stoppage, which subjects the glass to a waterjet impact over a relatively longer period of time. In fact, because the abrasive flow is sparse and only about 5% by volume of the jet, thus, 5% of the time while cutting, and increasing the speed will increase the distance between two successive impacts. Therefore, the likelihood of chipping increases. Further research is needed to maximize the abrasive concentration, which suggests that abrasive suspension jets may offer a superior performance.

##### MicroSD Singulation

MicroSD cards are thin electronic packages of approximately 0.76 mm (0.030 inch) in thickness, as depicted in Figure 35. To singulate a strip that contains several microSD cards, both waterjets and diamond saws have been used. A waterjet can cut a shaped contour with small corner radii; diamond saws cut straight lines to separate the microSD cards using existing handling systems.

Cutting these components requires high cutting speed, edge quality, accuracy, and precision. Some internal corners on flash cards necessitate the use of a mixing tube with a maximum width of 0.40 mm, which has not always been commercially available. The accuracy needed was better than 0.1 m, and a Cpk of 1.33 or better was also specified. A mixing tube of 0.38 mm in diameter and 63 mm in length was developed for this project by the supplier; additionally, sieved 220-mesh abrasives of tight particle size distribution were made available by the abrasive supplier. To meet volume production quantities, cut speeds of 45 mm/s on average were needed, which was proven to be well within the capability of the AWJ process. A 0.13 mm orifice at 379 MPa pressure and 0.75 g/s garnet abrasives were selected for cutting because a surface finish of less than 4 microns was specified.

Machine vision was used to locate the positions and inspect the cut results to meet the accuracy requirements for high volume production [52,60,61]. The AWJ process here consisted of loading five strips (300 mm × 100 mm) on a tray, loading the tray on a vacuum chuck, using a vision camera to locate the centers of the selected fiducials as the cut patterns with reference to these fiducials, then starting holes are drilled in all selected locations on the five trays, cutting the perimeter, inspecting the cuts with the vision camera for kerf width and some dimensions, removing the tray from the vacuum chuck after releasing the vacuum, loading the waterjet-cut strips for diamond saw cutting, and finally, inspecting some selected components to determine the process capability index, CPK. To ensure the reliability of the AWJ process for 24/7 operation, the following features were implemented: kerf width compensation using both vision and mixing tube wear characteristics; vacuum assistance for piercing holes; flushing for periodic mixing chamber cleaning to prevent clogging; an orifice vacuum sensor to detect changes in orifice health; and finally, using a machine tool collet to ensure the repeatability of a tool center point and axial alignment. A three-axis machine with high repeatability, accuracy, and two-location cutting was used.

##### Vision-Assisted Catalyst Slotting

Electrically heated catalysts (EHCs) are commonly used in modern gasoline and diesel vehicles equipped with advanced emissions control systems [58]. EHCs are particularly effective in reducing emissions during the critical cold-start phase, when conventional catalytic converters are less efficient. Commonly used materials include ceramic materials such as alumina (aluminum oxide) and ceria (cerium oxide), which have high surface areas and thermal stability.

Work in the early 1990s was performed to reduce the light-off times by cutting a serpentine path in an EHC element to increase its electrical resistivity, but without creating hotspots. The AWJ process was used to precisely cut between cells. A 0.5 mm diameter AWJ was used in order to cut along a cell line without affecting the surrounding cell walls. To improve the cutting quality, a jet lead angle of about 10 degrees was incorporated. A machine vision system was used to generate the required CNC program path for the EHC element in one station. The EHC was then mounted in another station for cutting. Machine vision was needed because the cell pattern was not consistent from one part to another, or from one side of an EHC element to the other side. Figure 36 shows an EHC element before and after cutting.

#### 6.1.2. Thick Materials

AWJ technology has demonstrated the capability of cutting thick materials such as 300 mm thick concrete [62,63], titanium, aluminum, and 600 mm thick glass. However, cutting thick materials may only be conducted for roughing or for obtaining accurate parts. The basic strategy for the accurate cutting of thick materials is related to both the process and kinematic issues:Process Issues: The AWJ process parameters should be selected to cut the required depth at the required speed and surface quality. A special AWJ cutting head may be needed to maximize the kinetic power of the abrasives, and also to collimate the AWJ beam to focus this power.Kinematic Issues: It has been found that angulating the jet with taper and lead angles is critical for obtaining accurate parts. This kinematic manipulation corrects for process physical phenomena, namely, kerf taper and trailback.

##### Kinematic Manipulation

Figure 37 shows data for 300 mm thick glass cutting at a normal impact angle (upper curve). When this curve is rotated to simulate the use of lead angles, it was observed that at a 3.39-degree lead angle, the maximum trailback occurred near the middle of the sample, at its greatest thickness. The length of this trailback was about 4 mm at this angle versus 117 mm when no lead angle was used. For shallow cuts, where this shape is close to a straight line, the trailback can be reduced to nearly zero when the appropriate lead angle is used.

Similar to rotating the kerf shape using a lead angle, the kerf profile can be rotated to minimize the taper on one side of the cut. Figure 38 shows a typical kerf width shape for 100 mm thick titanium. By applying a tilt angle of 1 degree, the taper was reduced on one side of the cut from about 2 mm to less than 0.5 mm. The accuracy of tilting the jet should be in the order of 0.1 degrees in order to accurately obtain the desired wall straightness. The other side of the cut thus became more divergent, to about 4 mm deviation, as shown in Figure 38.

##### Process Parameters

The selection of process parameters for thick material cutting should be the first step before kinematic manipulation. There should be sufficient jet power and abrasive flow rate to achieve deep penetration. The abrasive particle size must be as large as the surface finish specifications, which will enable larger abrasive particles to carry the momentum further. To maximize the momentum transfer from the waterjet to the abrasives, the mixing tube must be of sufficient length. Table 2 shows the minimal mixing tube lengths, using a mixing efficiency of 90%, based on analyses conducted by Hashish [3] and elaborated upon in Section 3.2.3.

Larger particles require longer mixing tubes; moreover, as the abrasive flow rate increases, shorter tubes can be used to attain the maximum velocity. For the commonly used 100-mesh abrasives, for example, a mixing tube length of only 33 mm is required for an abrasive loading ratio of 0.12. The typical mixing tube length used in industry for this case, however, is about 76 mm. The additional length is used to collimate the jet and to increase the ratio of the abrasive velocity to the water velocity to about 0.95. Notably, the additional 43 mm of mixing tube length contributes only 5% to the maximum possible velocity, as can be determined from the equations in Section 3.2.3.

##### Thick Glass Cutting

The precision cutting of 300 mm thick glass cores for large optical space telescopes was required for the lightweighting of Corning’s ULE thick glass to replace diamond wire saws. In this context, lightweighting is simply the conversion of a solid boule of glass into a hexagonal or triangular honeycomb structure, which significantly reduces its weight.

The task was to cut 300 mm thick ULE glass to produce a triangular or hexagonal honeycomb structure with relatively thin webs of 1.9 mm (+0.38 mm, −0.13 mm) in thickness. The initial parametric studies suggested that a relatively high power (larger orifice size) in the range of 100 to 150 kW was needed to reach the 300 mm depths with an acceptable taper. Orifices with 0.81 mm at 345 MPa were tested. Additionally, the mixing tube diameter was increased to 3.2 and 4.0 mm, while its length was increased from 100–150 mm to 300–600 mm to better collimate the AWJ and to maximize the kinetic power of the abrasives. Here, 50-mesh garnet abrasives were needed, as their larger particle mass could carry momentum down to the 300 mm depth. The results of the initial tests showed that the AWJ process may accomplish this challenge. Figure 39 shows the effect of speed on the kerf taper and trailback. One of the most important parameters that was identified as a necessary manipulator requirement was jet tilting to overcome the kerf taper, as well as the trailback effect around corners. These findings were implemented for thick glass cutting and then further developed for common precision cutting.

Figure 40 shows a setup that was used to cut small-scale demonstration parts for process validation using a specially designed tilt wrist to hold 600 mm long mixing tubes.

It was also necessary to rapidly pierce holes in the 300 mm thick glass, as the conventional piercing of this thickness could take over 20 min per hole. This is because the rate of penetration is gradually reduced due to the return flow of the jet [64,65]. Thus, pressure ramping and tool orbiting were applied. With these methods, holes were drilled in less than 5 min. Handling the glass residuals out of the glass block (boule) after separating it required using expandable foam inserts in the kerf to keep the residual from moving. It was then lifted vertically after affixing a hook in the drilled hole. To avoid the trailback effect near the end of the cut, a mechanical “fuse” approach was followed by creating a cut at the beginning of the shape cutting, as shown on the left side of Figure 41.

Cutting 600 mm thick glass was also tested, yielding similar results, but using a more powerful jet and 900 mm long mixing tube. Figure 41 shows a process verification cut to cut out a thin-walled shape that looks like an airfoil to demonstrate the tight-angle and curved wall cutting.

##### Thick Titanium Cutting (Shaping)

Thick titanium usage is found in jet engines, aircraft, and military tanks, to name a few applications. Some data on thick glass cutting have been presented in this study to illustrate the effect of angular compensation for kerf taper and trailback for precision cutting. Figure 42 shows the maximum cutting speeds for different thicknesses of titanium 6Al-4V and the associated trailbacks. The waviness of the cut surface is shown in Figure 43.

Economic analysis was conducted for the implementation of the AWJ process for roughing 108 mm thick integrated rotor blades used in jet engines, such as the F-135. The depth of cut varies from 149 to 170 mm. The advantage is that slugs with a higher residual value than chips will be produced. The slugs of material to be removed needed three cuts in order to unlock the material, as shown in Figure 44. The roughed-out shape was specified not to exceed 0.050 over the final shape. In production, a 650 mm rotor with 150 mm thickness was rouged out in about 17 h at 400 MPa pressure. It was demonstrated that a rotor with an approximate 1 m diameter requires 38 h at 600 MPa pressure, with a total operating cost of about USD 1500.

### 6.2. Edge Trimming (Composites)

Trimming is a process where cutting occurs on the perimeter of a part. Accordingly, piercing may not be needed. This section focuses on trimming carbon fiber composites, as this is the most dominant trimming application. The AWJ cutting and trimming of carbon fiber composites has become a standard process due to its advantages over solid tools. Typical problems that have been encountered with conventional solid tools are both related to surface finish and integrity. The common integrity issues are shown in Figure 45, and were further elaborated upon in prior studies by Hashish [66,67]. Environmentally, solid tools generate dust and carbon powder, which affect electrical systems and personnel. In addition, solid tools may not be able to access tight spaces as required due to the bulkiness of the spindles. Downtime is also associated with frequent tool change as routers and drills wear. The AWJ technology solves these problems, demonstrating its acceptability for machining CFRP parts for a wide range of applications, as discussed below.

#### 6.2.1. Composite Stringer Trimming

Carbon fiber composite stringers are commonly I-Beam or T-Beam stiffeners used in the structures of aircraft wings, fuselage, floors, doors, and other parts. These stringers need to be trimmed to their final sizes [68]. Based on the design of the aircraft, stringers may be made as free-standing or already attached to the part: a wing, for example. Boeing 787 wing stringers, for example, are trimmed before they are bonded to the wing cover panels, whereas Airbus 350 stringers are trimmed after they have been attached to the wing cover. To trim the stringers using AWJ tools, special cutting heads were used to fit between the flanges. In order to trim a free-standing stringer, at least four cuts are needed; however, for the Boeing 787 stringers, six cutting heads are used simultaneously to trim and bevel the stringer. Catcher cups have also been developed to catch the jet after cutting through the stringer, as depicted in Figure 46.

It is a requirement that the surface finish is 10 microns or better. It was shown as an example that a sidefire cutting head with a 0.25 mm orifice and using a 4.2 g/s abrasive flow rate could cut 15.5 mm thick sections at 7.9 mm/s at 380 MPa pressure, producing a surface finish of better than 10 microns when using 120-mesh abrasives. At this speed, the taper of the cut is about 0.8 degrees. Under the same conditions, 10.6 mm thick sections can be cut at 2.3 mm/s, and 8.9 mm thick sections can be cut at 13 mm/s.

A special AWJ machine has been developed to trim the four edges of the flanges of freestanding I-beam-style stringers. In addition, the lower flanges are beveled at a 45-degree angle. Six sidefire nozzles are simultaneously used to trim and bevel the stingers. Figure 47 (right) shows a traveling AWJ trimming system to trim stringers, while Figure 47 (left) shows a special end effector for trimming attached stringers.

Trimming co-cured stringers, such as those on the wing of the AB350, a sidefire AWJ nozzle may be used to fit in the space between the stringers; a catcher cup fits between the two stringers on the opposite side. A regular AWJ cutting head size may also be used for more efficient cutting if the space between the stringers can accommodate the cutting head. To trim near the root of the stringer, a special type of AWJ with a thin profile and a flat bottom was developed and used on the AB350 wing cover.

#### 6.2.2. Wing Skin and Fuselage Trimming

Wing covers and fuselage sections are relatively large composite structures; thus, relatively large envelop machines with lengths and widths of 40 m and 8 m, respectively, are required. These machines are hybrid, with two z-axis masts: one for an AWJ and the other for a router end effector. The wing or the fuselage section is supported on programmable posts to comply with the shape of the part and hold it with vacuum cups. Figure 48 shows examples of these machines; Figure 49 shows the end effectors which are currently used by Airbus and Boeing and their subcontractors for composite trimming. The waterjet wrists on these machines usually have a focal point just below the tip of the mixing tube with the same standoff distance. Catcher cups are attached to the wrist using C-shaped frames. A catcher cup, in its simplest form, is a carbide tube with an inlet cone and a side port for the spent jet to exit.

The trimming process of these large skins consists of several steps, including part loading, probing, machining, inspection, and then part unloading. The loading process starts by selecting programmable posts from a matrix of these posts, setting the selected posts to the correct height, probing the posts, loading the part, and then applying the vacuum. The probing process starts by probing the global frame, stringers, manhole, and the skin. The machining process starts by trimming the stringers with the correct tool configuration, changing the tool to trim the edges, changing the tool again, cutting the access hole, and finally, using the router for milling and drilling. The inspection and measurement of the parts is conducted, after which the part is lifted. It must be mentioned here that handling the cutouts is a critical process: the potential dropping of parts, causing damage near the end of the cut, must be avoided. Some interim fixturing is performed to ensure the safe transfer of weight for the cutout pieces. Figure 50 shows the sample data for cutting 27.2 mm thick composite, as may be found on some wing skin sections. The data show the taper that needs to be compensated for using the five-axis wrist. A cut cross-section with an 8-micron surface finish is shown.

#### 6.2.3. Robotic Clip Trimming

There are a large number of small composite parts, such as brackets and clips, that need to be trimmed and shape-cut, as mentioned above. For example, Figure 51 shows some composite clips that do not exceed 150 mm in size in any direction that need to be shape-cut out of a plane surface corner. To cut these parts, it is advantageous to manipulate them under a stationary jet to avoid complex robot plumbing with UHP tubes.

A robotic trimming cell has been developed using a Kuka robotic arm to hold and manipulate a part to be trimmed under a stationary jet. Figure 52 shows a sketch of this system. A gripper is used to hold the part. Part and TCP referencing routines have been developed for rapid trimming using offline programming. A small catcher cup has also been integrated into the cell for directing the waste into a collection drum instead of using a catcher tank. However, a catcher tank may also be used to submerge the part while cutting, so as to minimize noise and control the environment [69].

#### 6.2.4. Fan Blade Trimming

The weight of fan blades is a significant factor in propulsion system weights; weight increases the cascade throughout the engine system. The trend toward larger fans has thus driven the need for CFRP-material fan blades, with the added benefit of improved levels of damage and defect tolerance. Accordingly, GE, for example, uses composite fan blades on the GEnx engine, with larger and fewer composite blades. Figure 53 shows some example fan blades.

Recent tests performed to trim fan blades using the AWJ process showed great promise for trimming around the entire blade, with variations in thickness from a few millimeters at the tip to a few tens of millimeters at the root. Figure 54 shows an AWJ-cut surface on the edge of a fan blade. The ability of a multi-axis AWJ using a focal point wrist, as described above, was demonstrated using offline programming (such as Cenit, Stuttgart, Germany) and thickness-based cutting speed variation using advanced AWJ process models. Special attention to fixturing must be paid, especially if the fan blade titanium cap is to be trimmed after mounting on the previously trimmed composite blade (as is the case for the GEnx engine).

### 6.3. Cutting Hard Materials

Cutting hard materials such as ceramics, carbides, and ceramic matrix composites presents machining challenges due to their unusual strength and brittleness [70,71,72,73]. Two important CMCs are SiC/SiC and Al_2_O_3_/Al_2_O_3_, due to their ability to elongate. Conventional tools, such as diamond-coated grinding wheels, drills, mills, and routers, face short life expectancies, as well as time constraints, when used to machine these CMCs. Both factors increase the cost of machining using conventional tools. Additionally, these methods can create heat-affected zones (HAZs) in the part, or impart high tool loads on the workpiece, increasing chipping. The tool load can be significantly reduced through laser-assisted machining, which uses a laser to heat a thin layer of the target material during cutting [74]. AWJ technology has been used to cut a wide range of hard materials, such as those detailed in Table 3 [70].

Using aluminum oxide abrasives, a maximum cutting speed of 275 mm/min was observed in 2 mm thick sapphire sheets [71]. However, it is known that using aluminum oxide abrasives results in rapid nozzle wear [20]. To enhance the AWJ performance, nozzle inline vibration was used [75]. Another approach was to mix garnet and aluminum oxide. In order to mix garnet and aluminum oxide, a dual abrasive feed system was used [76] to blend the garnet and aluminum oxide before entering the cutting head. With 30% aluminum oxide at a total abrasive flow rate of 9.75 g/s (the same as in the case of 100% Al_2_O_3_), the maximum cutting speed dropped to 171 mm/min, which is about 61% of the abovementioned cutting speed with pure Al_2_O_3_. This clearly shows the advantage of improving the cost of cutting. The acceptable cutting speed for an acceptable surface finish and chip size was limited to 100 mm/min. Notably, at 400 MPa, the acceptable cutting speed did not exceed 51 mm/min. A significant drop in cutting speed with reduced pressures is typically observed in carbides and ceramics where the threshold abrasive particle velocity is relatively high. Accordingly, the effective particle velocity for cutting must be sufficiently higher than the threshold velocity to achieve reasonable cutting speeds.

A 5 mm thick sample of SiC/SiC composite was cut at different speeds, up to 228 mm/min at pressures of 380 MPa and 600 MPa, and various abrasive mixes at flow rates of 5.6 to 9.1 g/s. Figure 55 shows the taper of the cut where no surface damage was observed.

Observation of the striations when cutting hard materials with garnet revealed that the pattern is highly regular. Hashish [18] attributed this to the insensitivity of the cutting process to other changes in other factors.

Figure 56 shows a sample taper of the cut data trend as a function of the cutting speed. The taper of the cut decreases as the pressure increases, and also when using harder abrasives such as aluminum oxide.

Using diamond abrasives showed a significant increase in the cutting speeds of hard materials [71]. Table 4 shows the results obtained at a cutting speed of about 5.1 mm/s. All the thicknesses shown were cut through, but the mixing tube wore out rapidly while the tests were being conducted, and the top widths of the cuts gradually increased. In order to cut the above materials economically, including PCD, an improved mixing tube material is needed. GE Super Abrasives [77] patented a CVD process to manufacture diamond tubes for abrasive waterjet cutting. However, no progress was made in this area due to the cost of manufacture and the weak market demand.

The use of diamond powder in cutting applications requires the abrasives to be recycled for cost-effectiveness. Samples of the diamond powder were collected after cutting and did not show fragmentation.

### 6.4. Micro AWJ Cutting

Micro AWJ technology may be defined as the waterjet being 250 microns in diameter or less. This poses some challenges regarding the abrasive feed and ability to cut consistently. The first reported study on “micro” AWJ cutting has been detailed elsewhere [78], where a 177-micron AWJ was used at 690 MPa pressure. In this brief study, 25-micron and 76-micron orifices were tested. Figure 57 shows the nozzle used for testing, while Figure 58 shows a graph of the aluminum cutting data. Increasing the pressure from 50 MPa to 690 MPa increased the cutting rate 10-fold for the case of 25-micron orifice sizes. However, the power of the jet increased by a factor of 20. The use of 25-micron waterjets showed that orifices of this size are not sufficiently reliable, and require special water filtration. For the case of the 76-micron jet, the improvement was also linear. The power of the 76-micron jet was observed to be about 9-fold greater than that for the 25-micron jet. However, the speed improvement was only about 2–3-fold greater. This suggests that smaller diameter waterjets are more efficient if they are sufficiently capable of cutting through at the required edge quality.

Miller [79] introduced the term “improved entrainment abrasive waterjets (IAWs)”, which use steam instead of air to carry the abrasives from a hopper; the steam condenses in the cutting head, eliminating air. Liu [80] presented parts with fine features cut with a micro AWJ tool. A new approach for forming micro AWJs below 250 microns was patented [81,82], entraining abrasives as a paste or as a foam, as shown in Figure 59 (left). The benefit of using a paste is to facilitate the feeding of fine abrasives, and to also enable in situ recycling, as no drying will be needed.

A miniaturized AWJ cutting head was patented [83] using additive manufacturing to obtain curved passages (Figure 59 (right)) to include complex curved passages for improved feed.

To resolve the issue of reduced air flow (and abrasive feed) in micro AWJs, vacuum assistance was utilized [84]. Additionally, an automatic periodic mixing-chamber cleaning [26] was found to be important, as described in the micro-SD cutting section. Annoni et al. [85] used an AWJ cutting head with a 300-micron focusing tube and 200-mesh abrasives. Ojemertz [86] discussed the micro AWJ machining of titanium, nitinol, aluminum, and other materials using AWJs down to 200 microns in diameter. Figure 60 shows the size ranges of micro AWJ tools that have been tested and reported in the literature.

Figure 61 shows an example of a micro AWJ cutting head, developed by Craigen and Hashish [87], that has been used to cut small-sized parts on a wide range of thin (approximately a few millimeters) materials such as metal, glass, PCBs, and carbon fiber composites. This cutting head uses a 250-micron mixing tube with 50 mm length; 75- and 100-micron orifice sizes can be used to entrain 230-mesh abrasives from a mini hopper through a 1 m long hose. The location of the vacuum assistance port was modified to improve the flow field inside the mixing chamber.

An example cutting sample with this cutting head is shown in Figure 62. Intricate shapes with a high degree of flexibility were cut without any distortion as the forces of micro AWJs are relatively small.

Another example of micro AWJs is for sidefire cutting with a 150-micron diameter mixing tube with a 19 mm length, as shown in Figure 63. This design was used to allow access for the nozzle to machine adaptive grooves on the surface of gas turbine blades.

### 6.5. Drilling

In shape cutting, the jet may pierce the material first before starting to cut the required shape. In this case, the hole shape is not important. However, in this section, we discuss drilling, when the shape of the hole is required to meet certain requirements.

Upon the impingement of an AWJ on a surface, it starts eroding it through the repeated impact of the abrasive particles until it eventually pierces through. A visualization study [9] filmed the AWJ drilling process through transparent materials using high-speed photography. It showed the progression of the penetration process and that there is a tip at the front of the hole, not a blunt face. Figure 64 shows traces of the hole progression in Plexiglass and Lexan. The optical imaging of the cross-sections of blind holes drilled in steel showed similar shapes and progression patterns.

When the jet pierces through, the hole shape will change over time. Figure 65 shows the different shapes that may result.

In the subsequent sections, we discuss the drilling (piercing) of different materials for different applications.

#### 6.5.1. Metal

Figure 66 shows blind holes drilled at 345 MPa in 12.5 mm thick aluminum and steel for different drilling times [64]. Three holes were drilled for each selected time, as shown at the top of the pictures. The similarity of the shapes of the holes and random variations in depth were observed for a given drilling time.

Figure 67 shows the breakthrough drilling times for steel and aluminum with a drilling time ratio of about 2 for drilling in steel versus aluminum. This ratio does not change with depth, at least in the range shown in the figure. The diameters of the drilled holes measured about 0.6 mm, yielding an aspect ratio of about 20.

To speed up the drilling process, the abrasive particle kinetic power needs to be increased, while the reverse water flow needs to be reduced to minimize drag. Increasing the particle velocity can only be accomplished by increasing the pressure of the waterjet, while reducing the water flow rate requires reducing the orifice size. Increasing the pressure while maintaining the same power will result in a reduced water flow rate, which will assist under faster drilling speeds.

Tests were conducted by drilling 12.7 mm thick steel and aluminum at elevated jet pressures of up to 690 MPa. The drilling time results are shown in Figure 68. Increasing the pressure from about 300 MPa to 690 MPa linearly reduces the drilling time. Figure 69 shows a sample of holes drilled in a 20 mm thick steel block.

Hole trepanning is another way of speeding up drilling, especially for thick materials. Trepanning was tested for the drilling of small-diameter holes (less than 1 mm in diameter) in vent screws up to 51 mm long. It was suitable for rotating the workpiece used under a radially offset AWJ instead of trepanning (orbiting) the AWJ. The effect of eccentricity on drilling time progression is shown in Figure 70. It is clear that radially offsetting the jet helps increase the drilling rate, as the passage for return flow is now larger. The degree of offset is limited to within the jet diameter so as not to form a core in the middle.

The drilling of thin metals may result in exit burrs, as described in Section 6.1.1, which describes the cutting of thin materials. To reduce the burr, the hydrodynamic force needs to be minimized. For a given jet power, the higher the pressure, the lower the hydrodynamic force. In addition, the use of finer abrasives results in a lower burr height than larger abrasives.

#### 6.5.2. Brittle, Coated, and Laminated Materials

The drilling of brittle, coated, and laminated materials, such as glass, CMCs, TBC, and graphite–titanium laminates, introduces new challenges over drilling metals. These materials may crack, chip, spall, splinter, or delaminate. Accordingly, different strategies are used in drilling these materials.

##### Glass

It has been observed that when piercing glass using the AWJ process, cracking and chipping occur instantly at the top surface. This was attributed to the fact that only a waterjet without abrasives impacts the material when opening the UHP on/off valve and before the abrasives arrive at the cutting head through the suction of the waterjet.

To eliminate this problem, the first solution was to reduce the pressure of the waterjet. However, this reduces the ability of the jet to entrain air to carry the abrasives reliably. Accordingly, a vacuum assistance device [84] was developed to entrain abrasives into the cutting head before the waterjet is started, to ensure that a “pure” waterjet does not strike the material without abrasives. With this technique, upper surface chipping was eliminated. Additionally, it was observed that less cracking and chipping occurs when smaller diameter orifices are used. Reducing the orifice size reduces the amount of water that strikes the surface, which minimizes chipping at the top as well as the hydrodynamic loading down the hole. Figure 71 shows many holes drilled repeatedly in glass using the above methods.

Dynamic pressure control [76], where pressure is gradually increased, was found to be essential when drilling through thick glass. The gradual increase in pressure overcomes the effect of the return flow, which may cause the jet to stagnate and stop drilling, and increases the probability of glass radial cracking by the pressure inside the hole. Trepanning or orbiting the jet while piercing also reduces the effect of the return flow. It was observed that drilling through 300 mm thick glass may take up to 40 min at 30 MPa pressure using a 1 mm diameter orifice, a 4 mm diameter mixing tube of 600 mm in length, using about 30 g/s of 50-mesh garnet abrasives. When the pressure was ramped up from 27.6 MPa to over 69 MPa pressure, the time of drilling decreased to about 12 min. Reducing the orifice size to 0.5 mm and increasing the ramp rate dramatically reduced the drilling time to about 2 min. Figure 72 shows the effect of the orifice size and ramp rate on the drilling time.

The above conditions of using a 0.5 mm orifice size were repeated to drill 57 holes in a 300 mm × 300 mm × 3000 mm block of ULE glass. The results of drilling time repeatability are shown in Figure 73. The standard deviation in the hole drilling time was less than 3%.

##### TBC

Thermal barrier coatings (TBCs) are used in jet engines to enable operation at higher temperatures, and thus, higher efficiencies. For example, combustor liners are made of nickel-based alloys and coated with a yttrium-stabilized zirconia (YSZ) ceramic coating. To further increase the temperature, this TBC is cooled with air film pumped from the outside. The air enters through holes that need to be drilled in the combustor liner segments or other hot section components. Laser drilling causes this layer to burn and crack. Initial attempts to drill these components using the AWJ process yielded some defects, as shown in Figure 74, which could be chipping, cracking, or spalling of the TBC surface upon impact by the AWJ, teardrops, and shadow holes that may result from jet deflection. The jet deflection off the metal interface may cause a gouge, while the hole may look geometrically correct at the top surface.

To resolve these defects, some measures have been identified and successfully applied [88]:Vacuum assistance: The use of vacuum assistance was found to be vital in bringing the abrasives to the mixing chamber before firing the waterjet. This prevents chipping and cracking.Pressure and orifice size: Both the pressure and orifice size need to be minimized, as excess water will result in chipping and spalling. Reduced pressure will reduce the stagnation pressure, which also reduces the probability of chipping.Rotate while piercing: This is important when drilling at very shallow angles, such as below 25 degrees to the surface. In this case, the jet starts at a larger angle, either perpendicular or near perpendicular, and starts piercing while rotating to the final angle. The concept is to engage the jet with the metal substrate so as to prevent its rebound off the interface.Dwell time: The dwell time corrects for the shape of the hole, and will also remove defects at or near the exit side.Standoff distance: Reducing the standoff distance will reduce edge rounding at the entry. However, edge rounding is a desired feature in some cases.Mixing tube length: Increasing the mixing tube length will collimate the jet more and prevents hole enlargement at the entrance.Microphone detection: A microphone may be used to detect the jet breakthrough. This is important to control the hole shape by adding the proper dwell time.

Accurate holes can be drilled when some of these measures are used, as shown in Figure 75.

To establish a production process, Hashish and Whalen [89] conducted a study where TBC plates were drilled with 100 holes each to verify the process capability. To measure the air flow rate through the holes, round coupons were drilled to verify that the air flow rate requirements were met.

An adept five-axis manipulator was used to drill several shrouds to specifications with hundreds of holes. The CNC program backed up the hole locations on the metal side to drill from the TBC side, as shown in Figure 76.

##### Composites and Laminates

Drilling composites [90] and laminates requires special strategies to address the anisotropic nature of the material. Diamond tools with special designs are often used to drill composites such as carbon fiber, CMCs, and MMCs. However, the AWJ is one tool that was demonstrated to successfully drill all classes of composites and laminates when proper parameters and strategies were used. For example, Figure 77 shows a sample 0.75 mm hole drilled in alumina-based CMC material [71] without any adverse effects with the aid of vacuum assistance.

Figure 78 shows AWJ-drilled holes in a sample of a flame holder SiC/SiC material. The holes were drilled at different angles and orientations. The material cross-section shows some voids, but the holes exhibit no evidence of delamination or chipping. Garnet abrasives were used in drilling this hard material.

Figure 79 shows a sample measurement of hole circularity in an Al_2_O_3_/Al_2_O_3_ sample. It has been observed that hole circularity improves with dwell time after piercing to correct for the effect of the return flow and the somewhat anisotropic nature of the material.

An observation of AWJ drilling is upper surface frosting when the standoff distance is increased (Figure 80). This can be eliminated by reducing the standoff distance and using more collimated AWJs with relatively long mixing tubes. Additionally, similar to PCD drills, poor drilling conditions may cause delamination, which is the main mode of failure in this case. This delamination may occur in three locations: the top surface, interior surface, or bottom surface. In all these locations, delamination occurs when the impact of the waterjet exceeds a certain threshold stress while the penetration rate is limited below a certain limit. These threshold values vary at different locations while drilling. For example, when affecting a composite material with a plain waterjet (no abrasives), upper surface delamination may occur due to the relatively low penetration rate of the waterjet, as depicted in Figure 80. Moreover, the impact of the jet at high pressure may cause a shockwave impact that could result in delamination [91].

Failure by delamination, especially for thick composites, has been solved by optimizing the drilling conditions and following certain techniques and procedures. For example, vacuum assistance is used to ensure that abrasives are present in the nozzle when the waterjet is fired. The pressure and size of the waterjet must also be adequate for drilling. Typically, lower pressures and smaller jets are safer to use when drilling, as mentioned above for glass piercing. Accurate control over the dwell time after the jet breaks through the material is needed to control the hole size within certain tolerance limits. Breakthrough acoustic sensors have been used to detect jet penetration for automatic dwell time control. Figure 81 shows drilling through 25 mm thick composites using the above methods.

To maintain consistent drilling conditions, the mixing chamber of the AWJ nozzle should be periodically cleaned with tap water via a flushing port and a check valve. Figure 82 shows a cutting head used for drilling (and cutting).

Nozzle flushing was found to be important when drilling large numbers of holes, as may be needed for nacelles, for example. Flushing is used periodically to remove any abrasive buildup in the mixing chamber. This ensures process reliability. A test was conducted by Good Rich Aerospace into the drilling of carbon fiber composite nacelle parts and their mating aluminum parts. These holes are typically in the range of 1 mm in diameter. The perforations help to attenuate jet engine noise [92] and assist in de-icing. Figure 83 shows the drilled parts.

Prior to drilling the above part, flat samples were drilled to verify the process robustness, hole quality and size repeatability, speed, and any other issues. Figure 84 shows a flat sample drilled with 10,000 holes, 1 mm in diameter, in 1 mm thick material in a matrix of 100 holes by 100 holes. The drilling time was 0.9 s hole to hole. The Cpk of the hole size was over 1.6. Since actual parts may have over 300,000 holes, multiple heads are needed to produce this part in the required time.

#### 6.5.3. Shaped Holes (Shaped Mixing Tubes)

There are two types of shaped holes. The first type is making holes that are not round but with fixed cross-sections. The other type is to shape the hole along its axis, such as making convergent, divergent, or convergent–divergent holes. If the hole size is a few times larger than the diameter of the AWJ tool, then this process becomes a cutting process.

Square-shaped holes are required for the screens of the F-22 aircraft. The size of the hole is in the same range as the size of the mixing tube; thus, a square-shaped mixing tube was tested to make square-shaped holes. This proved to be successful for a certain number of holes, as the shape of the mixing tube gradually changed to a round shape.

Convergent or divergent hole shapes are dependent on the dwell time of the jet. At the time of jet breakthrough, the hole is typically convergent; however, with an additional dwell time, the hole may become divergent. Obtaining a convergent–divergent hole was tested by pivoting the AWJ tool around the throat of the hole. Figure 85 shows shelf holes drilled in a jet engine vane using this pivot approach. The AWJ diameter in this case needs to be at least one-half to one-third the size of the hole throat to allow for better hole shape control.

#### 6.5.4. Gun Drilling (Boring)

The concept of an AWJ deep hole drilling (boring) system has been tested [93], and a prototype [94] was then built and used to investigate the boring of refractory hard-to-machine materials for nuclear applications. In these case studies, 25 mm diameter rods of tungsten needed to be drilled with approximately 19 mm wide holes, 0.5–1 m in depth. The major components of the prototype included a modified conventional lathe for workpiece fixturing and rotation; an ultrasonic measuring system for measuring workpiece wall thickness; an ultrahigh-pressure AWJ boring tool mounted to the lathe traverse; and systems for controlling aspects of the AWJ process, such as abrasive feed and spoils collection.

Various boring string concepts involve jets at different angles and power levels to uniformly “mill” the face of the hole. A drill string with three waterjets firing inside a rectangular mixing tube was found to be the best option. Matching the material removal rate to the boring tool advance rate was difficult even when an ultrasonic feedback sensor was used. To overcome the above issues, a two-step process was developed, where drilling and reaming processes were used alternately. A pilot hole was drilled over a relatively short (200–300 mm) distance, followed by a reaming tool to increase the hole diameter without encountering a spike in the center or the associated gouging and worm holing effects. The material removal patterns were measurable with ultrasonic equipment, and were adequately repeatable. Figure 86 shows a schematic of the pilot hole drilling tool using a radially offset AWJ tool to achieve the trepanning effect.

With this approach, pilot holes were drilled in mild steel to a depth of several feet, and reamed to a uniform 19 mm diameter. A 300 mm long tungsten alloy workpiece was drilled and reamed along its entire length. A drilling rate of 250 mm per hour was achieved in tungsten alloy using the drill/ream approach, with a cost of about USD 50 per foot. Figure 87 shows some example holes.

#### 6.5.5. Drilling Model

Several analyses and modeling studies have been performed on the AWJ drilling process [88,95,96]. Simple momentum balance analysis of the drilling process [64] yielded the following equation for the drilling time:(26)$$t=\frac{\sigma_{f}}{2E_{d}K_{2}}\left[e^{2K_{2}h}-1\right]$$
(27)$$K_{2}=\frac{3C_{d}}{4S_{p}d_{p}}$$

In the above equation, $E_{d}$ is the power density of the AWJ; $d_{p}$ and $S_{p}$ are the diameter and specific gravity of the abrasives, respectively; and $C_{d}$ is the drag coefficient that reduces the abrasive particle velocity and can be determined experimentally.

This equation shows the importance of increasing the power density of the abrasive particles in order to reduce the drilling time. It also shows that the drilling time is exponentially proportional to the depth. As the depth increases, the abrasive kinetic energy needs to be increased; therefore, pressure ramping is important. Increasing the factor *K*_2_ in the above equation will also increase the drilling time; thus, efforts should be made to reduce *K*_2_. This equation clearly suggests reducing the drag coefficient, which can be accomplished by reducing the water flow rate. Moreover, widening the hole by trepanning distributes the return flow over a larger area, reducing the drag. This results in improving the drilling time. Increasing the abrasive specific gravity and size also helps to reduce the drilling time.

### 6.6. Milling

#### 6.6.1. AWJ Milling Process

To control and limit the depth of an AWJ cut, only small amounts of material need to be removed per pass; with many passes, the enquired depth can be reached. This requires either the jet to be weak, or moving at relatively high traverse rates [97,98,99]. The latter approach involves the use of masks composed of more resistant materials because high-speed (~10 m/s) contouring motion is difficult to achieve in practice. Three methods are typically used for controlled depth milling [98]:Liner Milling: In this method, Cartesian motion is used to scan the jet over the masked workpiece.Radial Milling: The jet moves radially over masked workpieces mounted on a rotating platter. The platter itself or a dome may be the workpiece in this case.Cylindrical Milling: Controlled depth milling can be achieved on both the outside and the inside of a drum on which workpieces are mounted.

Figure 88 shows radial and cylindrical milling concepts.

Masks can be cut with an AWJ or laser and mounted on the workpiece. This method has been used to mill isogrid patterns on the inside and outside of cylinders, cones, and domes [100], as will be described subsequently.

In AWJ milling, the depth of cut per pass is a function of the relative velocity between the AWJ and the part surface: this is called the surface speed, and is the amount of overlap of the AWJ per pass across the surface of the part. To achieve constant depth milling in radial milling, a kinematics model [56] was used to relate the surface speed and overlap to the linear velocity of the nozzle and the angular velocity of the turntable. Using this model, the nozzle and turntable motions could be programmed to produce constant-depth milling from the inside radius to the outside radius. Briefly, the rotational speed needs to be increased as the nozzle radial position approaches the center, yielding the nozzle radial velocity.

#### 6.6.2. Milling of Metallic Isogrid Structures

Isogrid structures are lightweight structures with high strength-to-weight ratios. Typical examples of isogrid patterns used in several aircraft and aerospace applications are those in jet engine casings and ICBMs. Solid tool milling may result in distortions and are limited to certain skin thicknesses and rib widths. Chemical milling is environmentally unacceptable, requires large baths, and is slow. AWJ milling offers solutions to these problems.

Under US Airforce Funding, a study was undertaken to develop the AWJ isogrid milling process and apply it to some model examples to determine the volume removal rates, the accuracy of milling to a specified depth, to establish milling data trends, and to address the economics of the AWJ milling.

Common hydraulic and abrasive parameters affect the milling results, as do some additional kinematic parameters including the speed, standoff distance, jet angle, and the degree of overlap. The simple test setup shown in Figure 89 was used to generate data on the milling results using cylindrical milling. A 1.22 m diameter cylinder was milled on the inside wall as a representation of a missile skin. The mask was cut out of steel and placed on the inside on an aluminum cylinder. This was performed twice to mill a total length of 400 mm. The milling time was about 6 h using a 20 kW AWJ at 310 MPa pressure. The milling depth variation was about 0.025 mm. Thin skins down to 2 mm were milled with about a 10-micron surface finish using 100-mesh garnet abrasives. To improve the surface finish, increasingly fine abrasives may be used successively.

#### 6.6.3. Milling of Gamma Ti-Al Heat Tiles

Aluminide-based alloys such as gamma titanium aluminide (gTi/Al) offer superior high-temperature performance with low weight and non-burn characteristics. For example, the milling of heat shield tiles used in the exhaust section of an advanced aerospace propulsion system was required. This material has limited heat treatability and low ductility at room temperature, which makes it difficult to machine and process by conventional means. One challenge was to pocket-mill relatively thin sheets of gamma Ti/Al sheets to thin skins. It was demonstrated that milling could be accomplished to 0.025 mm accuracy. The use of angled jets at about 15 degrees to the vertical and indexing of the jet was found to be essential in obtaining high-quality milling results without inducing undercutting problems near rib roots. The milling process relieves stresses, and accordingly parts need to be annealed before milling; otherwise, deformation will occur. The milling to thin skins was successfully demonstrated on 150 mm × 300 mm parts. Additionally, the milling process of dual rib height was developed and demonstrated [56]. This required the use of a dual mask approach. The basic tile milling geometrical requirements were a cell size of 16.5 mm × 16.5 mm, a skin thickness of about 0.5 mm, a rib width of 1 mm, and two rib heights of 3.3 and 0.86 mm. Figure 90 exemplifies some milled tiles.

#### 6.6.4. Milling of Glass Telescope Face Sheets

The AWJ-controlled depth milling of glass face sheets has been developed as the currently accepted process used for telescopes. Miles [101] presented some studies that were conducted with the author on milling different shapes and sizes of glass. Several types of glass have been milled, such as ULE^®^, Zerrodue^®^, fused silica, and fused quartz. Face sheets with 12.5 mm thickness and as large as 2 m in diameter have been milled to about 9.6 mm depth. The main objective of milling is to reduce the weight of the telescope. Milling the face sheet reduces its weight by about 60%. Figure 91 shows examples of AWJ-milled face sheets 2 m (left) and 1.2 m diameter (right). The mask was placed over the glass, with a thin layer of rubber or foam placed between the mask and the glass. Sometimes, aluminum tape is used instead to cover the surface of the glass to be milled. This prevents direct contact between the steel mask and the glass surface. Radial milling is used for milling these parts at about 350 MPa pressure using 120-mesh abrasives. The milling depth variation is within 0.1 mm. AWJ clock tilting of about 10 degrees is used to minimize taper and undercutting. Sandblasting with 220 mesh is sometimes used to further improve the surface finish, which is about 2 microns after milling. Routine measurement of the depth of milling progression using a Renishaw probe showed that the milling process is linear. To reduce the milling time of about 100 h, more than one cutting head may be used; in this case, a robotic system was preferable.

The shaping of glass for conformal optics by milling has also been demonstrated [102], where a block of glass can be machined and milled to convert it into a thin-walled surface circular in shape.

#### 6.6.5. Deep Milling

Increasing the depth of milling beyond about 20 mm will show the effect of wall taper more clearly, as the bottom surface area will obviously be smaller in size than the top surface area. To correct for these taper angles, jet angulation is required. The jet angles will continue to increase, starting from an initial value of 10 degrees to about 20 or 25 degrees as the depth of milling increases. This, however, may introduce undercutting or web thinning (for isogrid shapes) effects at the upper areas of the cavity walls. Accordingly, this should be accounted for by increasing the web thickness over the design value. The mask for deep milling should also be thick or made out of a harder material. For example, if a steel mask is to be used, then its thickness needs to be about one-half of the thickness of titanium, or one-third of the thickness of aluminum. Using a tungsten carbide mask will only be one-fifth of the required depth of milling, and could be used at least four–five times.

Figure 92 shows examples of deep pockets in titanium, aluminum, glass, and steel. The aspect ratio of depth to width is crucial, as narrow pockets may never reach the required depth due to jet deflection. As an example of AWJ milling performance, the titanium part shown in Figure 92 was milled in about 60 h. However, in the same time, the rotary milling platter could be populated with more parts to be milled simultaneously. A 2 m platter can be used to mill 60 parts at the same time. Typical parameters are 400 MPa pressure, a 0.38 mm orifice, and a 50-mesh garnet at about 1.5 lb./min. To reduce the milling time, multiple cutting heads can be used. This means that, if multiple nozzles are used on the same platter, further reductions in milling time can be achieved.

#### 6.6.6. Composite Repair

The use of the AWJ process for composite repair may involve removing the damaged material, preparing the surface, and rebuilding the damaged area [70,103,104]. The flexibility of AWJ cutting and milling in a wide range of materials and shapes makes it suitable for the repair of composite structures. Figure 93 shows typical joint designs used in adhesively bonded composite repair, which has benefited from cutting and milling with AWJ.

For repairs of large composite structures, scarf angles corresponding to 1:30 to 1:40 rise/run ratios are desired (1.43° < θ < 1.91°). Currently, these types of low-angle surfaces are created by hand with a small abrasive drum or disk tools. The process is time-consuming and imprecise. Initial tests addressed one-dimensional milling to study and identify the parameters for scarf angle control. Two-dimensional scarf milling was then tested to produce scarfed areas. The material removal rate and surface roughness obtained with 220-mesh abrasives in graphite epoxy were 60 μm/pass and 38–50 μm Ra, respectively. An AWJ system has been developed [105] where the robotic scanning of the damaged area is first performed, and then the AWJ system is applied to perform cutting and milling. Figure 94 shows experimental milled scarf joints in graphite epoxy. Similar results were obtained in astroquartz composites.

Moderate pressures (~100 MPa) are used for the removal of carbon fiber layers. The typical material removal rate for composite repair is in the range of 25 to 100 microns per pass when using 220-mesh garnet with 1.5 g/s abrasive flow rate.

### 6.7. Grooving

Grooving is a milling process typically involving relatively thin widths: this is the jet width. There are many applications where grooving is needed, but most common are cooling and heat dissipation applications, as may be encountered for some jet engine parts. The grove shape may require special strategies when applying the AWJ. For example, there may be limitations on the taper or the shape of the groove. In this section, we limit the discussion of grooving without using masks. In some cases, masks are used to make multiple grooves simultaneously. The examples of the AWJ grooves are shown in Figure 95. Pulp and paper stainless steel refiner machine plates require approximately 2 mm wide grooves of about 5 mm in depth. Three AWJ nozzles were used simultaneously to mill the grooves with the two outside AWJ nozzles tilted to the outside at 10 degrees to minimize the wall taper. No tight tolerances were required on the bottom depth uniformity, which allowed a faster AWJ milling process with economic advantages over other methods.

Thin-width grooves (~0.5 mm) are required in many applications, such as jet engine parts for cooling. Accordingly, mixing tubes of about 0.5 mm in diameter or less need to be used. Groove depth requirements vary between 0.5 mm and 2 mm. It was found that lower-power jets are preferred for use with moderate traverse rate levels without overshooting or causing vibrations. Pressures in the range of 250–300 MPa with 220-mesh abrasives produced acceptable grooves in metals, but depth uniformity was not satisfactory because some gouges and irregularities were observed, especially at lower traverse speeds.

Shape grooves were required for the conformal cooling of jet engine blades with a minimal width of less than 0.25 mm. After TBC spraying, the grooves basically become tubes with cooling fluid flowing through them. A micro AWJ cutting head was used to mill the bulb-shaped grooves with several passes at different angles to shape the bottom of the cavity, which also needed more passes in the center than on the sides. Figure 63 shows this cutting head using a 0.15 mm mixing tube to make a groove with an opening of 0.25 mm.

V-shaped and U-shaped grooves are required on some heat sink parts and were explored using AWJs with compatible sizes with the required groove sizes. While affecting the AWJ structure may benefit some applications, such as using shaped mixing tubes, it was found that the kinematics of the manipulation strategy are more critical for shaped grooves. An important advantage of AWJ grooving is the ability to obtain a thin skin, as shown in Figure 95.

#### Milling Model

The milling process generally involves traversing the jet many times over the area to be milled. During a single “sweep” over an area, due to the width of the swath of the jet and the overlap at the edge of each swath, the jet may cut over some zones more than once. Due to the high traverse rates used in milling, no cutting wear mode will be encountered, and the depth of milling per pass will only belong to the cutting wear mode of material removal. Assuming that the standoff distance does not affect the depth of milling per pass, the depth of milling per sweep, i.e., after $N_{p}$ passes, is expressed as follows:(28)$$h_{av}=\overset{N_{p}}{\underset{1}{\sum }}\frac{2m_{a}V_{a}^{2}}{\pi \;d_{m}\;u\;\sigma_{f}}$$

The lateral increment, ϵ, is related to the width of the jet. The profile of the depth is not uniform; therefore, jet overlapping is needed. It is assumed that the kerf width equals the mixing tube diameter, $d_{m}$, and thus the lateral feed increment, ϵ, will be as follows:(29)$$\epsilon =k\;d_{m}$$
where k is a fraction that was found to be best at about 0.7 for the uniform milling of many materials. The speed, u, in the above equation is the relative traverse speed between the workpiece and the AWJ; this is what affects the milling depth. In rotary milling, this speed is a function of the rotational speed, N. In cylindrical milling, the speed u = π DN, where D is the drum diameter. In radial milling, where the milling location changes radially, the velocity, ϵ, and the lateral increment, ϵ, must be kept constant, which requires variations in both the rotational speed and the radial traverse speed, $r^{.}$, as a function of the radial position. The expression [47] for this radial traverse speed is given as follows:(30)$$r^{.}=\sqrt{\frac{u^{2}\;\epsilon^{2}}{\epsilon^{2}+r^{2}}}$$

The rational speed as a function of the radial position is N = u/(2 π r).

### 6.8. Turning

Hashish [105] first introduced this process for turning hard-to-machine materials such as metal matrix composites. For example, magnesium boron carbide parts were turned with the AWJ process, showing that the machining rates are similar to those of magnesium or aluminum. With a solid tool, only diamond tools may be used, as the filler particles become the most influential, not the matrix. This is a significant difference between AWJ technology and solid tools, where AWJ sensitivity to the filler material is much lower than solid tools. Another advantage of AWJ turning is the ability to turn rods of high aspect ratios, as the AWJ forces are relatively low.

#### 6.8.1. Turning Operations

Turning is a process where the workpiece rotates while the tool reduces its diameter from an initial diameter, d1, to a final diameter, d2. However, with the AWJ process, this definition is broadened to include all AWJ machining operations on a rotating part. This includes diameter reduction or shaping, slicing, and spiral cutting. In diameter reduction, the jet will travel in the axial direction at a certain cut value depth of d. The resulting diameter should then be d1 − d2. However, the resulting diameter may be larger than this value based on the cutting speed and the overall effectiveness of the AWJ technology. In slicing, the AWJ may travel radially from the outside to inside until a slice is cut, or remain stationary at the top center of the rotating part until complete separation occurs. To machine a thread, which is a diameter reduction operation, the jet travels with varying depths of cut. To machine a spiral, the jet travels axially on the rotating part to either mill a spiral grove or completely separate two spiral parts as the depth of cut in this case reaches the center of the workpiece. Figure 96 schematically shows the different turning operations: examples of these operations are shown in Figure 97.

It was observed experimentally and from modeling studies [97,98,99,100,101,102,103,104,105,106,107,108,109] that the most critical factor affecting the surface finish is the traverse speed, which must be selected in order to prevent waviness or striations of the surface due to the macromechanics of the turning process. The effect of the abrasive grit size is then left on the surface as roughness that can be improved upon by reducing the grit size in subsequent finishing passes. Figure 98 shows an example of turning data [110].

#### 6.8.2. Turning Models

The problem in turning is to determine the final diameter (*d_f_*) as a function of AWJ and turning parameters. Hashish and Ansari [111] developed a slotting (slicing) model for a stationary jet located at certain radial position from the center of the rotating workpiece. The model in its simplest form is when this offset is zero, i.e., the AWJ is at the central location. In this case, the impact angle is 90 degrees. The relationship between the diameter, *d*, and time, *t*, is as follows:(31)$$d=\sqrt{d_{1}^{2}-\frac{m_{a}^{.}\;V_{a}^{2}\;}{\pi \;d_{m}\;\sigma_{f}}t\;}$$

Ansari [112] developed an AWJ model for the turning process after conducting visualization studies [113] of the cutting process. It was observed that turning is comparable to cutting on a curved interface with zones similar to those observed in linear cutting with the jet–material interface along a complex line on a curved surface. Figure 99 shows the terminology used for modeling.

The relationship between the penetration depth and workpiece radius can be expressed [39] as follows:(32)$$\frac{dr}{dh}=\frac{h-{\left(2r_{1}\delta -\delta^{2}\right)}^{1/2}}{{\left[r_{i}^{2}-2h{\left(2r_{1}\delta -\delta^{2}\right)}^{1/2}=h^{2}\right]}^{1/2}}$$

The depth, *h*, has two contributions based on the turning conditions. The first is *h_c_*, which occurs due to shallow angles of impact, and *h_d_*, which occurs due to impact at large angles similar to linear cutting. Both *h_c_* and *h_d_* can be determined using Finnie’s and Bitter’s erosion models based on the abrasive kinetic power, travel speed, and material properties, when these depths are estimated; the final diameter can be calculated as follows:(33)$$r={\left\{{\left(r_{1}-\delta \right)}^{2}+{\left[{\left(2r_{1}\delta -\delta^{2}\right)}^{1/2}-h\right]}^{2}\right\}}^{1/2}$$

#### 6.8.3. Segmental Turning

Instead of a turning operation and converting the material between the initial and final diameters to “chips”, chunks of material can be initially cut off to approach the required surface of revolution. The final step to perform is the turning process to a final diameter. Figure 100 illustrates this process when a required smaller diameter part is needed, or a larger bar is required. The steps show three cutting operations and a final turning operation, resulting in saving larger chunks of material. The AWJ turning process need not start from a surface revolution, and with the proper selection of the turning parameters, an irregular non-surface of revolution can be converted into a surface of revolution in one or a few passes. This has a significant effect on the time of machining in addition to saving materials. Hashish [66] explored this approach on turning materials such as metals and composites.

### 6.9. Near-Net Shaping

To demonstrate the AWJ shaping capability in the mid-1980s soon after introducing the AWJ technology, a chess knight was selected for testing. A five-axis Adept robot was used to manipulate the jet, while a sixth axis rotary indexer, controlled by the same robot controller, was used to hold and manipulate the workpiece. Figure 101 shows the steps of shaping the chess piece using cutting, turning, grooving, and drilling operations in a single setup [114].

The roughing of gamma titanium aluminide from cast blocks 52 × 52 × 237 mm in size into near-net-shaped blades has also been introduced into production, with significant savings of material loss (~75%), as two blades can be produced from one ingot, as shown in Figure 102. A robotic system was used for flexibility, and was volumetrically compensated in order to meet the ±0.5 mm tolerance requirement. A 0.33 mm orifice in a 1 mm diameter mixing tube 150 mm in length with 120-mesh abrasives at about 10 g/s was capable of roughing the part in about 15 min, producing several surfaces to be finished mechanically.

The capability of abrasive waterjets for cutting relatively thick sections of steel was demonstrated for the rapid fabrication of automotive tooling molds. This process involves the assembly of an array of uniquely profiled sections of steel to form a mold core or cavity. In addition, this method enables the incorporation of conformal cooling (or heating) passages. The profiling and beveling of each steel section is accomplished with an abrasive waterjet. Figure 103 shows cuts in 100 mm thick aluminum to form a 3D laminated part to illustrate the concept.

### 6.10. Surface Modification

Several waterjet surface modification processes have been investigated and applied commercially. Among these applications are stent polishing, automotive cylinder bore texturing, and peening.

For stent polishing, two processes have been developed [115]. The first is polishing the outside diameter. In this case, the stent was mounted on a steel mandrel, which was rotated in a lathe and subjected to relatively low-pressure (~200 MPa) pure water or abrasive waterjets to clean off the laser dross and any other attached material. The use of 220-mesh garnet abrasives was found to provide the best results without cutting into the stent. The second process was to fire an AWJ inside of a rotating stent (~100 rpm) to uniformly clean its interior walls. Figure 104 shows this process. A special machine was built to allow an accurate radial offset of the AWJ, so as to perform a more effective and rapid polishing operation. Relatively small-diameter waterjets (~0.075 mm) were used in this operation. Longer stents may be reversed to polishing from the other side. A 100 mm long stent required about 30 s for internal polishing and about 15 s for external polishing. Therefore, the polishing of stents was accomplished at a rate of about one stent per minute, which was acceptable.

Surface texturing [116,117] is another surface modification process that can either be accomplished with a pure waterjet or an abrasive waterjet. For aluminum automotive cylinders, pure fan-type waterjets are used to texture the internal surface prior to the thermal spray-coating process. Waterjets produce about 8–12-micron Ra surface finishes, which result in a bond strength of approximately 42 MPa, roughly twice that obtained with grit blasting. The basic operation for waterjet cylinder bore surface preparation involves the use of a rotary tool with a single or multiple side-firing jets. This tool is fed axially into the cylinder bore to roughen its surface. The typical parameters are a pressure of 300 MPa with a 0.4 mm diamond fan orifice, at a 18 mm standoff distance, and under a 3 mm/s feed rate. The tool rotational speed is about 600 rpm. It takes about 1 min to prepare the engine block with one tool in every cylinder. Figure 105 shows a typical waterjet-textured surface.

Peening is used in many industries, such as the aircraft and automotive industries, in order to increase the fatigue life of critical components. In waterjet peening, the impact of waterjets on surfaces results in creating residual stresses that are beneficial to enhancing fatigue life [118,119,120]. Figure 106 shows a typical peening test for a round test specimen. The S–N curves suggest that waterjet peening improved the fatigue limit by about 15%, with the stress amplitude at 108 fatigue life cycles increasing from 200 MPa for test specimens with as-machined surfaces to 230 MPa for those with the waterjet-peened surfaces. However, at high stress amplitudes (for example, 350 to 430 MPa), there is little difference in cycles to failure for the unpeened and peened test specimens. At stress amplitudes of less than 350 MPa, the peened test specimens show increased fatigue lives.

Fatigue crack extension data showed that crack extension in the unpeened test specimen occurred from crack initiation at 4600 cycles to final fracture at 12,600 cycles. For the waterjet-peened test specimen, crack extension occurred from crack initiation at 9650 cycles to final fracture at 18,300 cycles. The general trend observed from these data was that waterjet peening tends to delay crack initiation by a factor of two. Moreover, the fatigue cracks growth life increased by 44%.

Peening with AWJ has been investigated [33] for application in metal orthopedic devices. In the study, the influences of AWJ peening on the compressive residual stress, surface texture, and fatigue strength of stainless steel (AISI 304) and titanium (Ti_6_Al_4_V) were studied. The residual stress of the AISI 304 samples ranged from 165 to over 460 MPa. Using the optimum treatment parameters for maximizing the residual stress, the endurance strength of Ti_6_Al_4_V was increased by 25% to 845 MPa. The average surface roughness (Ra) resulting from AWJ peening of the AISI 304 ranged from 5.08 mm to 13.64 mm. The surface texture resulting from AWJ peening was dependent on the treatment conditions, and the Ra increased with the increasing jet pressure and particle size.

### 6.11. 690 MPa Pure Waterjet Cutting

Cutting with pure waterjets offers the advantage of no abrasives being used. Accordingly, it is advantageous to use pure waterjets whenever possible. One of the motivations to increase waterjet pressure is to extend the cutting applications of pure waterjets. It is expected, however, that the cutting rates will be lower than those of AWJs, although other factory or economical requirements may be better met.

Hashish et al. [78] explored the use of 690 MPa pure waterjets for thin sheet metal cutting. Waterjets at 619 MPa were demonstrated to cut a wide range of thin metallic sheets (1.6 mm) at rates up to 4.23 mm/s. Two zones of cutting were identified with distinguished surface morphologies, but the surfaces were rougher than those obtained with the AWJ technique and showed more burrs. It was also observed that the cutting speed increased as the standoff distance increased. However, the cuts displayed rounded edges at the top surface of the material. Most importantly, cutting at 690 MPa produced no delamination in the fiberglass composite materials (as compared to cutting at 414 MPa), and a relatively high cutting speed could be maintained.

Marble tiles about 10 mm thick were cut with 600 MPa pressure waterjets at different speeds. The results were compared to cuts performed at 400 MPa pressure. The results showed that edges without striations could be obtained at reasonable cutting speeds at 600 MPa. This may be beneficial in some tile cutting situations when thin kerfs or when intricate internal shapes are needed. Figure 107 shows images of bottom cuts in a sample tile.

The cutting of PCBs, which are made of fiberglass, and carbon composites have benefited from cutting at elevated pressures especially when the cutting paths encounter copper materials. Figure 107 shows a cut at 759 MPa pressure using a 0.07 mm orifice. The standoff distance of about 20 mm was vitally important in cutting through the material structure without delamination. The cutting speed was about 3 mm/s.

The cutting of relatively thin thermoplastic automotive composites was tested at pressures over 400 MPa to determine whether delamination would occur at these pressures or not. Figure 108 shows the results of the cutting speeds at which the cut edges were deemed acceptable by the customer. The results show that significant speed improvements are obtained by increasing the pressure.

### 6.12. Hybrid Machining

There are two types of hybrid machining centers based on the use of the different machining processes. In the first type, two or more machining processes are used to perform different functions on the same part. For example, an AWJ tool is used to cut, while a mechanical drill is used to drill. In the second type, two or more machining processes are used to perform machining functions on the same cut. For example, an AWJ tool is used to cut, and a mechanical tool is used to surface-finish the cut. In both cases, the different technologies are integrated on the same machining center; thus, the term “hybrid machine” is used.

The use of AWJ technology with other tools to complete a machining process has been implemented in practice in several application practices. The most common application is using both an AWJ and a high-speed spindle router for the machining of aircraft composites. This hybrid machining center is used to trim carbon fiber composite structures with an AWJ tool and route, drill, and countersink holes using special spindles and mechanical tools. Figure 109 shows a typical dual-mast hybrid waterjet–router system used for such parts as aircraft wings, tail fins, and some fuselage sections. Some machines can be as large as an aircraft wing, while others may be specific to a smaller class of parts.

Today, almost all aircraft composites are trimmed on similar composite machining centers, where the AWJ tool is used for trimming finished edges, while high-speed spindles are used for routing and countersinking fastener holes.

An example of a hybrid machine where two different technologies are working on the same cut is the AWJ-EDM. In this case, the AWJ tool cuts a surface with sufficiently close tolerances such that the wire EDM finishes the cut by not fully engaging with the material, but by partially engaging to remove the roughness of the surface. This speeds up the EDM process by a factor of two or three. In addition, not all surfaces on a part need to be finished by EDM. Figure 110 shows a hybrid AWJ-EDM machine that was codeveloped by Flow International and Sodic. Figure 110 (upper left) shows an aluminum part that was first cut with an AWJ tool and then finished with wire EDM. The total cutting length on this part is 736 mm, which was cut in 20 min by the AWJ tool, producing a 2.3-micron surface finish. The EDM finish pass took 50 min. This is about a 50% to 65% time saving over machining with only EDM.

This machine, however, encountered significant challenges in keeping fine abrasive particles from the EDM side; eventually, the two processes were deemed incompatible to be on the same machine. However, the use of two separate machines to benefit from the time saving is being applied in industry today for a wide range of materials, such as graphite and D2 steel.

Other hybrid machine examples are found in the stone and tile industry, where waterjets are used to cut the interior shapes and saws are used for external straight-line cutting. Other tools are used for beveling and bull nosing. Another example is the combination of AWJ and plasma on one machine. Plasma is used when accuracy or heat-affected zones are not critical, whereas waterjets are used to produce higher quality edges or when the material cannot be cut thermally. The two processes can be used on the same part to maximize productivity using a special software that selects the waterjet or the plasma based on the edge specifications.

### 6.13. Cryogenic Jets

The use of cryogenic and ammonia jets has been investigated for several applications, such as cutting in nuclear environments or for demilitarization. Cryogenic liquids such as nitrogen, CO_2_, and argon have been tested to form pure liquid and abrasive–liquid jets.

To control and improve the performance of cryogenic jet tools, Dunsky and Hashish [121] indicated the need for the accurate thermodynamic control of the cryogen in upstream conditions. Figure 111 shows the effect of upstream cooling on a liquid nitrogen jet. Additionally, although not a cryogenic liquid, liquefied CO_2_ jets were generated at pressures of up to 345 MPa, in spite of the fact that CO_2_ does not exist as a liquid under normal atmospheric conditions. The transient thermodynamic conditions at speeds of over 600 m/s need to be further investigated. Cryogenic jet tools extend the range of waterjet tools in many applications.

As with waterjets, the addition of abrasives to cryogenic jets enhances their performance. The entrainment of abrasives in a cryogenic jet, however, is more complex due to possible abrasive feed line freezing and plugging. The tests conducted with entrained abrasives in liquid nitrogen jets showed that the ACJ cut similarly to the AWJ [122]. Figure 112 shows a steel plate cut with a 117 MPa (17-ksi) ACJ. The use of abrasives such as pelletized CO_2_ in liquid nitrogen jets was tested [123], yielding marginal results. The success of this approach has resulted in a zero-added-waste cutting and cleaning process.

The use of ammonia as a liquid (not cryogenic) was developed [48] for the demilitarization of chemical rockets. Ammonia was found to be useful for neutralizing the rocket chemicals using special procedures, resulting in harmless salt waste. The abrasive–ammonia jet perforates and cuts the rocket, while a pure ammonia rotary tool is used to wash out the propellants from inside the rocket. This process was implemented in a military chemical facility to demilitarize rockets safely.

## 7. New Trends

AWJ cutting technology has been a pivotal tool in various industries, as presented in this study, offering precise and versatile cutting capabilities. Recent advances in technology, particularly machine learning (ML), Internet of Things (IoT), and artificial intelligence (AI), have the potential to further revolutionize the way we utilize AWJ machining. These technologies can significantly enhance the efficiency, accuracy, and sustainability of AWJ machining processes, leading to improved outcomes for industries worldwide. This section presents some specific emerging areas.

### 7.1. Predictive Maintenance (IoT and AI)

IoT-enabled sensors and AI algorithms can be integrated into waterjet cutting machine components such as pumps (which is currently in practice) to continuously monitor the health and performance of critical components. These sensors can gather real-time data on factors such as temperature, pressure, vibrations, and wear and tear. Machine learning algorithms can analyze these data to predict when maintenance is needed, allowing for proactive maintenance scheduling. This predictive maintenance approach minimizes downtime and reduces operational costs by ensuring that the machine is always in an optimal condition.

### 7.2. Precision and Quality Improvements (ML and AI)

Machine learning algorithms can be trained using historical cutting data to optimize cutting parameters, including the speed, pressure, and nozzle distance. These algorithms can learn patterns and correlations between the cutting parameters and the resulting cut quality. By constantly analyzing and adjusting these parameters, machine learning can help achieve higher precision, smoother cuts, and overall improved product quality.

### 7.3. Real-Time Processing Optimization

AI algorithms will process real-time data from the AWJ machining processes. This requires the real-time monitoring of the cutting results, adjusting parameters dynamically to optimize cutting results. This includes making rapid modifications based on variations in parameters, material properties, thickness, and other conditions such as the warping of thin sheet plates while cutting [124]. AWJ software [125] has been used to adjust jet angles to control the taper and trailback. AI can adapt the cutting path sequence based on cutting observations to ensure the best cut quality and reduce scrap material, thus enhancing operational efficiency and minimizing waste. Another example is using acoustic signals to detect unexpected changes and adjust for them. It was found that breakthrough acoustic signals are useful when piercing small-diameter holes, as described above, but no work has been carried out on acoustic monitoring during piercing and adjusting parameters before undesirable results occur.

### 7.4. Remote Monitoring and Control (IoT and AI)

IoT-enabled waterjet cutting machines can be remotely monitored and controlled through a centralized system. Operators can access real-time data on machine performance, production rates, and other critical metrics from anywhere. AI algorithms can provide insights and recommendations for adjustments or process improvements, ensuring optimal performance and minimizing downtime.

### 7.5. Waste Reduction and Environment Control

By analyzing historical cutting data and trends, AI and machine learning can optimize material usage from the inventory, reducing waste and improving resource efficiency. The recycling of water and abrasives is critical in minimizing waste, but smart decisions on when to recycle or not will be based on statistics and experience that machine learning an AI will be of great benefit. These technologies can suggest optimal cutting patterns that minimize abrasive waste and scrap material, saving significant costs for businesses. Additionally, AI can provide insights into recycling opportunities for waste materials, including abrasives, mixing tubes, and workpieces, which will promote sustainability and environmental responsibility.

## 8. Final Remarks

Abrasive waterjets were first introduced a little over 40 years ago; since then, they have been used in over 100 countries and 50 industries. The rapid growth and expansion of AWJ technology are due to many of its characteristics. It is capable of cutting almost any material without adverse effects; thus, it has become a standard tool in many industrial fields. It has also revived some old industries, such as stone and tile workshops. Its advent was timely with the introduction of carbon fiber composites for aircraft structures, and it became an essential tool in every aerospace factory. Many entrepreneurs found this to be an ideal tool for starting a business due to its versatility, relatively low cost, and intuitiveness of operation. The introduction of personal computers was quickly adapted by developing predictive software, then becoming part of the machine control architecture. A trend over the years has been to improve the accuracy of the cut parts, and thus, improved longer-life mixing tubes were developed as well as improved software for shape cutting, which also included active and automatic taper control. The operating pressures of waterjets have also increased, from 200 MPa to 600 MPa, over this period, while improving pump reliability. IoT technology is currently used for pump maintenance, and will eventually be implemented for process monitoring and control. The extension of the capability of the AWJ process for micromachining has already started, with commercial hardware capable of 200-micron kerf widths, leading to sub-100-micron kerf widths. The advances in materials will continue to benefit AWJ tools with longer-life components such as mixing tubes, pump components, valves, and abrasives. Additive manufacturing will significantly enhance the functionality of the many parts in an AWJ system, such as cutting heads, pulp valves, and abrasive fluidic devices. Environment control will be a focus of the future, towards overall greener operations. The focus of AWJ 5.0 is on enhancing the interaction between humans and machines to achieve better results. This will be through the integration of artificial intelligence, machine learning, and the Internet of Things (IoT), which are core to Industry 4.0 and 5.0.

## Figures and Tables

**Figure 2 materials-17-03273-f002:**
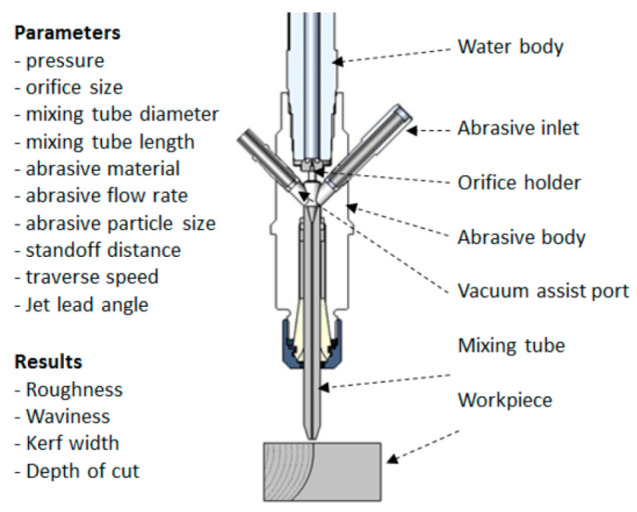
AWJ parameters.

**Figure 3 materials-17-03273-f003:**
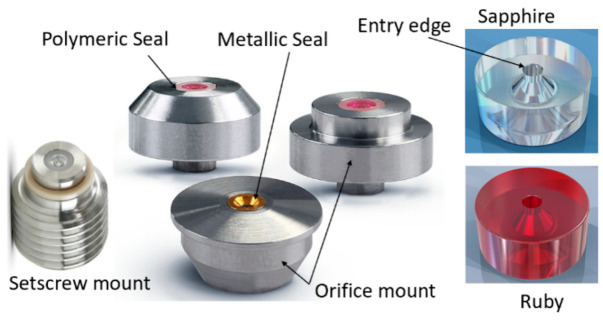
Waterjet orifices.

**Figure 4 materials-17-03273-f004:**
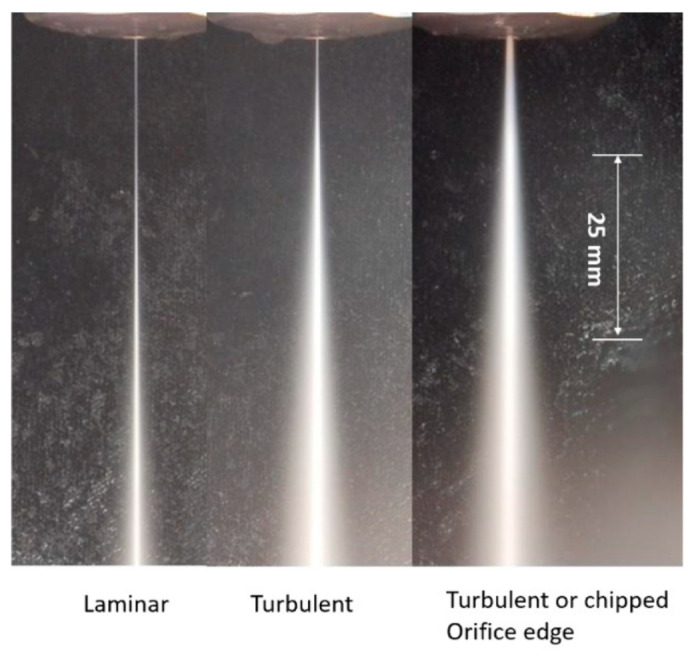
Waterjets with different upstream conditions at 350 MPa pressure.

**Figure 5 materials-17-03273-f005:**
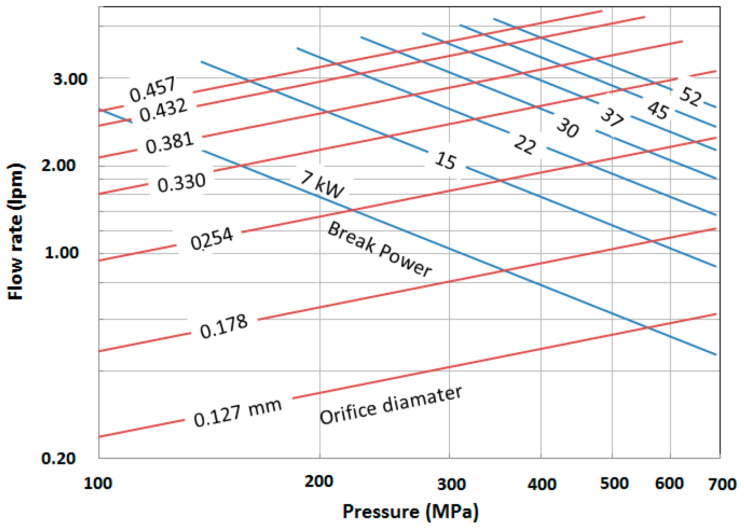
Waterjet flow, power, and orifice size relationships.

**Figure 6 materials-17-03273-f006:**
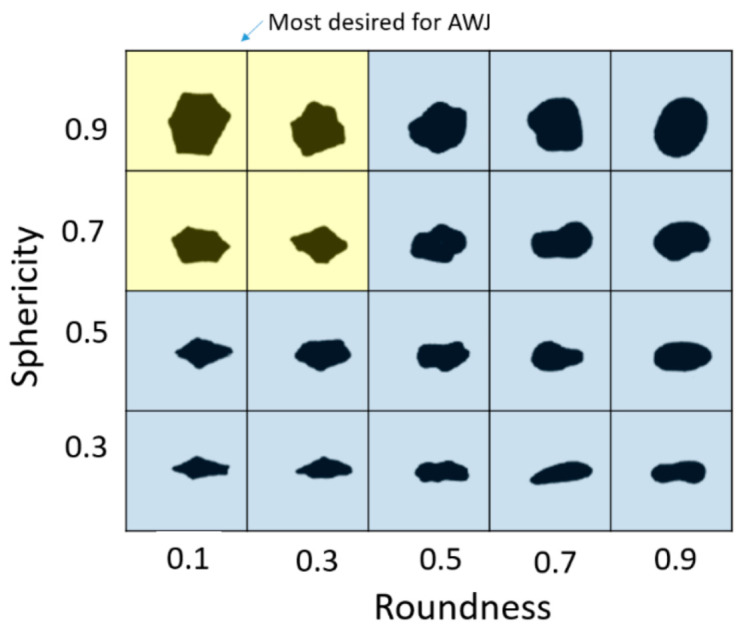
Abrasive particle shape indices and preferred zone (upper left corner).

**Figure 7 materials-17-03273-f007:**
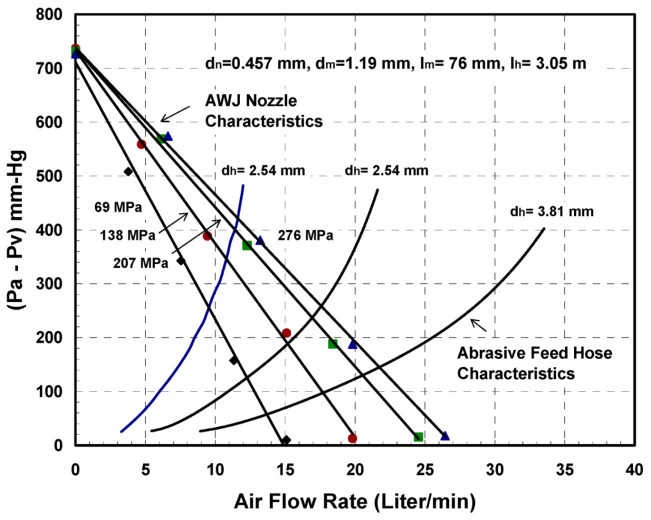
AWJ nozzle and suction hose characteristics [24].

**Figure 8 materials-17-03273-f008:**
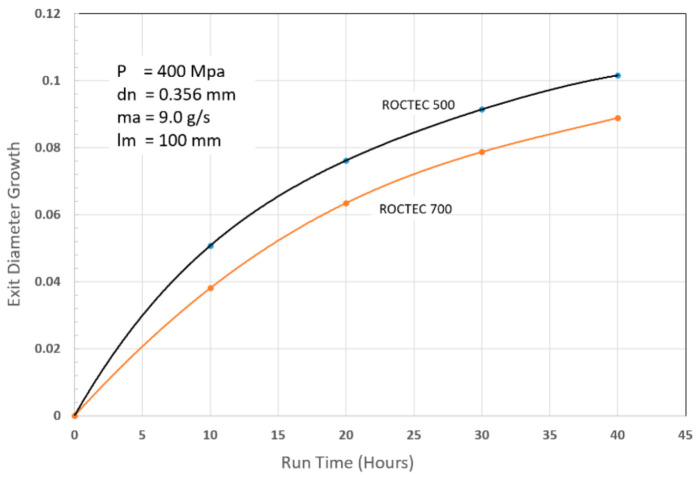
Example of mixing tube wear.

**Figure 9 materials-17-03273-f009:**
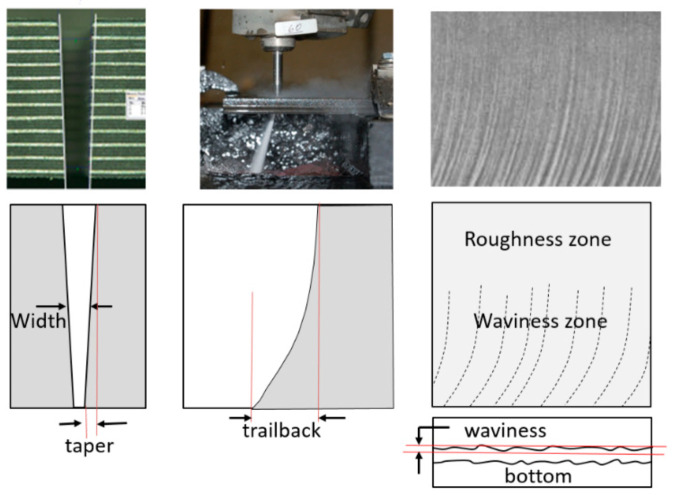
Geometrical attributes of waterjet cutting.

**Figure 10 materials-17-03273-f010:**
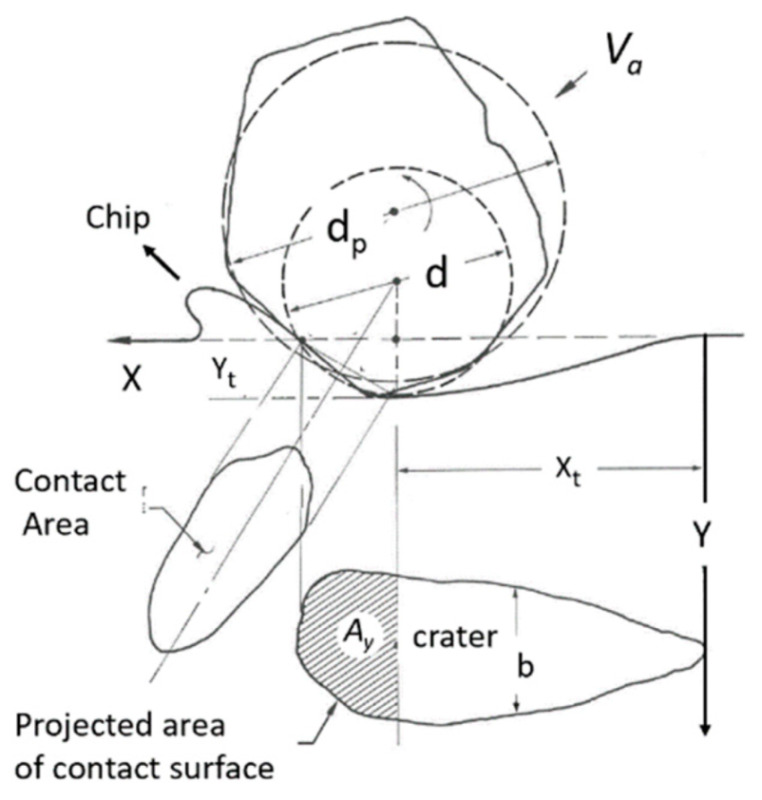
Erosion model.

**Figure 11 materials-17-03273-f011:**
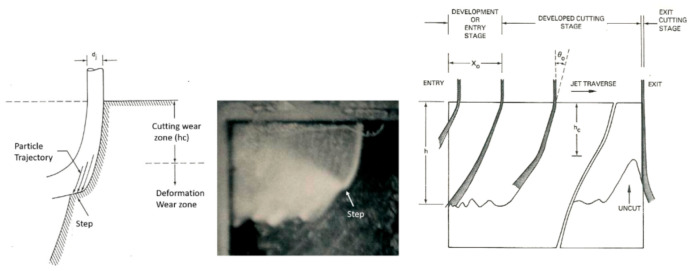
AWJ cutting model showing step and cutting stages.

**Figure 12 materials-17-03273-f012:**
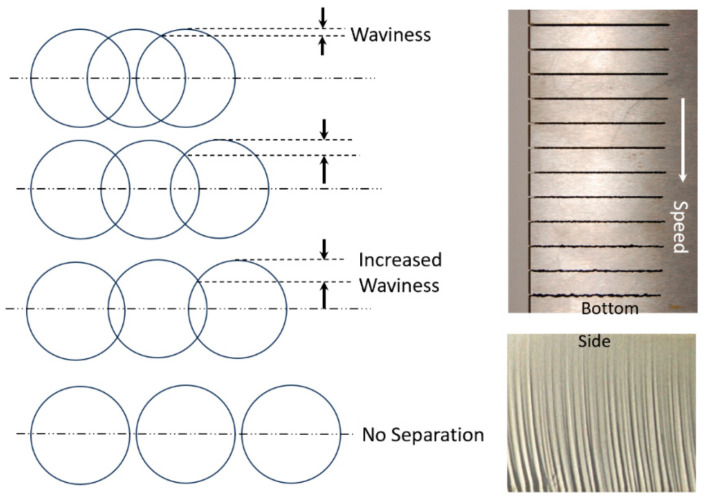
Idealized model of surface waviness.

**Figure 13 materials-17-03273-f013:**
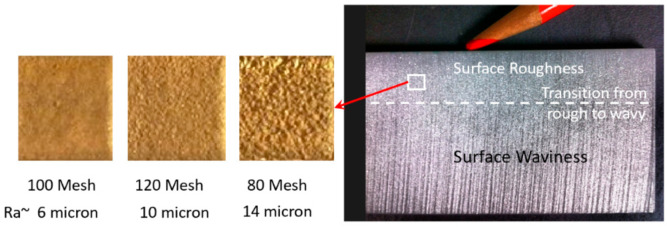
Effect of abrasive particle size on surface roughness.

**Figure 14 materials-17-03273-f014:**
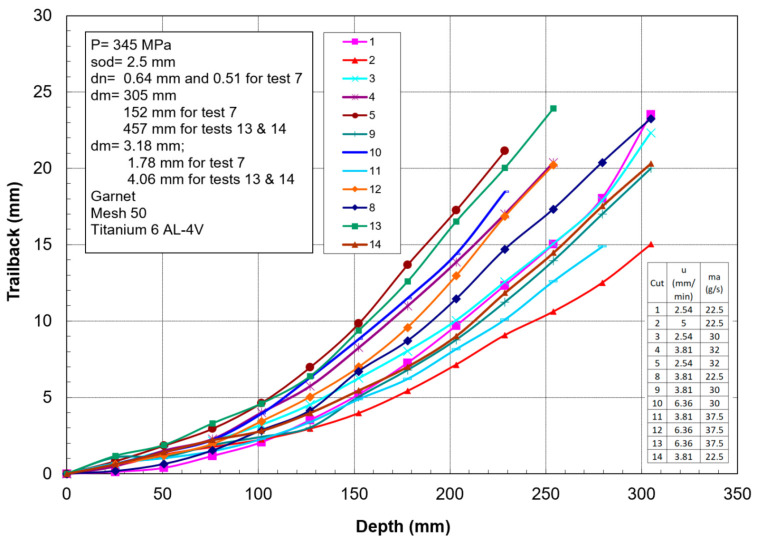
Trailback geometry for several cuts in titanium.

**Figure 15 materials-17-03273-f015:**
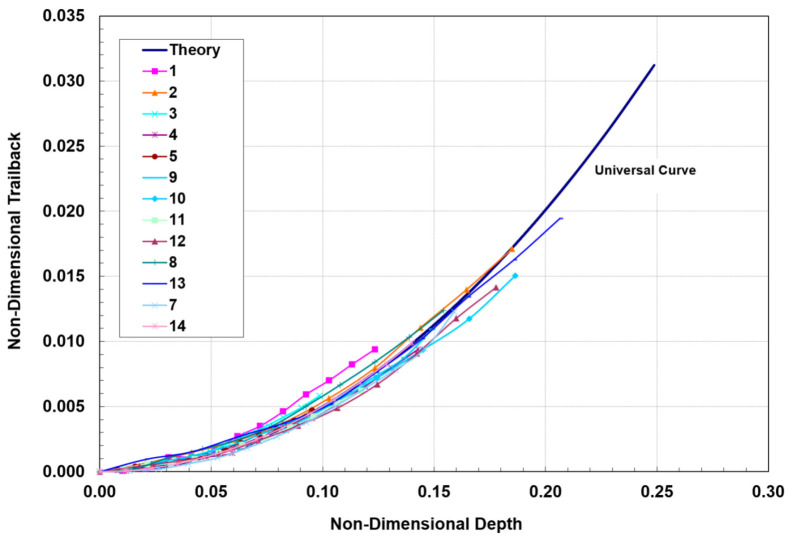
Universal kerf shape and data from Figure 14.

**Figure 16 materials-17-03273-f016:**
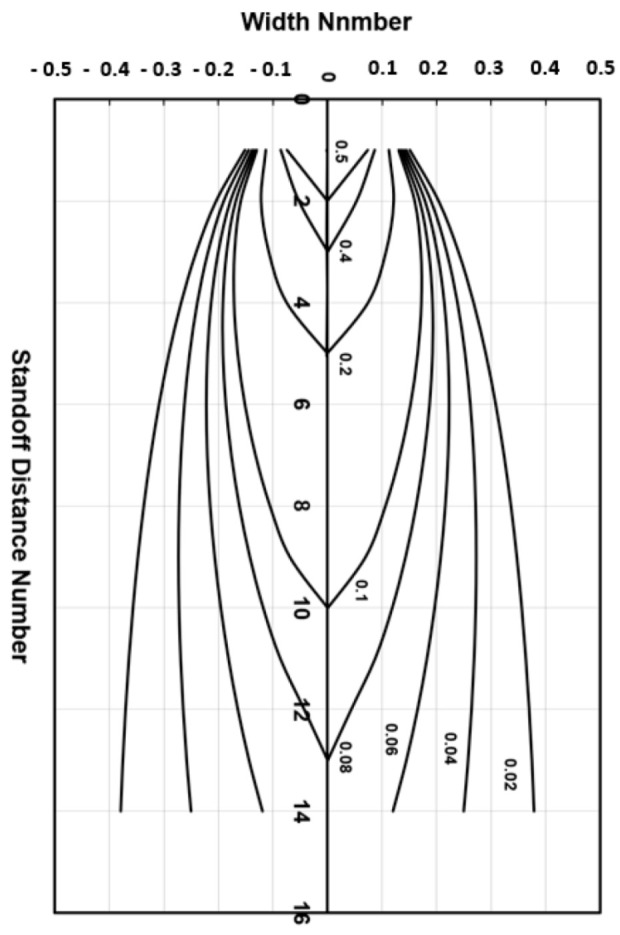
Theoretical waterjet kerf profiles.

**Figure 17 materials-17-03273-f017:**
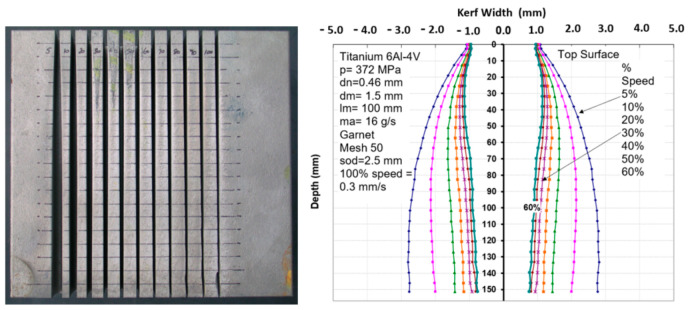
Kerf width profile in 150 mm thick titanium.

**Figure 18 materials-17-03273-f018:**
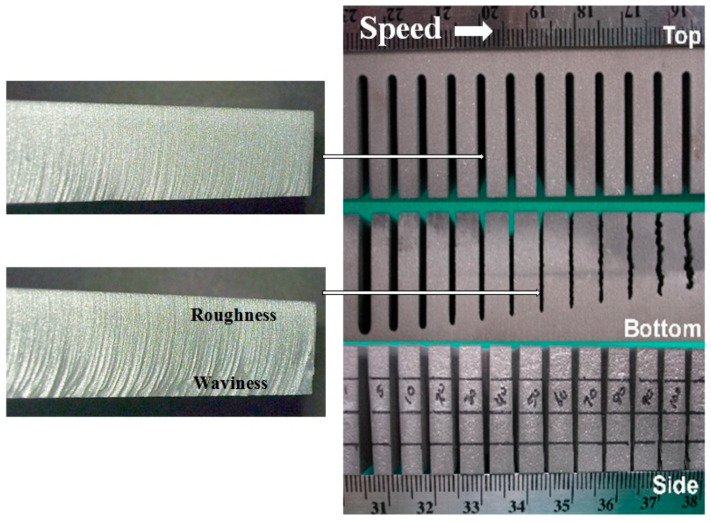
AWJ cuts showing kerf top, bottom, side, and surface morphology.

**Figure 19 materials-17-03273-f019:**
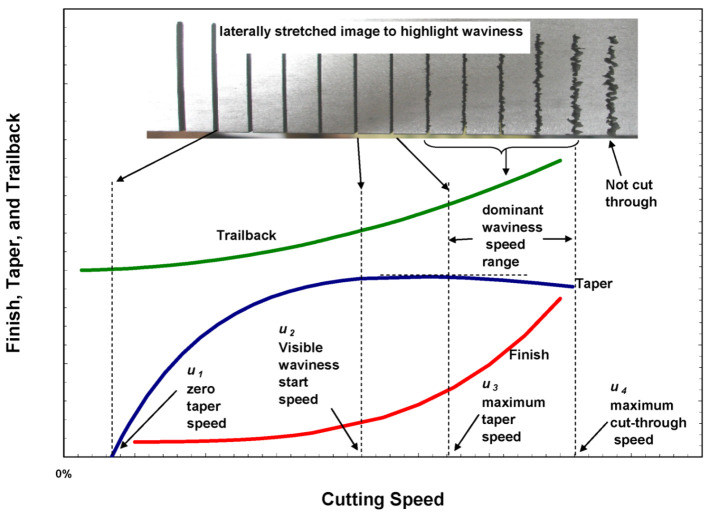
Cutting speed zones.

**Figure 20 materials-17-03273-f020:**
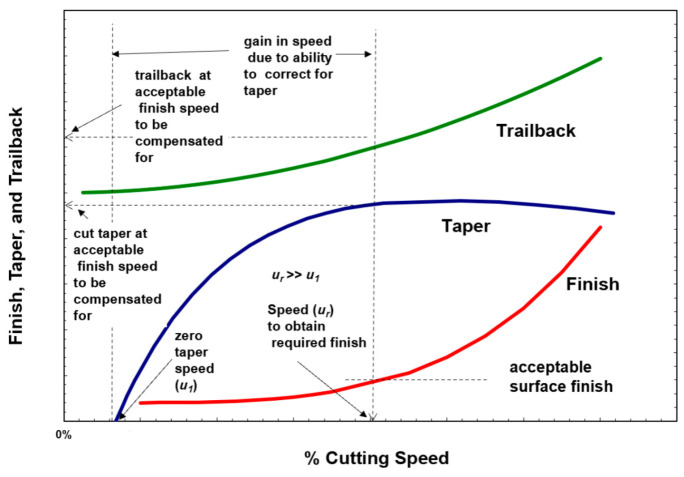
Gain in speed due to dynamic jet tilting.

**Figure 21 materials-17-03273-f021:**
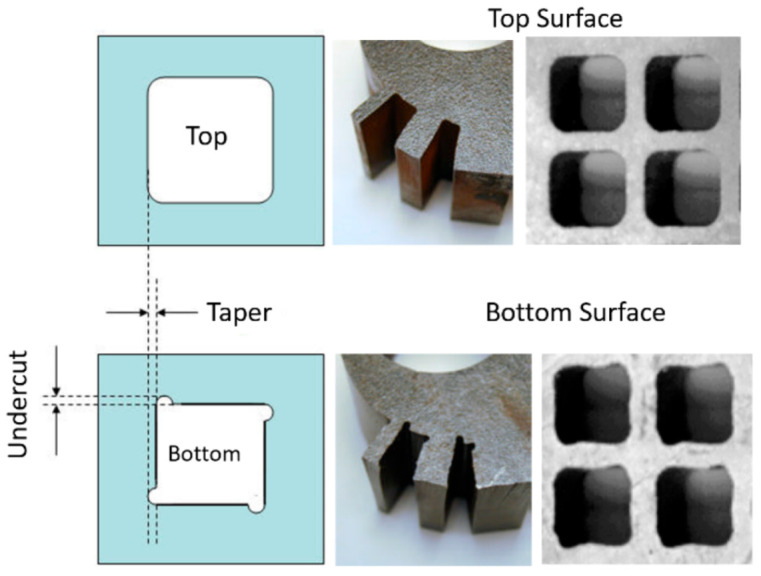
Undercutting at the bottom of cuts.

**Figure 22 materials-17-03273-f022:**
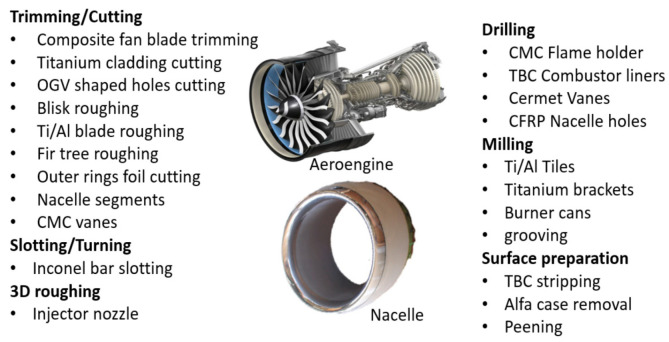
Examples of AWJ applications in jet engines.

**Figure 23 materials-17-03273-f023:**
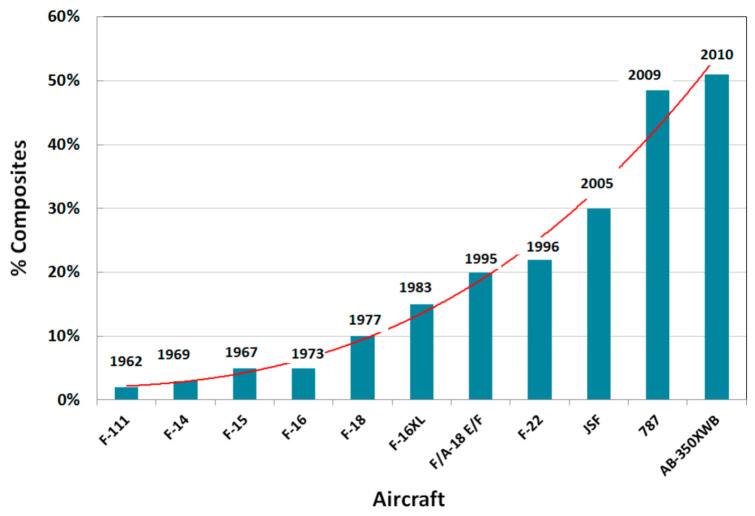
Composite use in aircraft.

**Figure 24 materials-17-03273-f024:**
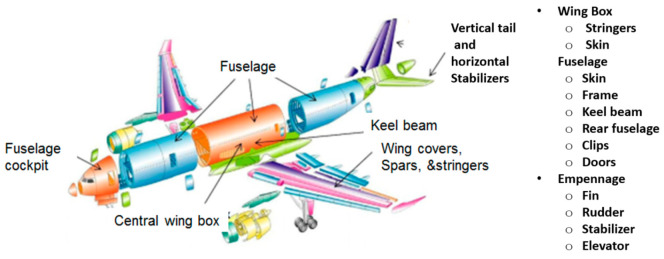
CFRP parts on the Airbus 350.

**Figure 25 materials-17-03273-f025:**
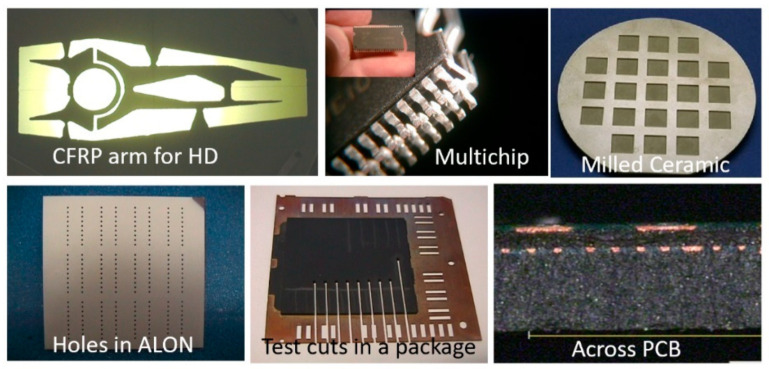
Examples of cutting microelectronic parts.

**Figure 26 materials-17-03273-f026:**
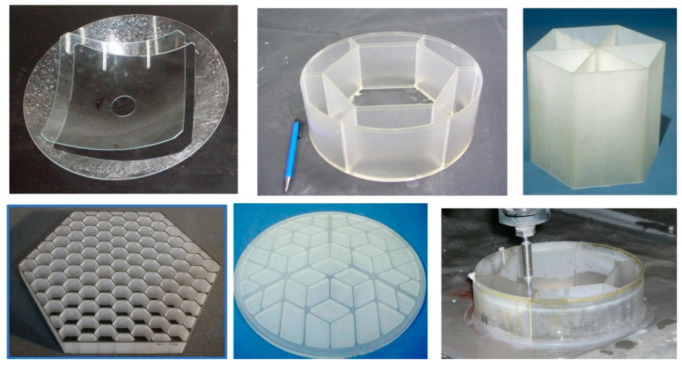
Examples of AWJ glass cutting and milling.

**Figure 27 materials-17-03273-f027:**
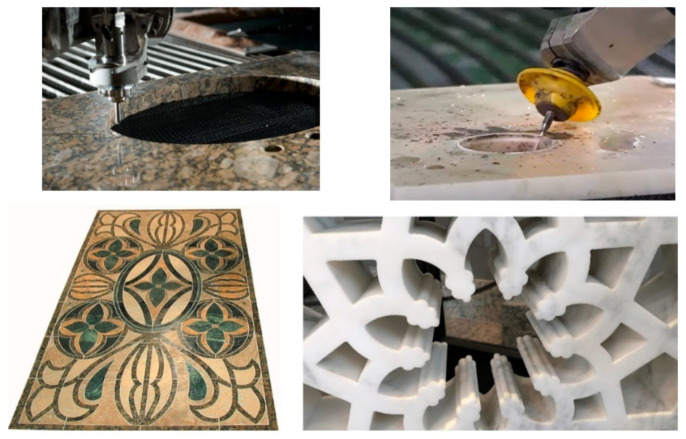
AWJ stone cutting and inlay.

**Figure 28 materials-17-03273-f028:**
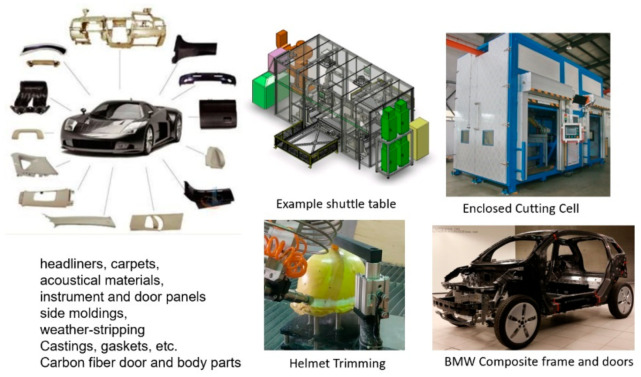
Examples of automotive applications.

**Figure 29 materials-17-03273-f029:**
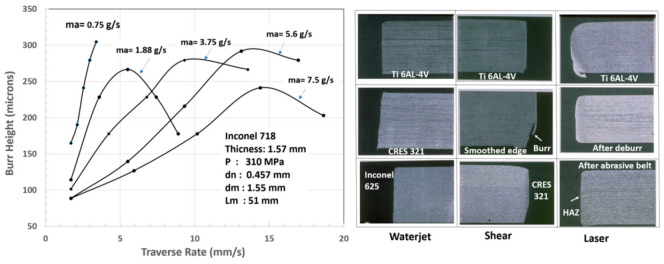
Example of thin sheet metal cutting data.

**Figure 30 materials-17-03273-f030:**
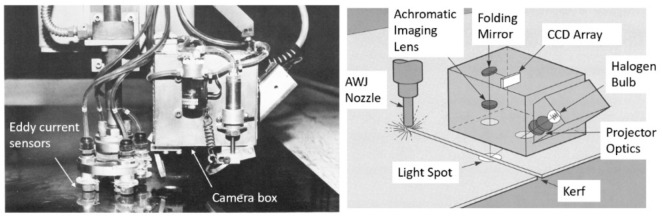
AWJ end effector with height sensor and vision camera.

**Figure 31 materials-17-03273-f031:**
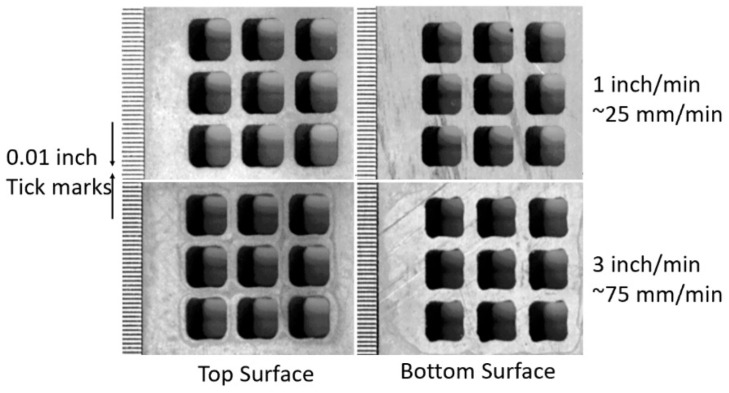
Sample holes for vent screens.

**Figure 32 materials-17-03273-f032:**
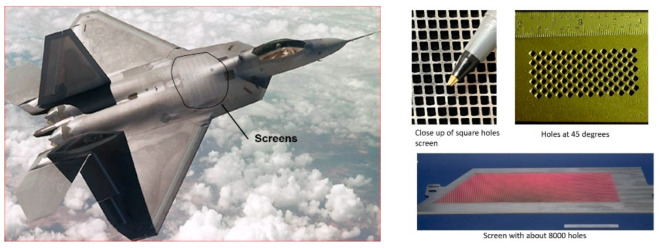
Titanium screen cutting.

**Figure 33 materials-17-03273-f033:**
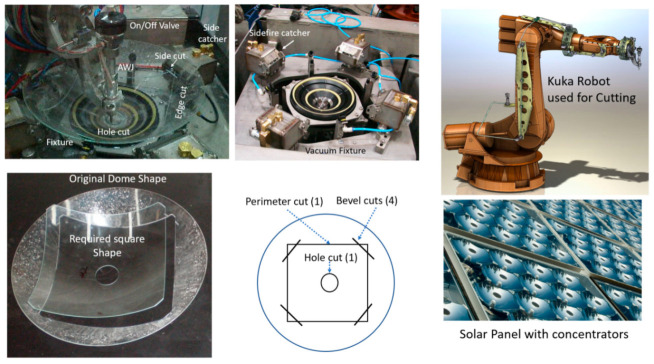
Precision robotic cutting of glass domes for solar panels.

**Figure 34 materials-17-03273-f034:**
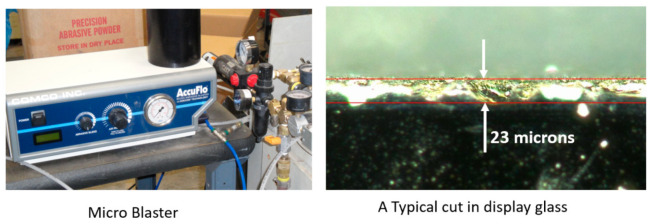
Microblaster abrasive feed for fine abrasives.

**Figure 35 materials-17-03273-f035:**
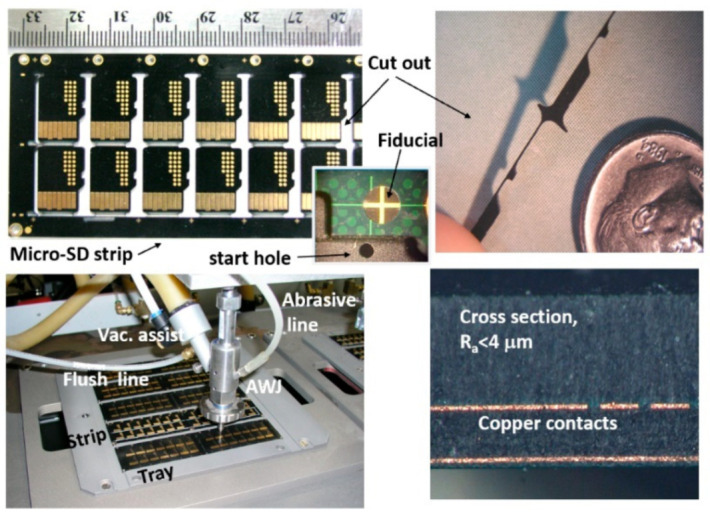
MicroSD singulation using AWJ technology.

**Figure 36 materials-17-03273-f036:**
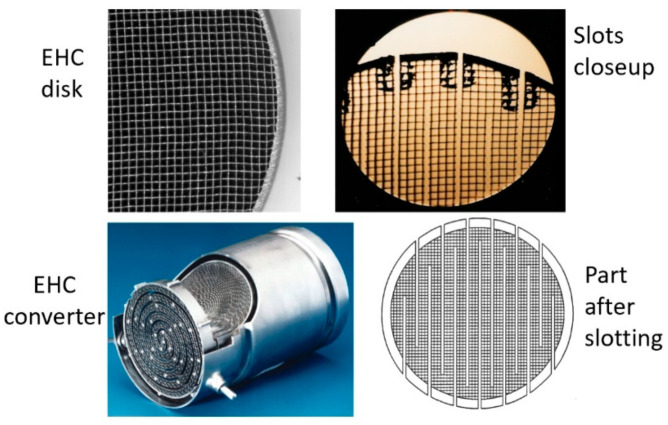
EHC before and after AWJ cutting.

**Figure 37 materials-17-03273-f037:**
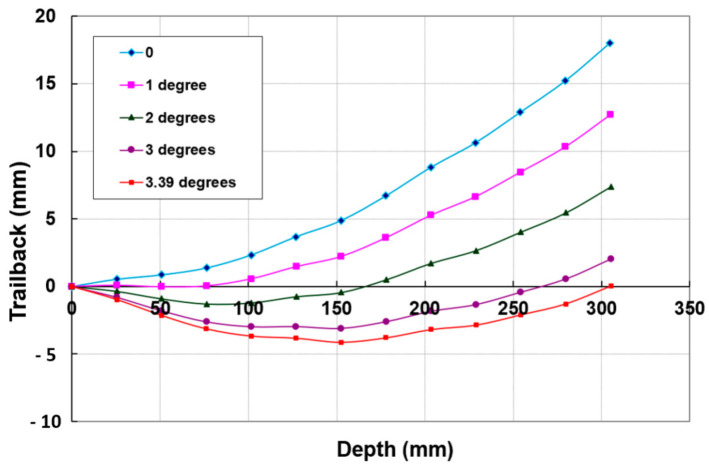
Trailback curve rotation with lead angles in glass.

**Figure 38 materials-17-03273-f038:**
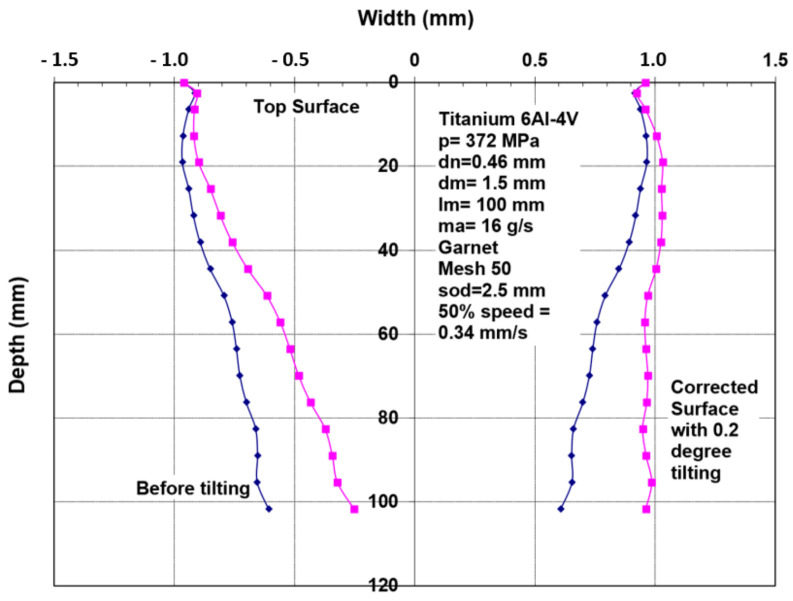
Kerf width profile rotation in 100 mm thick titanium.

**Figure 39 materials-17-03273-f039:**
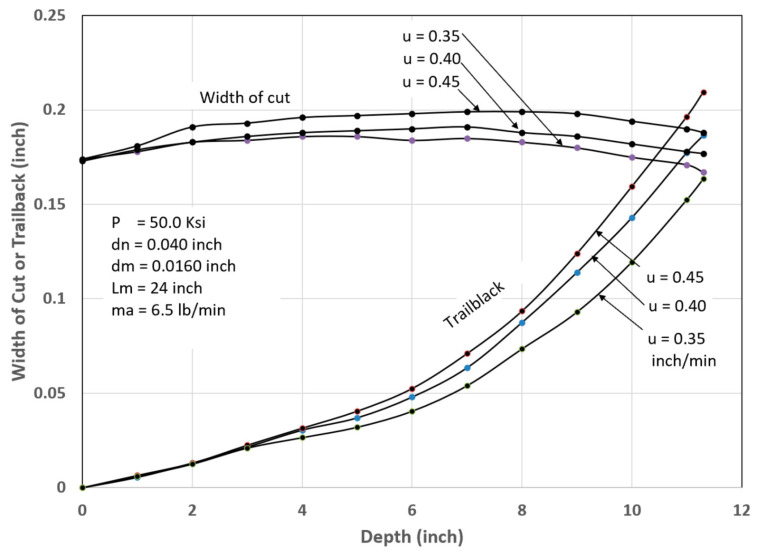
Example data on kerf width and trailback.

**Figure 40 materials-17-03273-f040:**
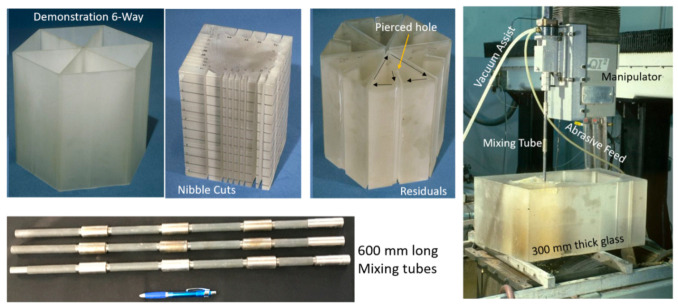
Thick glass cutting of small demo parts. The arrows show the cutting path.

**Figure 41 materials-17-03273-f041:**
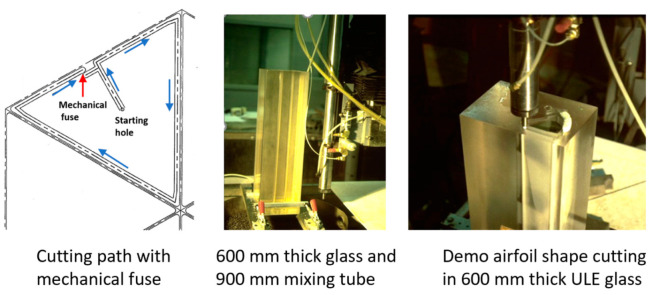
Cutting 600 mm thick glass using a 900 mm long mixing tube. The arrows show the cutting path starting from the center.

**Figure 42 materials-17-03273-f042:**
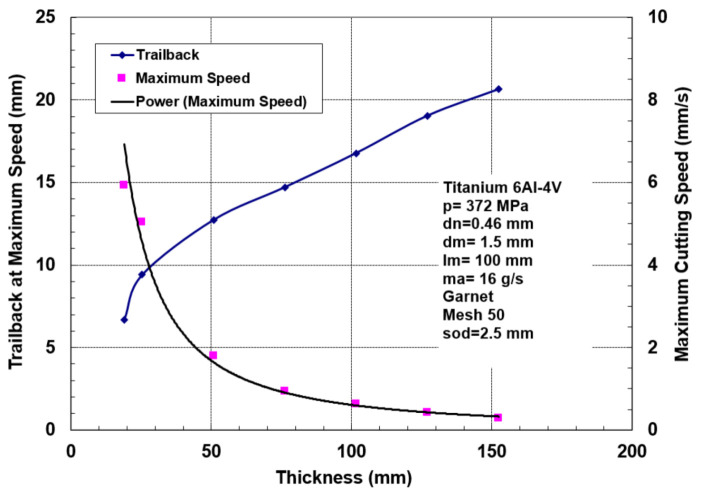
Maximum cutting speeds and trailbacks in titanium.

**Figure 43 materials-17-03273-f043:**
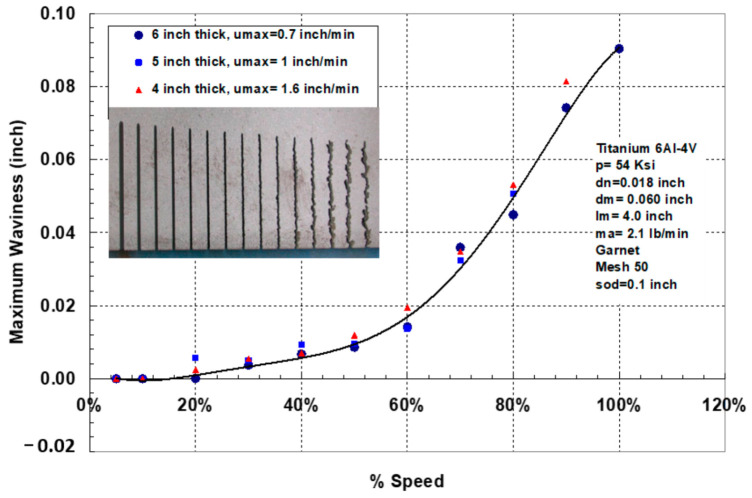
Surface waviness as a function of cutting speed.

**Figure 44 materials-17-03273-f044:**
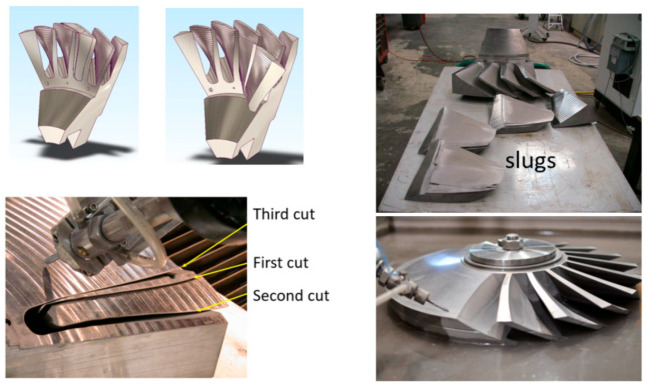
Near-net shaping of integrated jet engine rotor blades.

**Figure 45 materials-17-03273-f045:**
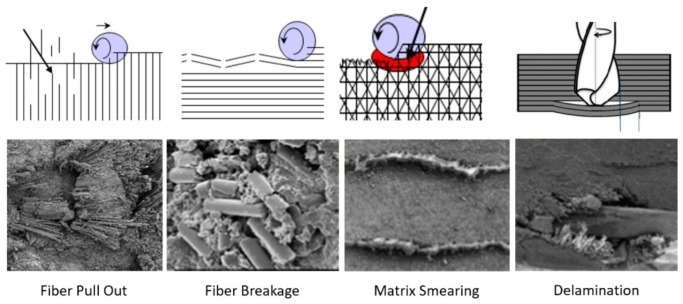
Potential integrity problems faced when using solid tools.

**Figure 46 materials-17-03273-f046:**
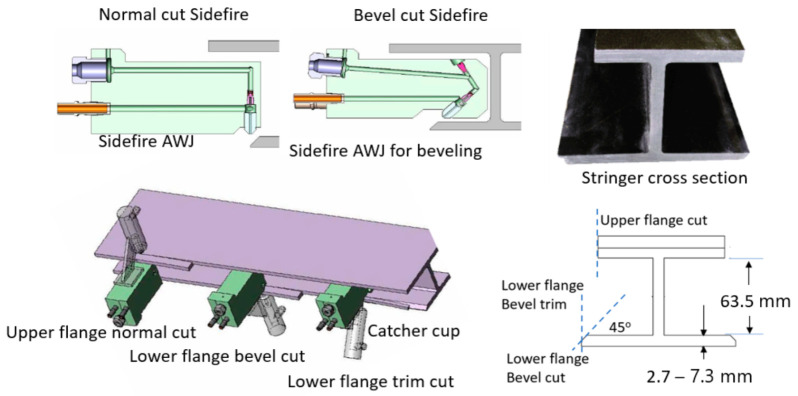
Stringer trimming for the Boeing 787 aircraft.

**Figure 47 materials-17-03273-f047:**
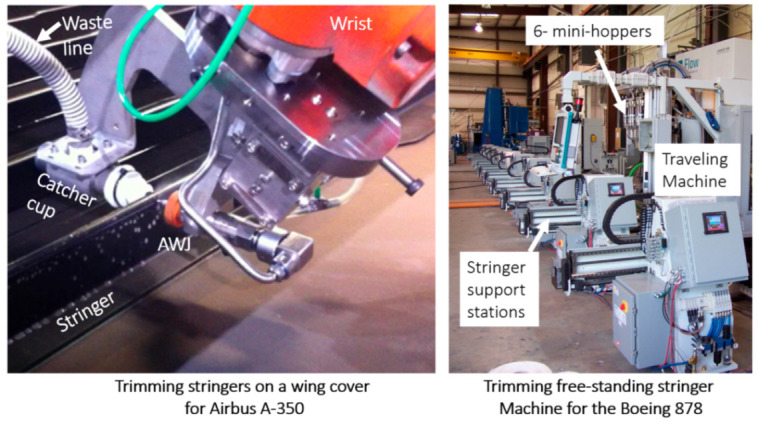
Stringer trimming machine for the Boeing 787 wing stringers.

**Figure 48 materials-17-03273-f048:**
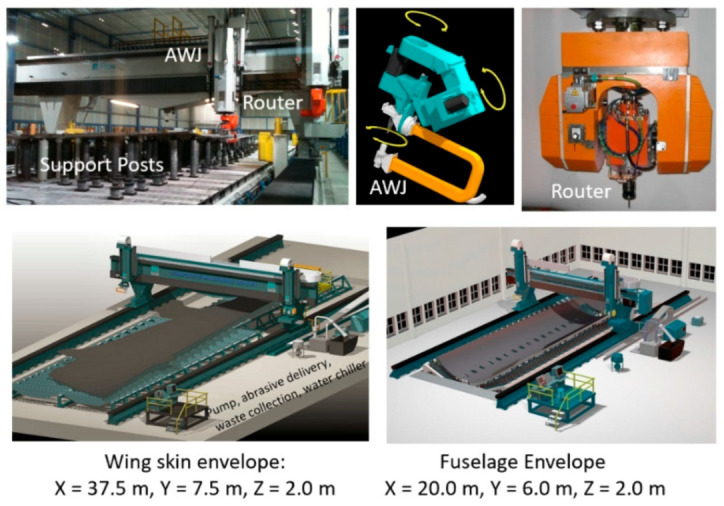
AWJ composite machining systems.

**Figure 49 materials-17-03273-f049:**
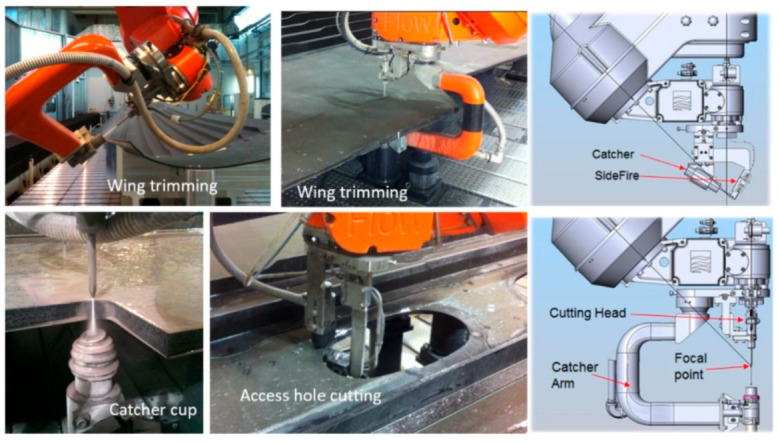
Focal point C-frame end effector for composite trimming.

**Figure 50 materials-17-03273-f050:**
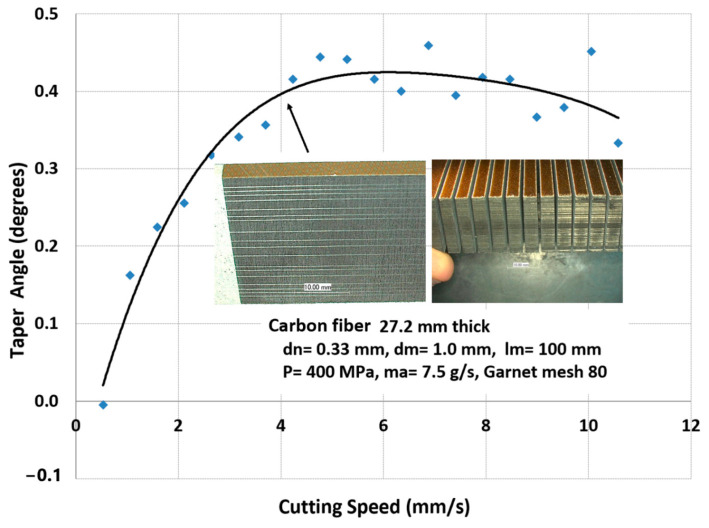
Sample AWJ-cut taper and finish.

**Figure 51 materials-17-03273-f051:**
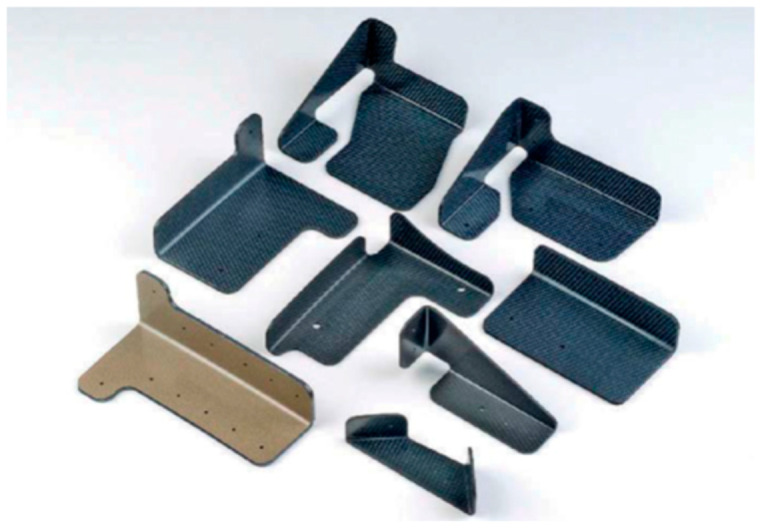
Examples of small composite parts.

**Figure 52 materials-17-03273-f052:**
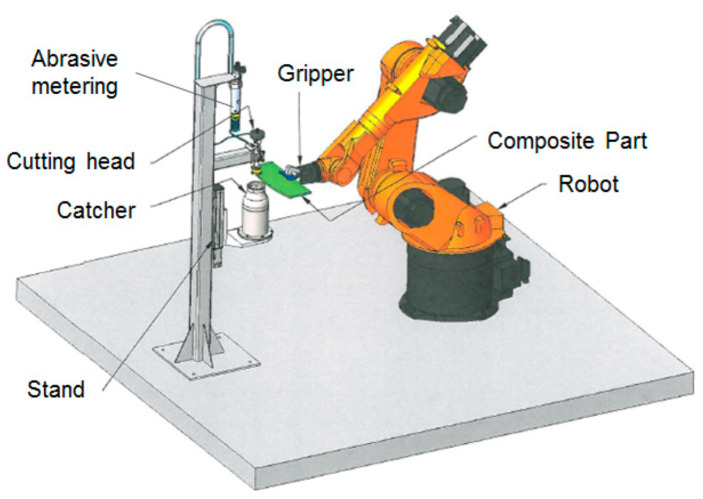
Robotic trimming of small composite clips.

**Figure 53 materials-17-03273-f053:**
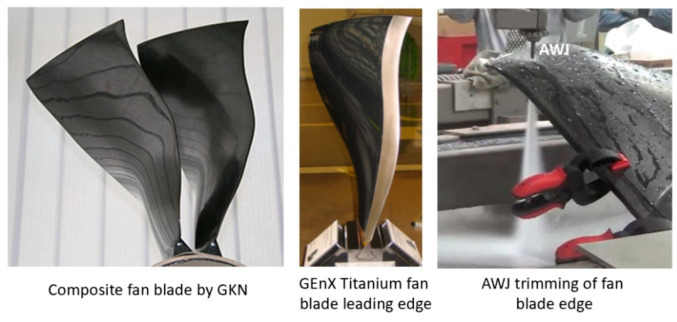
Geometry of fan blades and a blade trimming test.

**Figure 54 materials-17-03273-f054:**
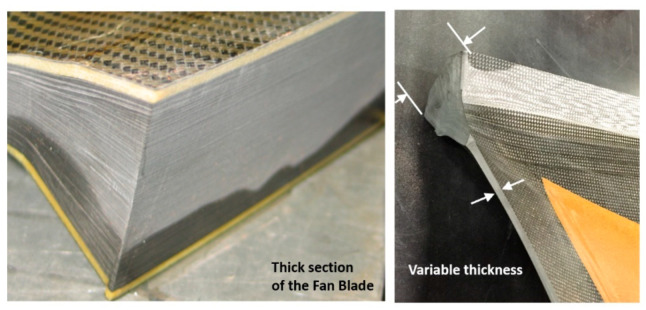
A varying thickness portion of an AWJ-trimmed fan blade.

**Figure 55 materials-17-03273-f055:**
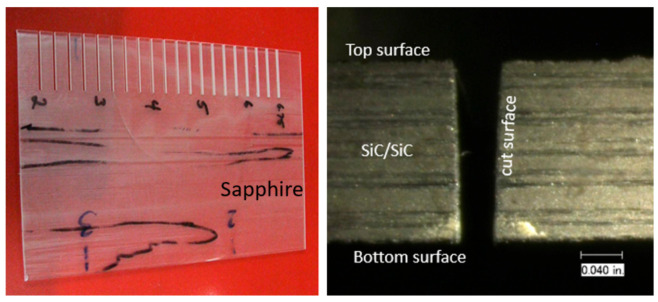
Cuts made in sapphire and SiC/SiC composite material.

**Figure 56 materials-17-03273-f056:**
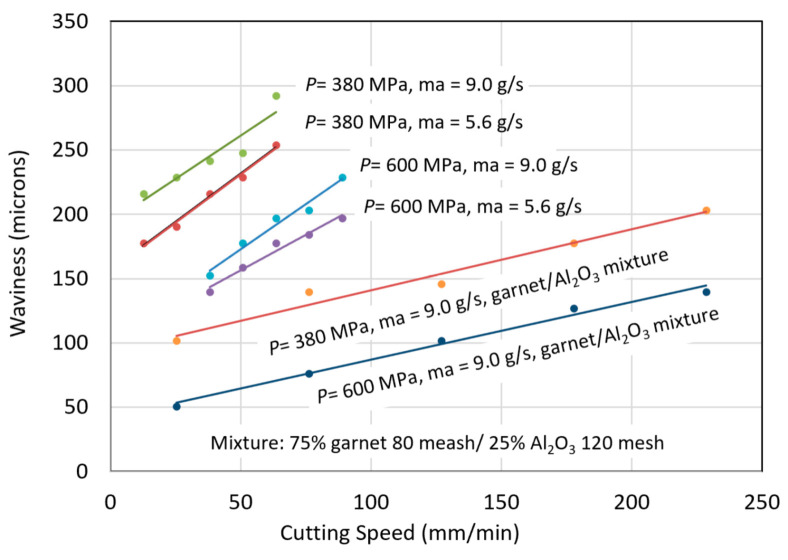
Effect of the cutting speed on the taper of the cut in SiC/SiC material.

**Figure 57 materials-17-03273-f057:**
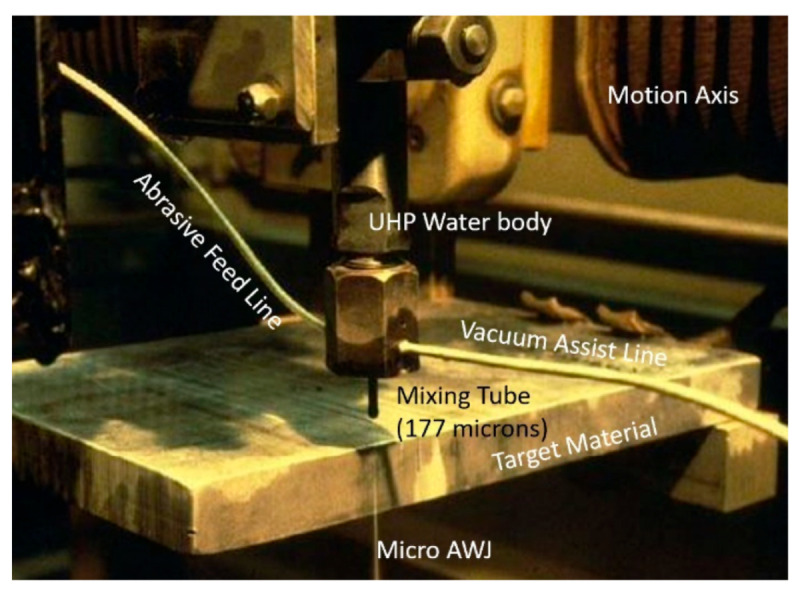
Micro AWJ with a 177-micron mixing tube [78].

**Figure 58 materials-17-03273-f058:**
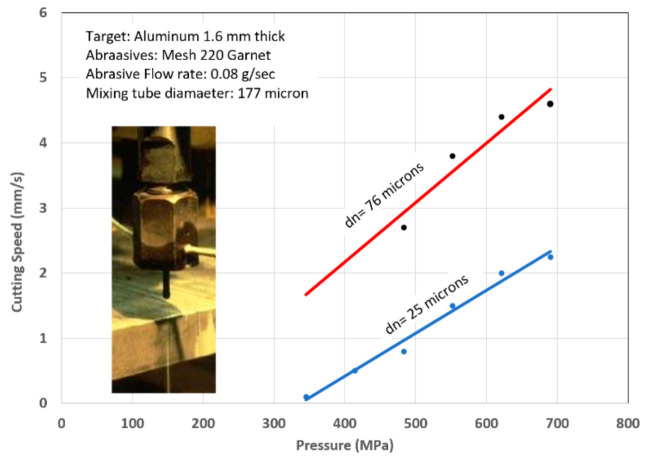
Cutting results with a micro AWJ tool.

**Figure 59 materials-17-03273-f059:**
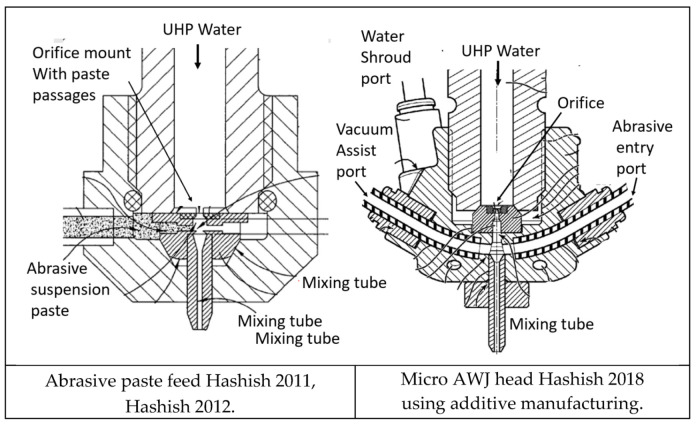
Patented micro AWJ cutting heads [81,82,83].

**Figure 60 materials-17-03273-f060:**
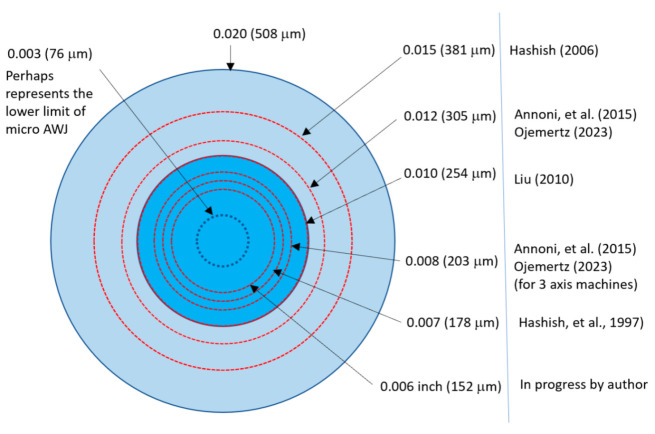
Micro AWJ sizes from the literatures [60,78,80,85,86].

**Figure 61 materials-17-03273-f061:**
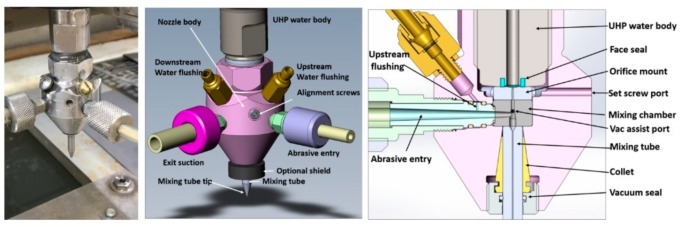
Micro AWJ cutting head, Craigen and Hashish [81].

**Figure 62 materials-17-03273-f062:**
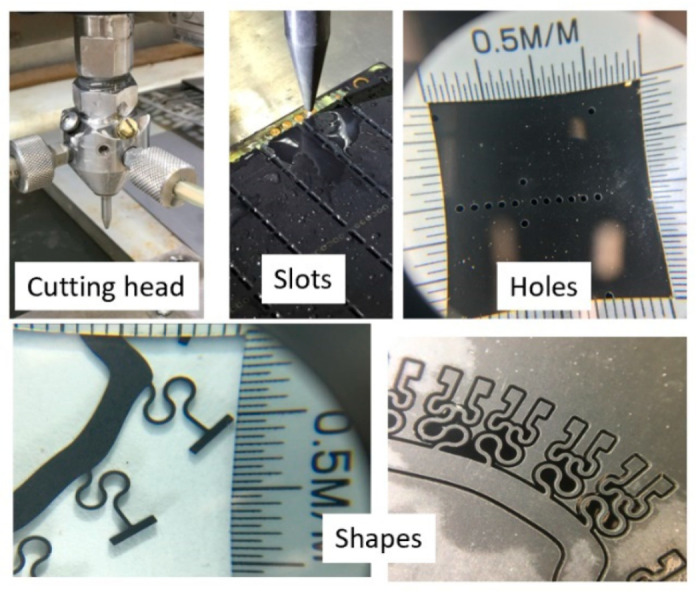
Example features machined with micro AWJ technology.

**Figure 63 materials-17-03273-f063:**
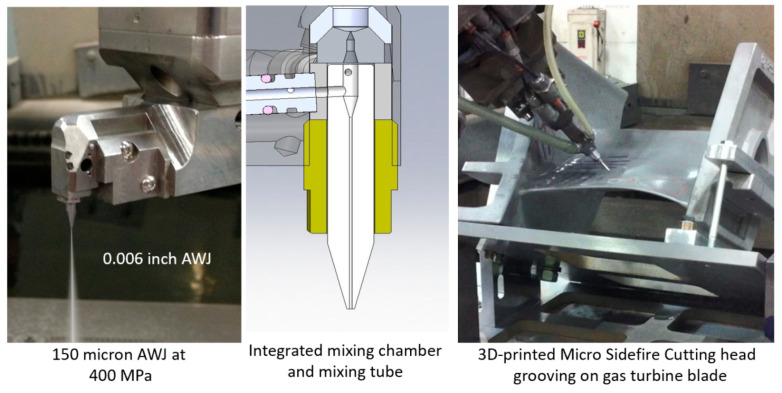
Micro sidefire AWJ cutting head for groove milling.

**Figure 64 materials-17-03273-f064:**
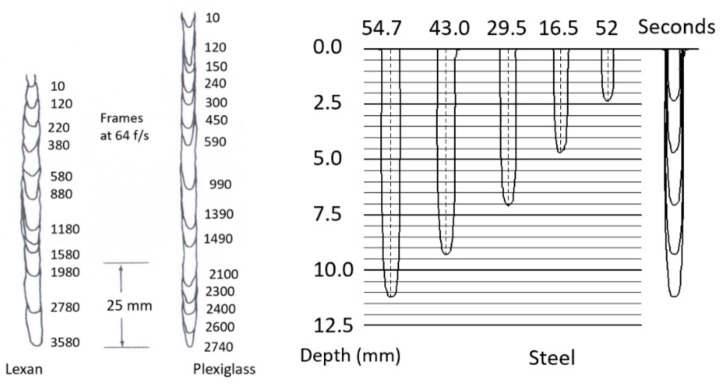
Hole piercing progression examples.

**Figure 65 materials-17-03273-f065:**
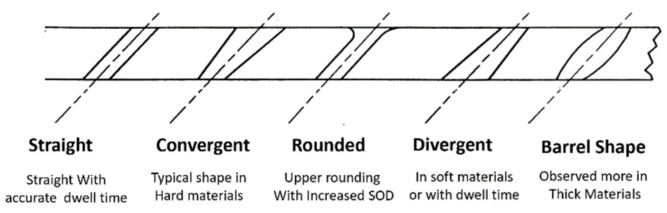
Possible hole shapes with AWJ piercing.

**Figure 66 materials-17-03273-f066:**
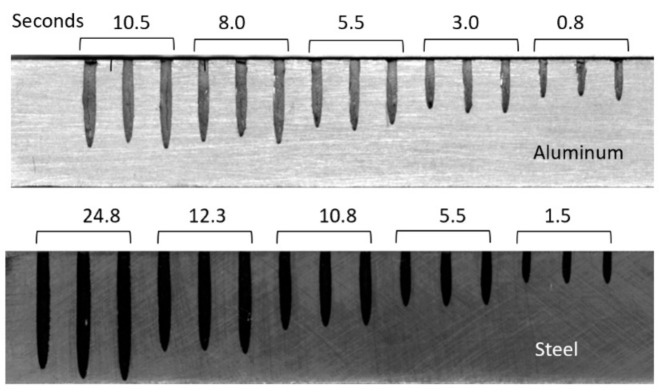
Cross-sections of holes drilled in aluminum and steel (*d_n_* = 0.18 mm, *d_m_* = 0.51 mm, *P* = 345 MPa).

**Figure 67 materials-17-03273-f067:**
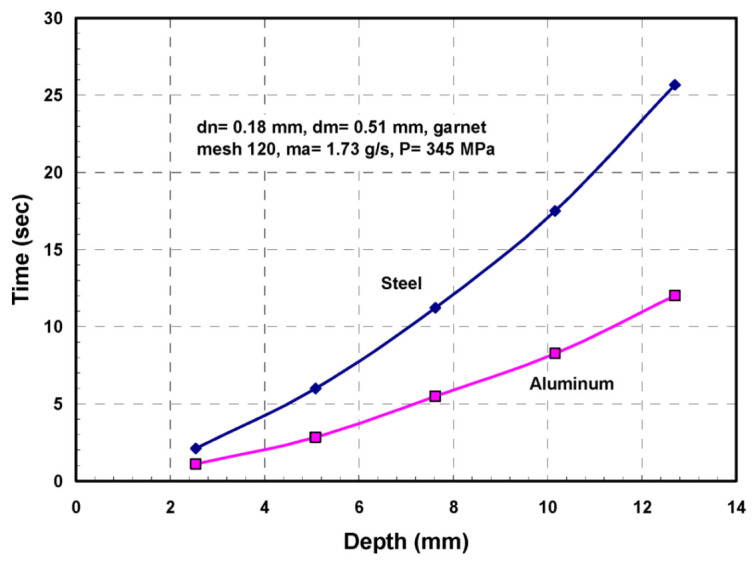
Drilling times for steel and aluminum at different depths.

**Figure 68 materials-17-03273-f068:**
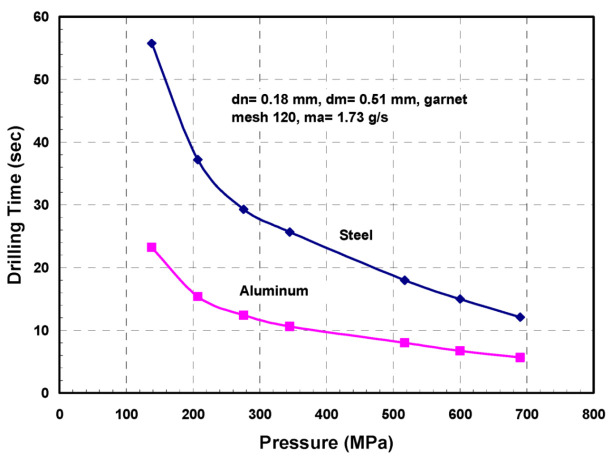
Effect of pressure on the drilling time for 12.7 mm thick steel and aluminum.

**Figure 69 materials-17-03273-f069:**
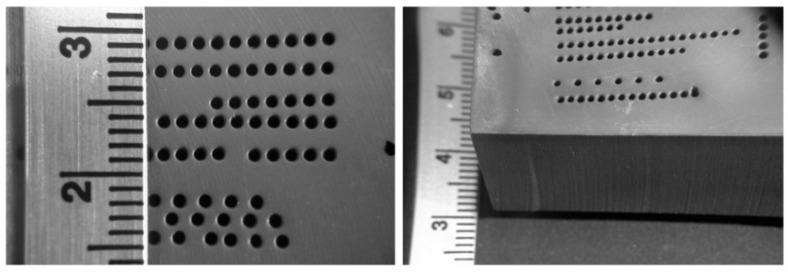
Small-diameter holes drilled in a 20 mm thick steel block.

**Figure 70 materials-17-03273-f070:**
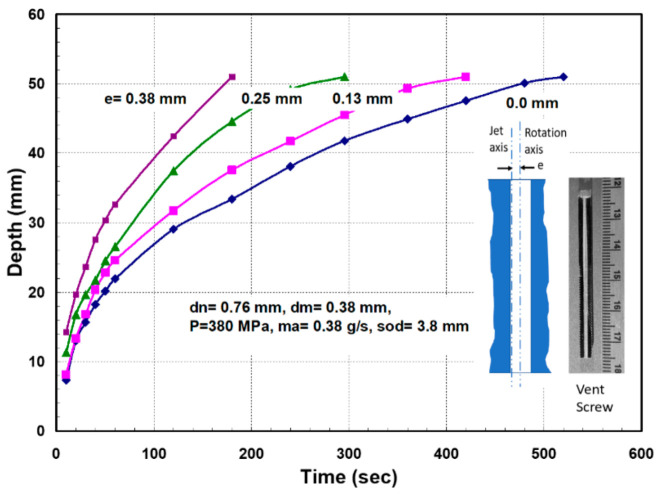
Hole depth progression in steel at different eccentricities.

**Figure 71 materials-17-03273-f071:**
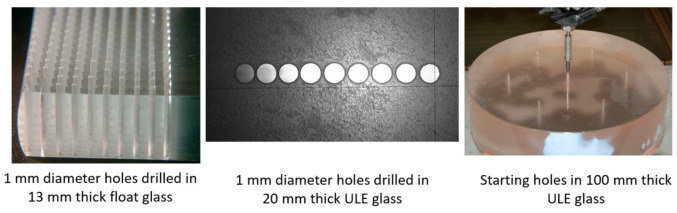
Holes drilled in glass.

**Figure 72 materials-17-03273-f072:**
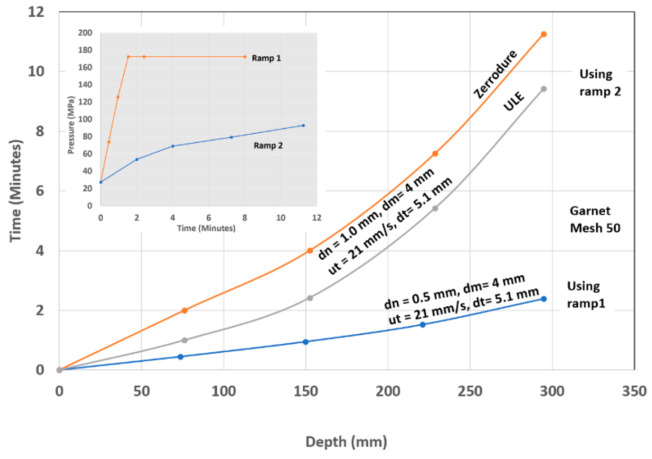
Drilling rate in thick glass under different conditions.

**Figure 73 materials-17-03273-f073:**
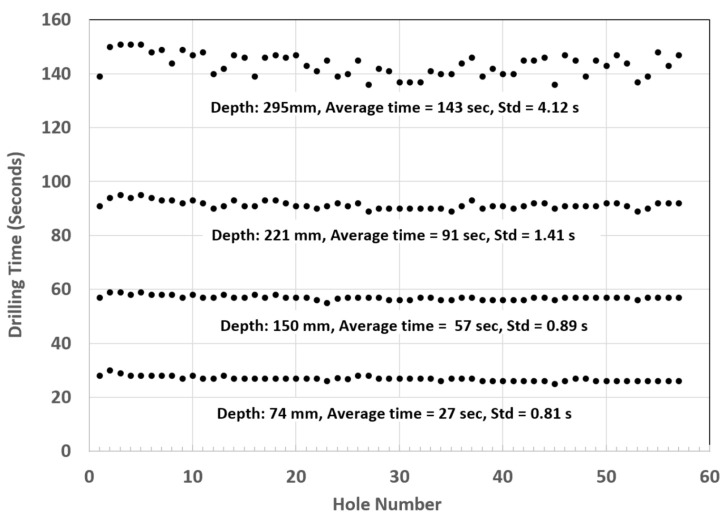
Repeatability of hole drilling in thick glass.

**Figure 74 materials-17-03273-f074:**
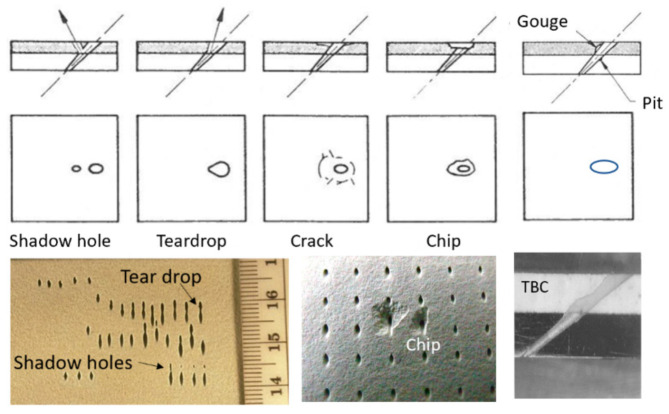
Possible problems of drilling a TBC using AWJ technology.

**Figure 75 materials-17-03273-f075:**
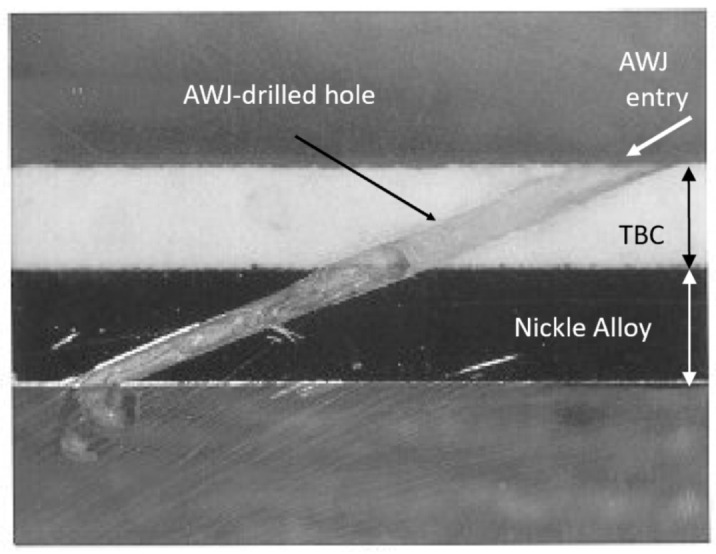
Accurately drilled hole at 30 degrees.

**Figure 76 materials-17-03273-f076:**
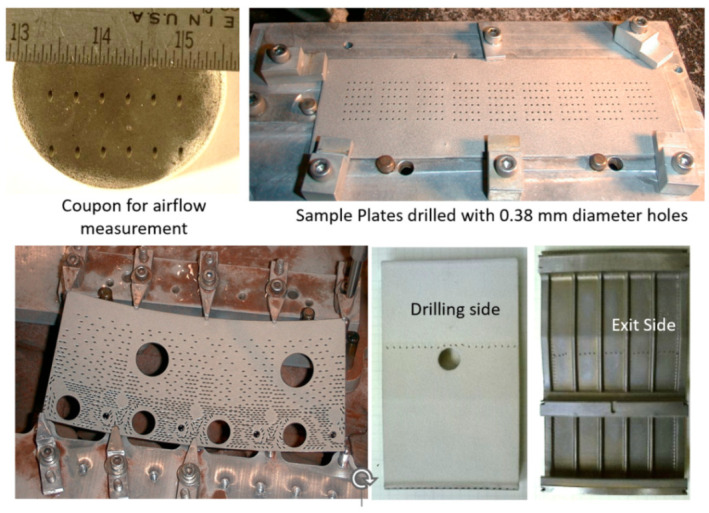
AWJ-drilled holes in TBC.

**Figure 77 materials-17-03273-f077:**
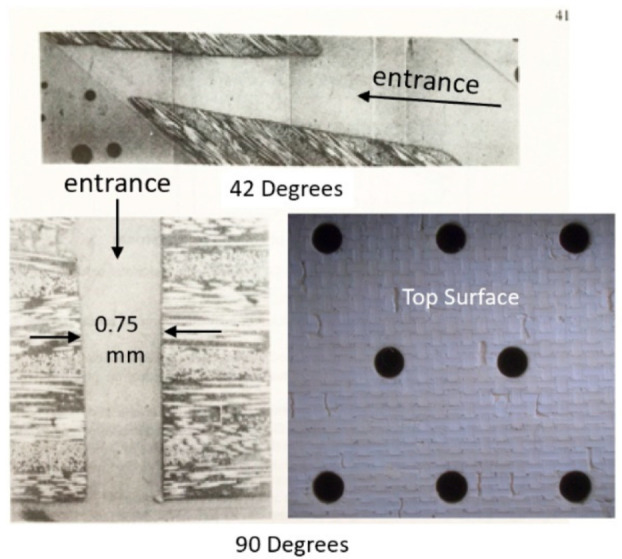
AWJ-drilled hole in an alumina/alumina CMC with no adverse effects.

**Figure 78 materials-17-03273-f078:**
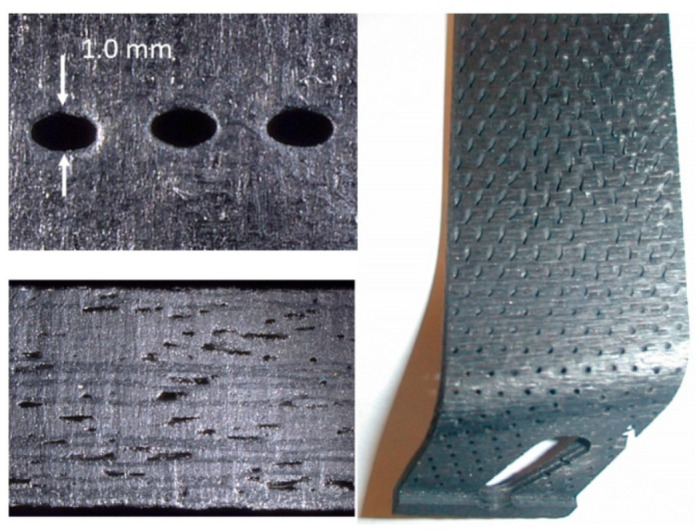
AWJ-drilled shaped hole in SiC/SiC material.

**Figure 79 materials-17-03273-f079:**
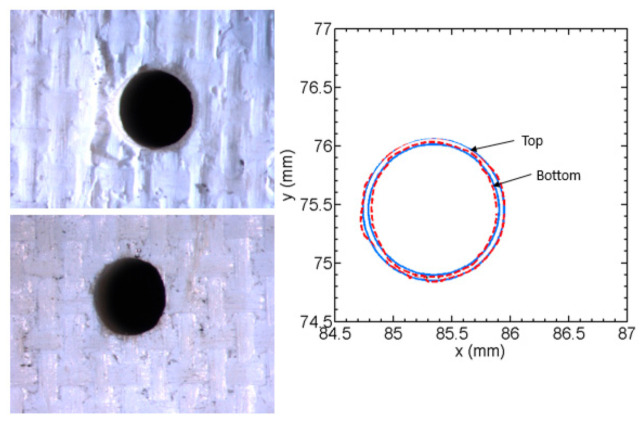
Hole circularity measurement in an alumina CMC.

**Figure 80 materials-17-03273-f080:**
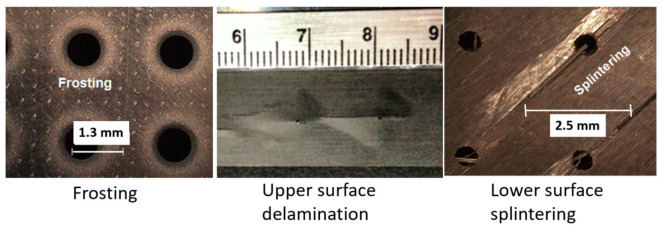
Potential failure modes in AWJ drilling.

**Figure 81 materials-17-03273-f081:**
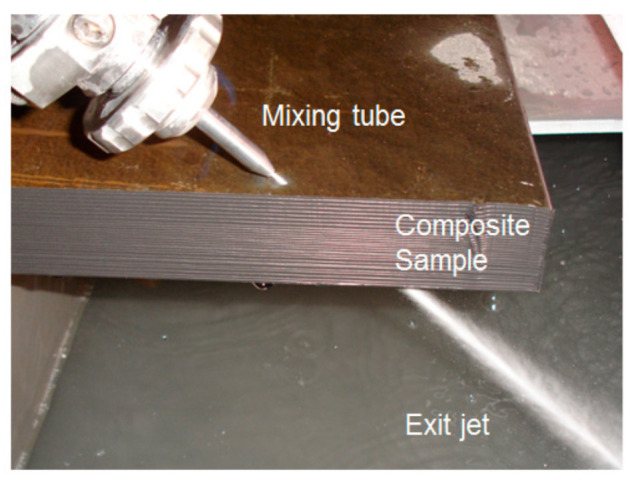
AWJ piercing through a 25 mm thick composite at a shallow angle.

**Figure 82 materials-17-03273-f082:**
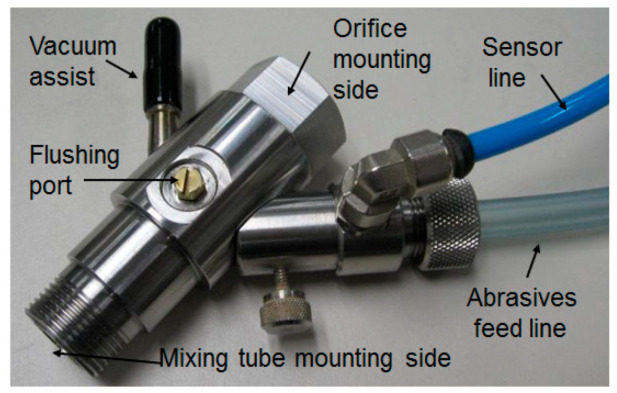
Cutting head for precision cutting and drilling.

**Figure 83 materials-17-03273-f083:**
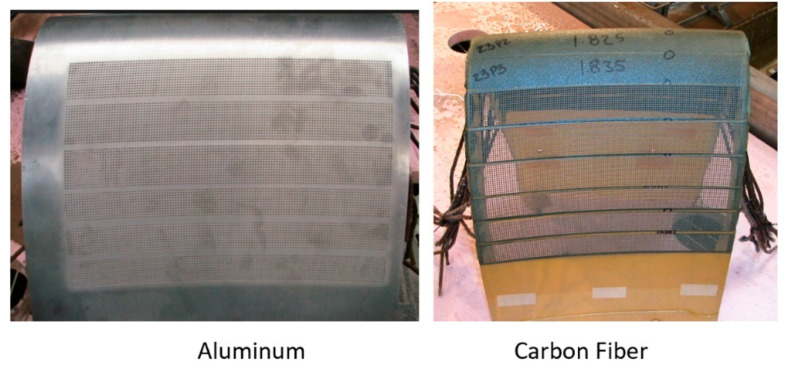
Matching holes drilled in aluminum and carbon fiber composite.

**Figure 84 materials-17-03273-f084:**
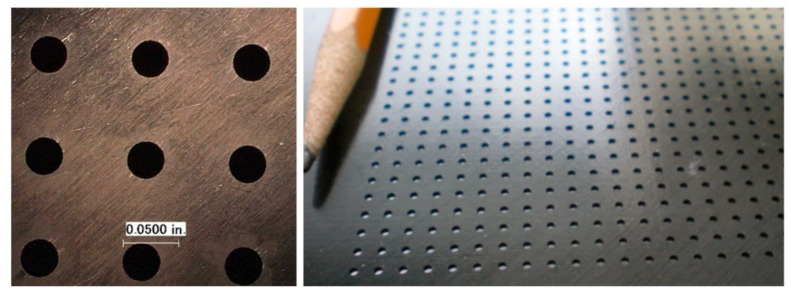
Closely packed AWJ-drilled holes in 1 mm thick CFRP.

**Figure 85 materials-17-03273-f085:**
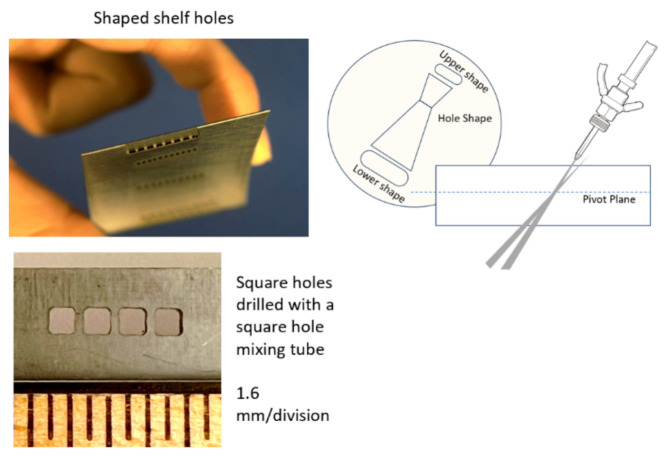
Example of pierced shaped holes.

**Figure 86 materials-17-03273-f086:**
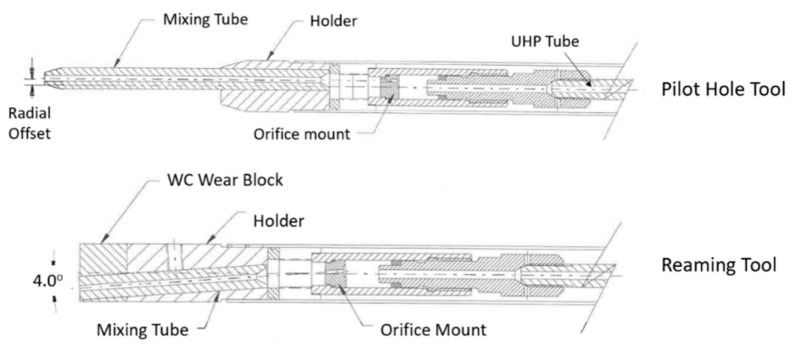
Hole boring pilot and reaming AWJ tools.

**Figure 87 materials-17-03273-f087:**
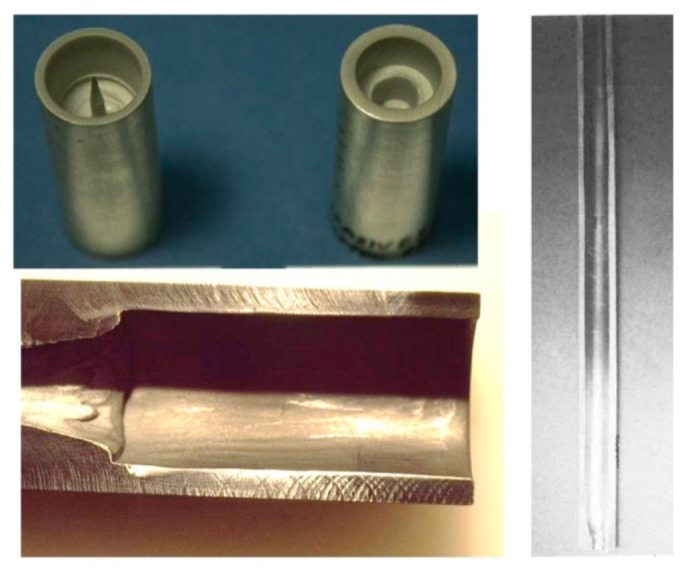
AWJ-bored holes.

**Figure 88 materials-17-03273-f088:**
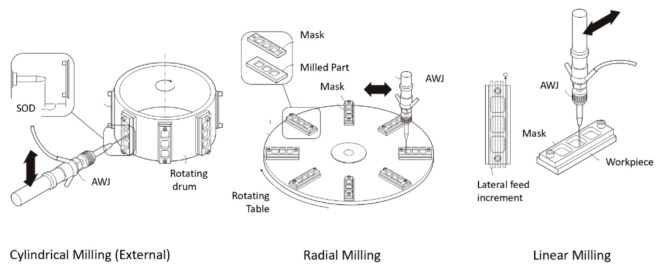
Milling methods using masks.

**Figure 89 materials-17-03273-f089:**
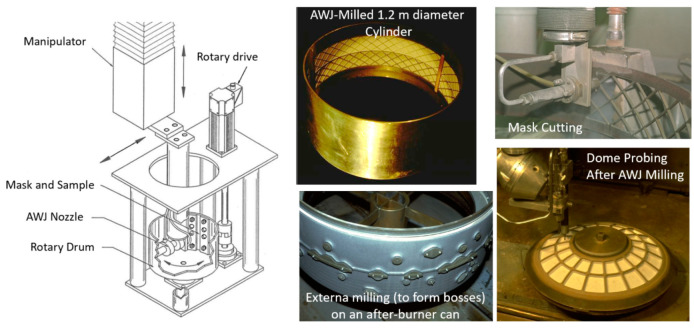
Isogrid milling (radial and cylindrical).

**Figure 90 materials-17-03273-f090:**
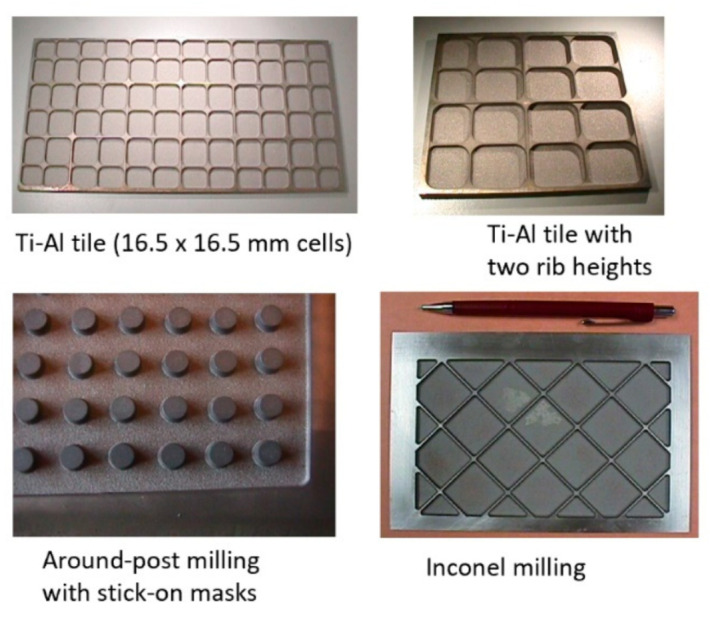
Milling of jet engine exhaust heat shield tiles.

**Figure 91 materials-17-03273-f091:**
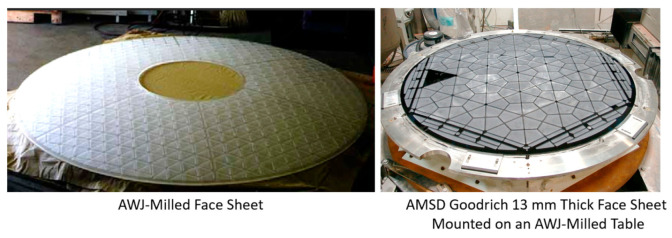
Milling of telescope face sheet glass.

**Figure 92 materials-17-03273-f092:**
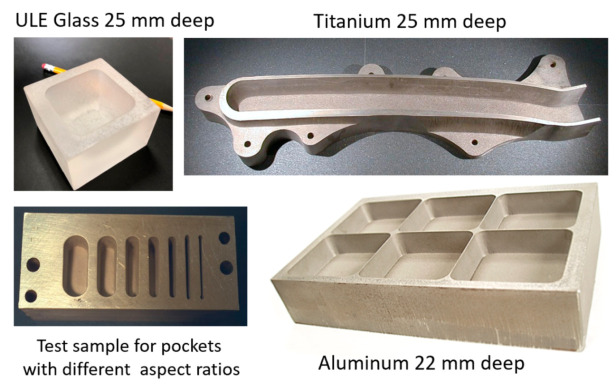
Deep pocket milling examples.

**Figure 93 materials-17-03273-f093:**
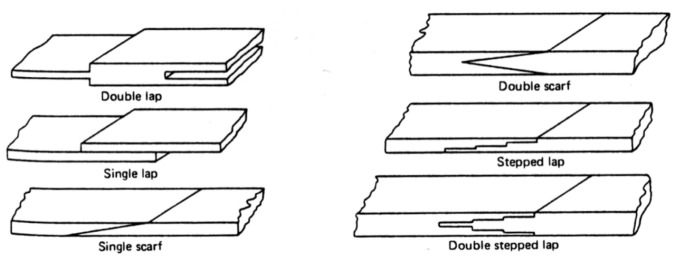
Typical composite joint designs.

**Figure 94 materials-17-03273-f094:**
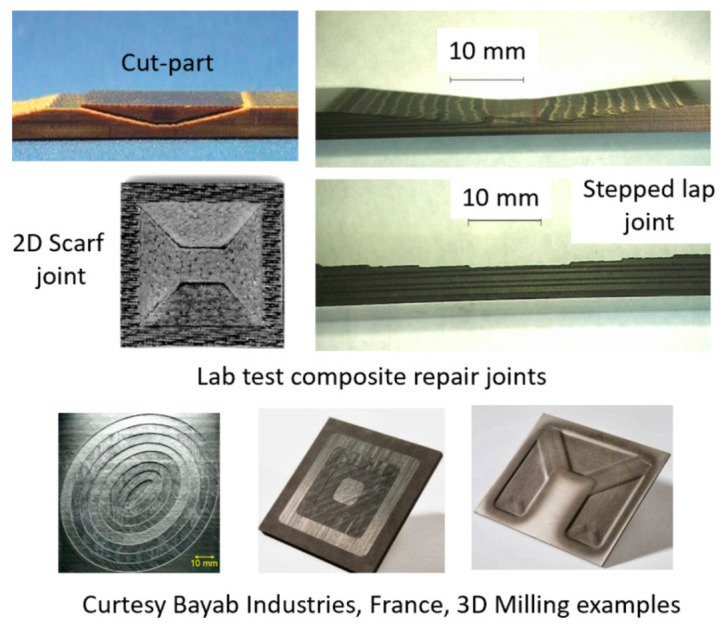
AWJ-milled composite repair joints.

**Figure 95 materials-17-03273-f095:**
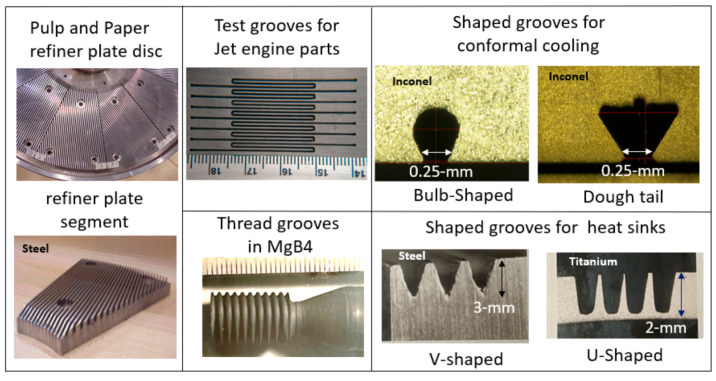
AWJ grooving examples.

**Figure 96 materials-17-03273-f096:**
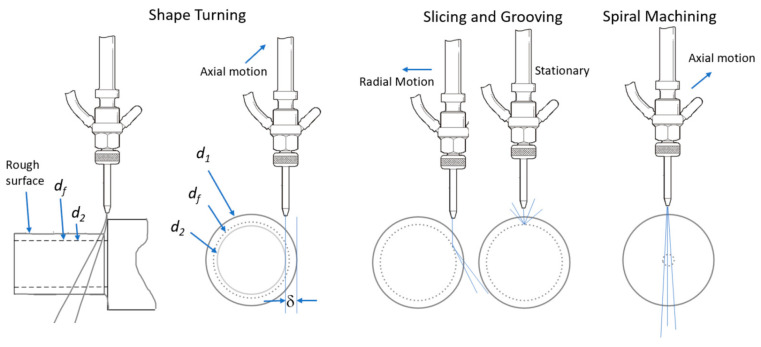
Turning operations.

**Figure 97 materials-17-03273-f097:**
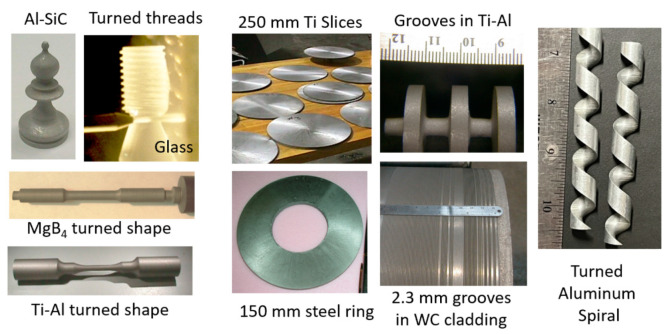
Example samples of turning operations.

**Figure 98 materials-17-03273-f098:**
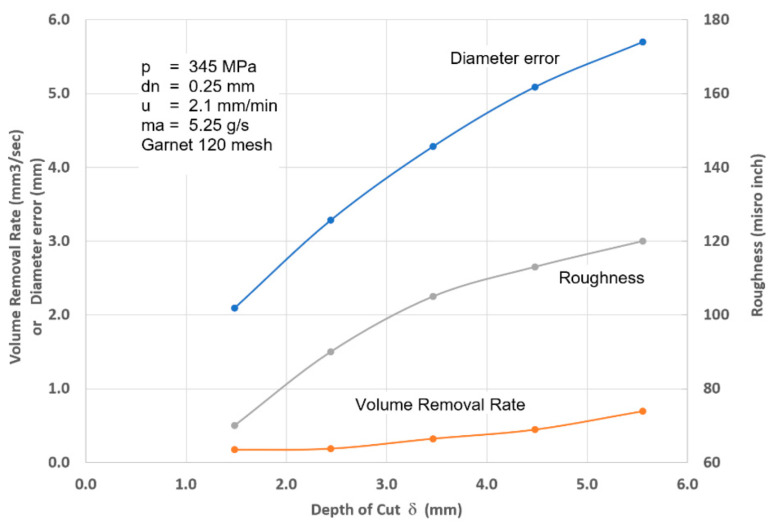
Example turning trends using Ti-Al target material.

**Figure 99 materials-17-03273-f099:**
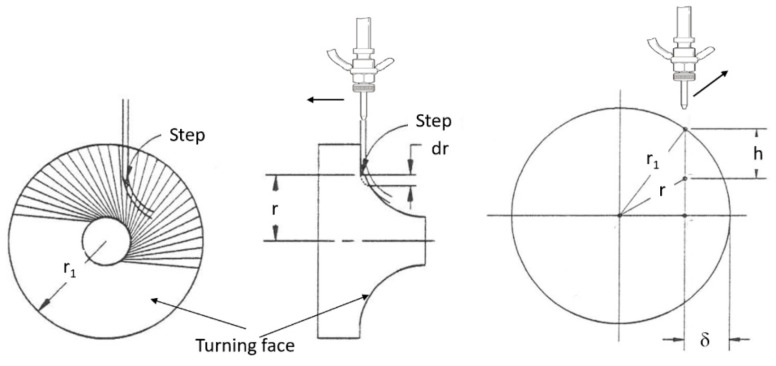
Turning process parameters used for modeling.

**Figure 100 materials-17-03273-f100:**
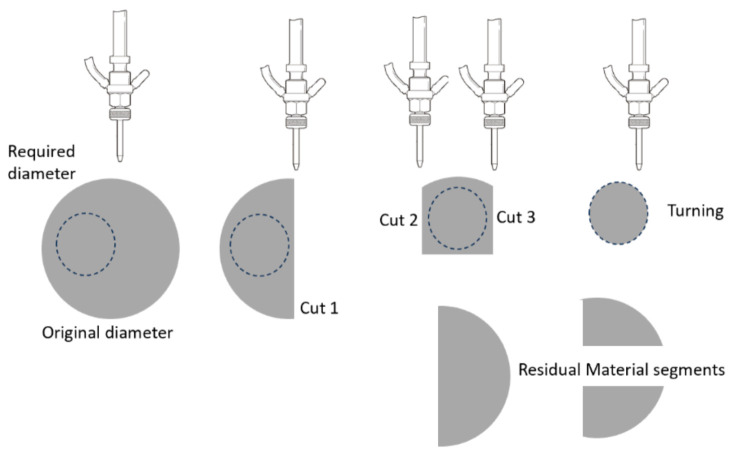
Segmental turning.

**Figure 101 materials-17-03273-f101:**
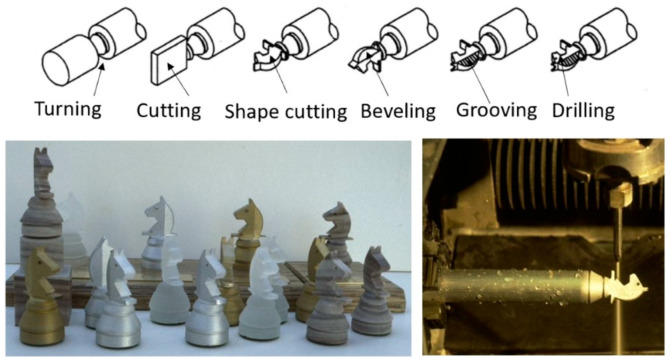
Example of early multi-operation machining.

**Figure 102 materials-17-03273-f102:**
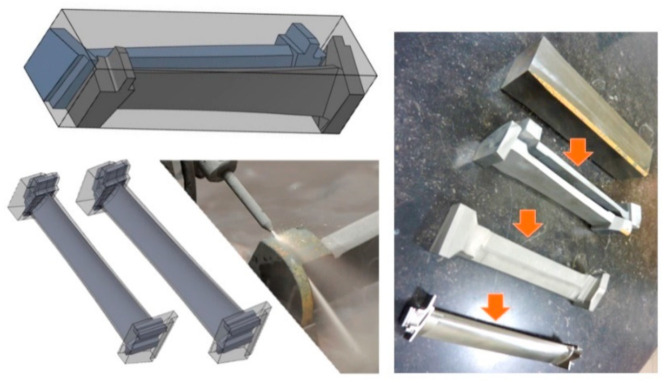
Shaping of titanium aluminide blades.

**Figure 103 materials-17-03273-f103:**
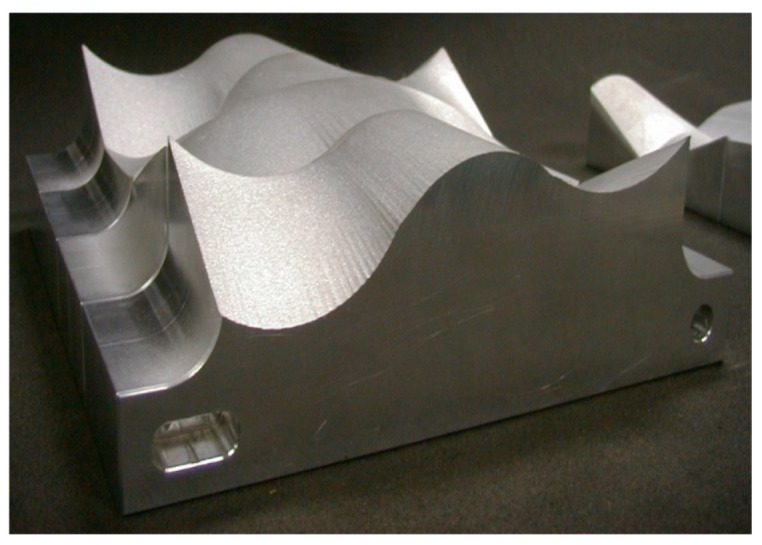
AWJ cuts for a 3D laminated object.

**Figure 104 materials-17-03273-f104:**
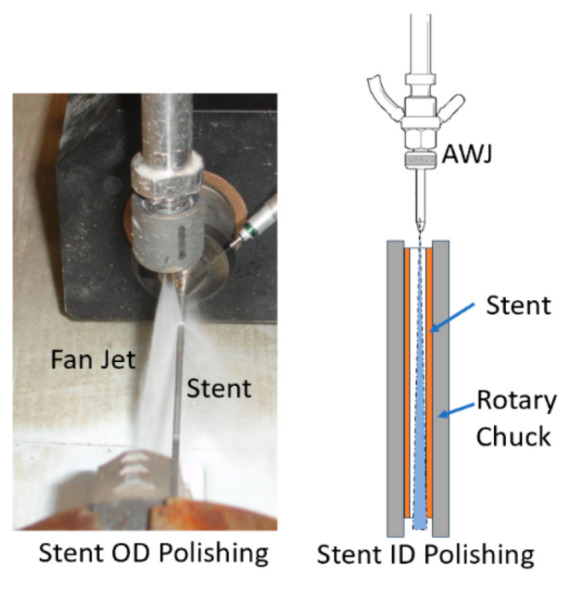
Stent polishing.

**Figure 105 materials-17-03273-f105:**
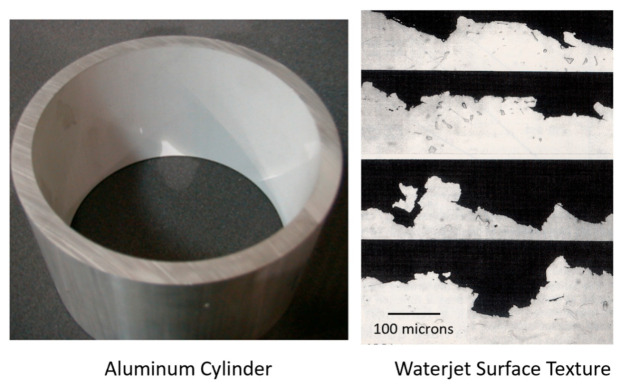
Waterjet texturing of automotive cylinder bore.

**Figure 106 materials-17-03273-f106:**
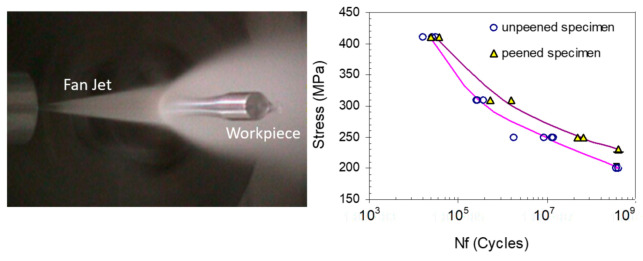
Effect of waterjet impact of fatigue parameters for 7075-T6 aluminum.

**Figure 107 materials-17-03273-f107:**
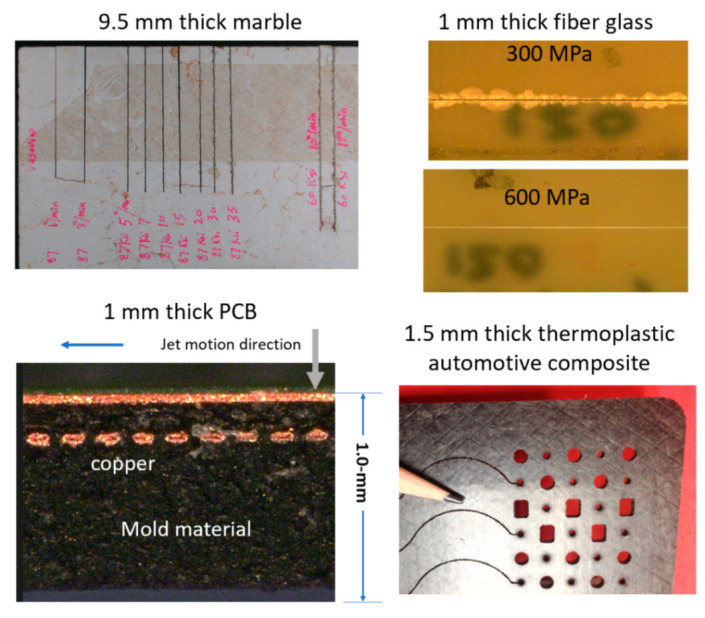
Examples of pure waterjet cutting at elevated pressures.

**Figure 108 materials-17-03273-f108:**
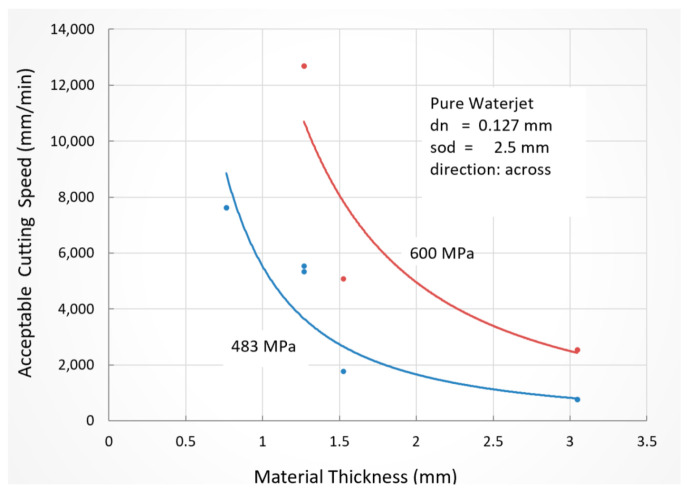
Effect of pressure on non-delamination cutting speed.

**Figure 109 materials-17-03273-f109:**
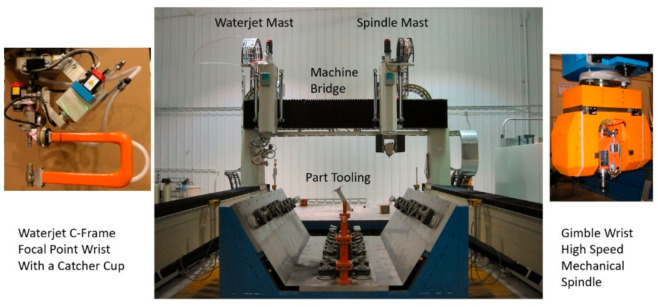
Mechanical–waterjet hybrid composite machining system.

**Figure 110 materials-17-03273-f110:**
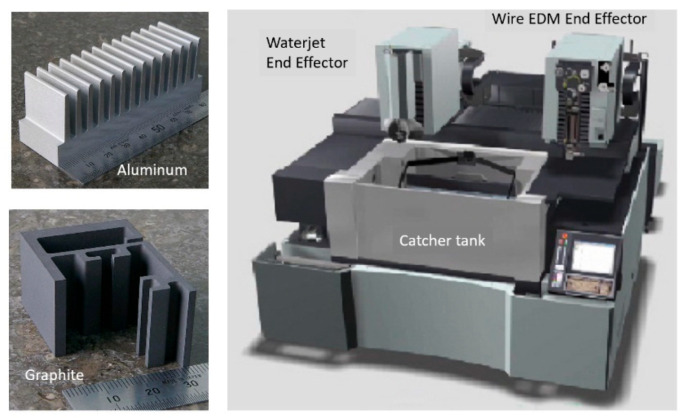
AWJ-EDM hybrid machine.

**Figure 111 materials-17-03273-f111:**
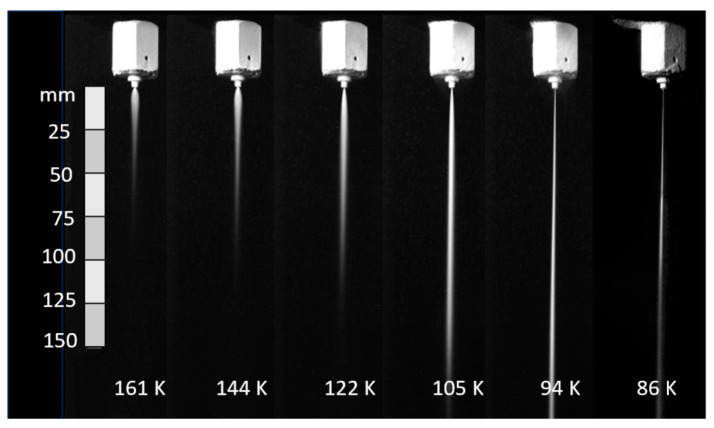
Effect of upstream cooling on nitrogen jet structures.

**Figure 112 materials-17-03273-f112:**
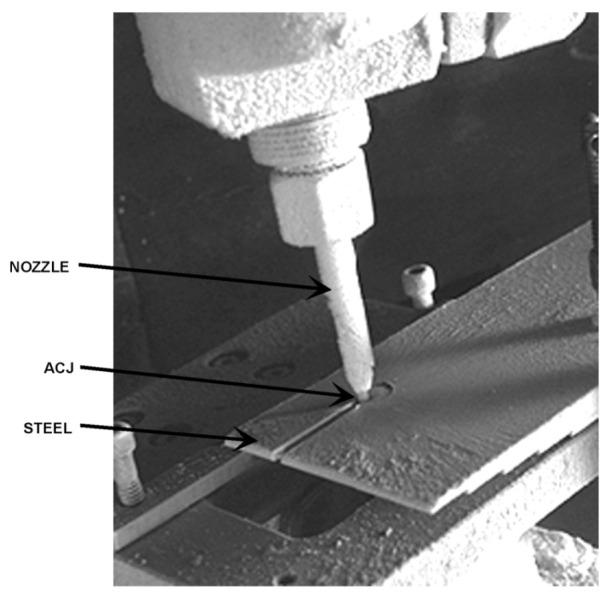
Steel plate cut with a liquid nitrogen ACJ.

**Table 2 materials-17-03273-t002:** Mixing tube length for *λ* = 0.9.

Mesh No.	$r={\dot{m}}_{a}/{\dot{m}}_{w}$	0.1	0.12	0.15	0.2	0.25
	*d_p_* (mm)	Mixing Tube Length, *l_m_* (mm)				
16	1.65	439	423	401	369	340
36	0.76	202	195	185	170	157
60	0.38	101	98	93	85	78
80	0.25	67	65	62	57	52
100	0.13	34	33	31	28	26

**Table 3 materials-17-03273-t003:** Sample cutting speeds (mm/s) for selected ceramics and CMCs [61] at *P* = 345 MPa, *d_n_* = 0.299 mm, *d_m_* = 0.762 mm, and 80-mesh garnet.

Material	Thickness (mm)						
	0.8	1.6	3.2	6.4	12.7	19.1	50.8
Toughened zirconia		0.9		0.7	0.4	0.3	
Dense zirconia			0.8	0.7			
SiC fiber in SiC		1.1	0.6	0.5			
ZrO_2_-MgO			0.8	0.7			
Al_2_O_3_/CoCrAly [80%/20%]			1.0	0.7			
Al_2_O_3_/CoCrAly [60%/40%]			1.0	0.7			
C-glass	100.0	90.0	80.0	60.5	40.0	20.0	6.0
Al_2_O_3_/SiC [7.5%]			2.7	1.4			
SiC/TiB_2_ [15%]			0.3	0.2			

**Table 4 materials-17-03273-t004:** Cutting results using diamond abrasives (*).

Material	Thickness or Depth	Width		
		Top	Bottom	Wall Taper
	mm	mm	mm	Degree
US Synthetic Carbide/PCD (*)	9.53	1.09	0.58	1.53
ROCTEC 500 plate (WC composite)	7.49	1.70	0.58	4.27
Whisker-Reinforced Ceramic	7.11	1.85	0.74	4.49
Al_2_O_3_ + TiC blank	6.10	2.03	0.91	5.24
WC w/10% Cobalt Binder Blank	5.08	2.03	0.74	7.27
AMALOX 87 Alumina Ceramic	6.35	2.24	1.17	4.80
AMZIROX 86 HIPed Zirconia Ceramic	6.35	2.31	0.91	6.28

*P* = 414 MPa; *d_n_* = 0.460 mm; *d_m_* = 1.0 mm; *ma* = 7.5 g/s; diamond powder; *u* = 5.1 mm/s. (*) 2 mm thick PCD, total button thickness of 13 mm.

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
