# Peer review of "Abrasive Waterjet Machining"

_materials, 2024, doi:10.3390/ma17133273_

Round 1

Reviewer 1 Report

Comments and Suggestions for Authors

The article is remarkably interesting and broadly presents the AWJ technology. The author presented a description of AWJ process, and its influencing parameters are in this paper along with process models for the AWJ tool itself and also for the jet-material interaction. Example applications and basic models for this process also are presented in this paper. To demonstrate the versatility of the AWJ machining process, data in this paper cover a wide range of materials such as metal, glass, composites, and ceramics and also a wide range of thicknesses from 1 mm to 600 mm. The paper focused on presenting actual industrial applications to present advances in the technology market meeting challenging demands of accuracy, speed, and cost. Future trends of industry 5.0, AI, and IOT are also presented.

The paper perfectly fits the profile of the Materials journal. I haven’t fundamental doubts, but paper requires improving in following range:

Noticed errors

1.       All equation numbers should be in round brackets, not square brackets.

2.       Page17. Is: 300-mm thick; should be 300 mm thick. Needs to be improved throughout the work.

3.       Page 26 below Figure 25. Kerf width reduction must be converted to mm. Needs to be improved throughout the work.

4.       Page 34 top. 0.75 lb./min garnet abrasives flow rate must be converted to kg/min

5.       Subchapter 6.1.2 Thick Materials. The thickness of materials should be converted from inches and expressed in millimeters.

6.       Figure 61. Bad reference to literature. Is: [2022], should be (probably) [82]

7.       Figure 64. Requires corrections. In the left drawing, the numbers on the vertical axis are difficult to read. Additionally, converting units from inches to millimeters, even roughly assuming that 1 inch approximately equals 25 mm are needed.

8.       Page 68 bottom. Is: A 1.22 m cylinder was milled; should be: A 1.22 m diameter cylinder was milled.

9.       Page 69. 400 micro inch surface finish needs conversion to mm.

10.   Page 72 bottom. Which means 60 @m/pass and 38-50 @m Ra?

11.   Page 82. What is the relationship of the [S-N] curves to Figure 106?

12.   Page 83.

Figure xx means which one? Logic would dictate Figure 108.

Subchapter 6.11 100 Ksi pure waterjet cutting. Pressure level must be converted to MPa

Small errors, I hope typographic only:

1.       Pages 7/8, 21/22, 28/29, 41/42, 43/44. Bad pagination.

2.       The citation of illustrations should be uniform throughout the text. For example, Figure 26 is in bold font and Figure 27 is in standard font.

3.       Figure 27 caption. Is: Figure 27; should be: Figure 27

4.       Page 51.

Is: Figure 59(right); should be: Figure 59 (right)

Is: Figure 60shows; should be: Figure 60 shows

5.       Page 61 bottom.

Is Al2O3/ Al2O3; should be: Al2O3/ Al2O3.

Is: material..; should be: material.

6.       Page 69 top. Is: 20-KW; should be: 20-kW

7.       Page 76. Is: [99-102-103- 104], should be: [99-102,103,104], or [99,102-104]

8.       Page 82. Is: [113-114-115]; should be: [113-115], or [113,114,115].

9.       Page 86. Is: CO2; should be: CO2

10.   Page 88 top. Is: 7. final remarks; should be: 7. Final remarks

Author Response

  1. All equation numbers should be in round brackets, not square brackets.
    Response: done
  2. Page17. 300-mm thick; should be 300 mm thick. Needs to be improved throughout the work.
    Response: done in all the document
  3. Page 26 below Figure 25. Kerf width reduction must be converted to mm. Needs to be improved throughout the work.
    Response: corrected in all the document.
  4. Page 34 top. 0.75 lb./min garnet abrasives flow rate must be converted to kg/min.
    Response: done and in all other occurrences.
  5. Subchapter 6.1.2 Thick Materials. The thickness of materials should be converted from inches and expressed in millimeters.
    Response: corrected.
  6. Figure 61. Bad reference to literature. Is: [2022], should be (probably) [82]
    Response: corrected but also added more references

7.Figure 64. Requires corrections. In the left drawing, the numbers on the vertical axis are difficult to read. Additionally, converting units from inches to millimeters, even roughly assuming that 1 inch approximately equals 25 mm are needed.
Response: done.

8.Page 68 bottom. Is: A 1.22 m cylinder was milled; should be: A 1.22 m diameter cylinder was milled.
Response: corrected.

9.Page 69. 400 micro inch surface finish needs conversion to mm.
Response: done in all the document.

  1. Page 72 bottom. Which means 60 @m/pass and 38-50 @m Ra?
    Response: corrected as the font was wrong.
  2. Page 82. What is the relationship of the [S-N] curves to Figure 106?
    Response: Figure 106 shows S (stress) versus N (number of cycles)
  3. Page 83. Figure xx means which one? Logic would dictate Figure 108.
    Response: corrected as Figure 107
  4. Subchapter 6.11 100 Ksi pure waterjet cutting. The pressure level must be converted to MPa.
    Response: done

Other Edits

1.Pages 7/8, 21/22, 28/29, 41/42, 43/44. Bad pagination.
Response: This comment is not clear as page numbers seem to be in sequence.

2. The citation of illustrations should be uniform throughout the text. For example, Figure 26 is in bold font and Figure 27 is in standard font.
Response: done for all Figures.

3. Figure 27 caption. Is: Figure 27; should be: Figure 27
Response: done

4 .Page 51. Is: Figure 59(right); should be: Figure 59 (right), Is: Figure 60shows; should be: Figure 60 shows
Response: done

5. Page 61 bottom. Is Al2O3/ Al2O3; should be: Al2O3/ Al2O3.
Response: corrected in all occurrences.

Is: material..; should be: material.
Response: corrected.

6.Page 69 top. Is: 20-KW; should be: 20-kW
Response: kW is used instead of KW or Kw in all occurances.

7.Page 76. Is: [99-102-103- 104], should be: [99-102,103,104], or [99,102-104]
Response: done

8.Page 82. Is: [113-114-115]; should be: [113-115], or [113,114,115].
Response: done

9. Page 86. Is: CO2; should be:CO2
Response: done

10. Page 88 top. Is: 7. final remarks; should be: 7. Final remarks
Response: done

Reviewer 2 Report

Comments and Suggestions for Authors

This paper gives a comprehensive review on the sphere of Abrasive Waterjet Machining, which is well edited. In my perspective, this paper can be published after minor revision listed below.

1. In Part 4.1 on page 11, there is an editing error, please check it.

2. Please check if Equation [13],[14] and [15] are wrongly written.

3. Clearer Figures should be given, e.g.,  Figure 10, Figure 11, Figure 29, etc.

4. Table 1 seems to be incomplete.

Author Response

In Part 4.1 on page 11, there is an editing error, please check it.
Response: Corrected

  1. Please check if Equation [13],[14] and [15] are wrongly written.
    Response: The proportionality symbol was clarified
  2. Clearer Figures should be given, e.g., Figure 10, Figure 11, Figure 29, etc.
    Response: Figures were edited for better clarity
  3. Table 1 seems to be incomplete.
    Response: The table was out of margin and is corrected

Reviewer 3 Report

Comments and Suggestions for Authors

This manuscript has introduced the Abrasive Waterjet (AWJ) machining system, including the components and and its machined surface integrity, as well as AWJ applications. The topic is appropriate, however, as a journal review, the current manuscript is overloaded with basic knowledge and lacks in-depth summarization of scientific research findings. These also lead to the whole review is more inclined to carry the results of other people's research, and the pile of text not only did not provide convenience to the readers, but also difficult to understand the current progress of the AWJ field. All in all, a major revision is recommended before publication.

1.The content of the abstract is inappropriate and does not exhibit the main issues covered in the manuscript.

2.Sections 2 and 3 contain a presentation of AWJ structure, abrasives, and also processing evaluation indexes, mechanisms, etc. The overall is numerous and complex, so it is recommended to streamline the subheadings and content according to the framework. It is advised to refer to the existing review reports in the AWJ field.

3.There are missing informations in section 3.2 and it is required to be supplemented.

4.Instead of simply listing the results of existing studies, authors are need to summarize the findings as well as provide objective insights. Such as the equations in section 3.3.1, is it the most advanced theoretical model, or any help to the current development of the industry? What are the advantages of these equations over previous studies? These aforementioned exists in the full text.

5.Be sure to carefully check the format and fonts of the whole text, e.g. the font size of the level 4 subheading (in section 6) is not consistent with the previous one.

6.The conclusion section lacks substantive insights and an outlook on the AWJ field.

7.Reference citations need to be checked, e.g. in page 11 (section 4.1.), an error has occurred. In addition, the references are largely classic studies from before the 21st century and there are hardly any reports from the last 5 years.

Comments on the Quality of English Language

Minor editing of English language is required.

Author Response

1. The content of the abstract is inappropriate and does not exhibit the main issues covered in the manuscript.

Response: The abstract has been re-written to better reflect the contents of the manuscript

2. Sections 2 and 3 contain a presentation of AWJ structure, abrasives, and also processing evaluation indexes, mechanisms, etc. The overall is numerous and complex, so it is recommended to streamline the subheadings and content according to the framework. It is advised to refer to the existing review reports in the AWJ field.

Response:  Yes, it was confusing, and it has been corrected with better streamlined headings.  Also, additional review references have been added.

3. There are missing information in section 3.2, and it is required to be supplemented.

Response: Issue corrected as a portion of the table was outside the margins.

4.I nstead of simply listing the results of existing studies, authors are needed to summarize the findings as well as provide objective insights. Such as the equations in section 3.3.1, is it the most advanced theoretical model, or any help to the current development of the industry? What are the advantages of these equations over previous studies? These aforementioned exists in the full text.

Response: The equations in section 3.1.1 are basic fluid mechanics equations, however, the point of the reviewer is well taken regarding the other equations.  The equations presented in this paper are mainly from the author’s work, mostly have not been addressed by others.  For example, the width of the jet, equation 25 has no equivalent in the literature as no reference was found addressing the width of the jet.  The second paragraph in the introduction has been modified to reflect this.

5.Be sure to carefully check the format and fonts of the whole text, e.g. the font size of the level 4 subheading (in section 6) is not consistent with the previous one.

Response: Checked and corrected.

6. The conclusion section lacks substantive insights and an outlook on the AWJ field.

Response:  The final remarks are not intended by concluding remarks but rather a reflection on the current and future status.  Making a list of conclusions is possible but will be long to cover the many topics of this paper.  However, if a conclusions section is essential, then I will add a section.

7. Reference citations need to be checked, e.g. in page 11 (section 4.1.), an error has occurred. In addition, the references are largely classic studies from before the 21st century and there are hardly any reports from the last 5 years.

Response: References have been checked.  Also, additional more recent review papers and references have been added. 

Round 2

Reviewer 3 Report

Comments and Suggestions for Authors

In revised draft, the all issues recommended by the reviewers have been answered, so I suggest the paper publishing in Journal.